# A global climatological perspective on the importance of Rossby wave breaking and intense moisture transport for extreme precipitation events

Andries Jan de Vries[1,2]

[1]ETH Zürich, Institute for Atmospheric and Climate Science, Zürich, Switzerland
[2]Max Planck Institute for Chemistry, Atmospheric Chemistry Department, Mainz, Germany

*Correspondence to*: Andries Jan de Vries (andries.devries@env.ethz.ch)

**Abstract.** Extreme precipitation events (EPEs) frequently cause flooding with dramatic socioeconomic impacts in many parts of the world. Previous studies considered two synoptic-scale processes, Rossby wave breaking and intense moisture

transport, typically in isolation, and their linkage to such EPEs in several regions. This study presents for the first time a global and systematic climatological analysis of these two synoptic-scale processes, in tandem and in isolation, for the occurrence of EPEs. To this end, we use 40-year ERA-Interim reanalysis data (1979-2018) and apply object-based identification methods for (i) daily EPEs, (ii) stratospheric potential vorticity (PV) streamers as indicators of Rossby wave breaking, and (iii) structures of high vertically integrated horizontal water vapor transport (IVT). First, the importance of

these two synoptic-scale processes is demonstrated by case studies of previously documented flood events that inflicted catastrophic impacts in different parts of the world. Next, a climatological quantification shows that Rossby wave breaking is associated with > 90 % of EPEs over central North America and the Mediterranean, whereas intense moisture transport is linked to > 95 % of EPEs over many coastal zones, consistent with findings of atmospheric river-related studies. Combined Rossby wave breaking and intense moisture transport contributes up to 70 % of EPEs in several subtropical and extratropical

regions, including (semi)arid desert regions where tropical-extratropical interactions are of key importance for (heavy) rainfall. Odds ratios of EPEs linked to the two synoptic-scale processes suggest that intense moisture transport is stronger associated with the occurrence of EPEs than Rossby wave breaking. Furthermore, the relationship between the PV and IVT characteristics and the precipitation volumes shows that the depth of the wave breaking and moisture transport intensity are intimately connected with the extreme precipitation severity. Finally, composites reveal that subtropical and extratropical

EPEs, linked to Rossby wave breaking, go along with the formation of upper-level troughs and cyclogenetic processes near the surface downstream, reduced static stability beneath the upper-level forcing (only over water), and dynamical lifting ahead (over water and land). This study concludes with a concept that reconciles well-established meteorological principles with the importance of Rossby wave breaking and intense moisture transport for the formation of EPEs. Another conclusion with major implications is that different combinations of Rossby wave breaking and intense moisture transport can reflect a

large range of EPE-related weather systems across climate zones, and can thus form the basis for a new classification of EPE

regimes. The findings of this study may contribute to an improved understanding of the atmospheric processes that lead to EPEs, and may find application in climatic studies on extreme precipitation changes in a warming climate.

## 1       Introduction

Extreme precipitation events (EPEs) frequently cause dramatic socioeconomic impacts in many parts of the world. They can induce flash floods, riverine floods, landslides, and debris flows, resulting in loss of life and damage to infrastructure and property (Ashley and Ashley, 2008; Barredo, 2007; Terti et al., 2017). For example, in the year 2017, floods were responsible for the largest share in fatalities (35 %) and affected people (60 %) due to natural disasters (EM-DAT, 2018). Records of the Emergency Events Database (EM-DAT, from the Centre for Research on the Epidemiology of Disasters,

publicly available at https://www.emdat.be, assessed on 15 April 2020) indicate that floods left worldwide more than 275 million fatalities, affected almost 8 billion people through injuries or loss of homes, and caused a damage of about 8.9 trillion US dollars during the period of 1900-2019. These devastating societal impacts underline the importance of an improved understanding of the atmospheric processes that lead to the formation of EPEs.

Synoptic weather systems that drive EPEs vary across climate zones. In the tropics and lower latitudes, EPEs are typically associated with tropical cyclones (Khouakhi et. al., 2017; Franco-Diaz et al., 2019), tropical easterly waves (Ladwig and Stensrud, 2009; Cretat et al., 2015), and monsoon lows/depressions (Hurley and Boos, 2015). Baroclinic systems dominate in extratropical regions, including extratropical cyclones, warm conveyor belts, and fronts (Hawcroft et al., 2012; Pfahl and Wernli, 2012; Catto and Pfahl, 2013; Papritz et al., 2014; Pfahl et al., 2014; Catto et al., 2015). Other well-known

phenomena that predominantly affect higher latitude regions are tropical moisture exports (Knippertz and Wernli, 2010; Knippertz et al., 2013) and atmospheric rivers (Lavers and Villarini, 2013a,b; Waliser and Guan, 2017), whereas low-level jets are often found at lower latitudes (Chen and Yu, 1988; Monaghan et al., 2010). It is worth to point out here that these weather systems should not be interpreted as independent phenomena, but point to processes that are intertwined with another and share many similarities (Knippertz et al., 2013; Dacre et al., 2015, 2019).

Weather systems that can lead to EPEs in subtropical regions are much less well-known. Precipitation and heavy precipitation events in these typically dry regions have been associated with tropical plumes (Wright, 1997; Knippertz and Martin, 2005; Rubin et al., 2007), referring to elongated middle and upper-level cloud bands that reach from the tropics in a poleward and eastward direction (McGuirk et al., 1988). Other phenomena are bound to specific regions such as monsoon

surges and moisture burst in southwestern North America, the southwestern slopes of the Himalaya's, and Australia (Favors and Abatzoglou, 2013; Martius et al., 2013; Pascale and Bordoni, 2016; Berry and Reeder, 2016; Vellore et al., 2016),

tropical temperate troughs in southern Africa (Todd and Washington, 1999; Hart et al., 2013), and the active Red Sea Trough in the Middle East (Kahana et al., 2002; De Vries et al., 2013). These weather systems typically involve tropical-extratropical interactions, whereby the extratropical forcing interacts with the tropical circulation through the intrusion of an upper-level trough into low latitudes (Knippertz, 2007; De Vries et al., 2018). As a result, a poleward incursion of tropical moisture reaches into the dry subtropics where it can support the formation of EPEs.

Several objective identification methods have been developed to detect and investigate these synoptic weather systems. For example, automated tools detect tropical plumes based on the geometry of cloud structures in satellite-based observations (Hart et al., 2012; Fröhlich et al., 2013). Eulerian approaches have been applied to reanalysis data and model output for the identification of extratropical cyclones based on minima in sea level pressure or maxima of low-tropospheric relative vorticity (Wernli and Schwierz, 2006; Neu et al., 2013), fronts inferred from changes in the horizontal wind direction or horizontal wet-bulb or equivalent potential temperature gradients in the lower troposphere (Berry et al., 2011; Simmonds et al., 2012; Schemm et al., 2015), and atmospheric rivers using vertically integrated tropospheric moisture content and/or moisture transport (see below). Lagrangian approaches served the detection of warm conveyor belts based on ascending air parcel trajectories (Eckhardt et al., 2004; Madonna et al., 2014) and tropical moisture exports defined by moist and poleward-moving air parcel trajectories (Knippertz and Wernli, 2010). A comprehensive overview of the identification and climatology of such Eulerian and Lagrangian flow features is presented by Sprenger et al. (2017).

Another meteorological process that has been associated with EPEs is Rossby wave breaking. This process refers to the non-linear phase of planetary wave amplification, which can result in irreversible mixing of air masses from stratospheric high-latitude origin into tropospheric low-latitude regions, and vice versa, as observed in early studies (McIntyre and Palmer, 1983; Appenzeller and Davies, 1992). In a potential vorticity (PV) framework, Rossby wave breaking can be diagnosed from the formation of elongated PV filaments, referred to as PV streamers, that eventually can split off from the main stratospheric (tropospheric) air reservoirs over the poles (tropics), known as cutoffs (Hoskins et al., 1985; Wernli and Sprenger, 2007; Portmann et al., 2020a). Such PV streamers and cutoffs can favour EPE formation through inducing enhanced moisture transport and forcing for ascent via reduced static stability beneath the positive upper tropospheric PV anomaly and dynamical lifting ahead (Funatsu and Waugh, 2008; Schlemmer et al., 2010). Accordingly, many case studies have linked Rossby wave breaking to EPEs in several regions, including, but not limited to, southwestern North America, northwest Africa, the Alpine region, the Mediterranean, southern Africa, the Middle East, Pakistan and the Himalaya's, and East Asia (Massacand et al., 1998; Knippertz and Martin, 2005, 2007; Argence et al., 2006; Hart et al., 2010; Martius et al., 2013; De Vries et al., 2016; Vellore et al., 2016; Portmann et al., 2020b; Tsuji and Takayabu, 2019). Systematic climatological analyses; however, are limited to the Alpine region (Martius et al., 2006), the Middle East (De Vries et al., 2018), South Africa (Favre et al., 2013), and North America (Abatzoglou, 2016; Barbero et al., 2019; Moore et al., 2019).

Therefore, it is not yet known in which other parts of the world and to what extent Rossby wave breaking is of high relevance to the formation of EPEs.

All aforementioned weather systems that have been associated with EPEs have at least one aspect in common; they drive intense moisture transport to the region of extreme precipitation. Moisture transport is often quantified in a Eulerian 100   framework using vertically integrated horizontal water vapor transport (IVT), a diagnostic that became very popular during the last decade. For example, IVT has successfully served in numerous studies to identify so-called atmospheric rivers (e.g., Newell et al., 1992; Lavers et al., 2012; Guan and Waliser, 2015; Mundhenk et al., 2016a) that have been linked to precipitation extremes and flooding in many parts of the world, in particular the West Coast of North America and Western Europe (Ralph et al., 2004, 2006; Lavers et al., 2013a; Waliser and Guan, 2017). Also, IVT has been directly related to 105   precipitation extremes (Moore et al., 2015; Froidevaux and Martius, 2016; Grazzini et al., 2020) and served as a proxy to elucidate moisture transport pathways that feed heavy rainfall events (Swales et al., 2016; Tan et al., 2019). Furthermore, recent studies employed IVT to improve the medium-range prediction of hydrometeorological extremes (Lavers et al., 2014, 2016; Mahlstein et al., 2019) and to assess (thermo)dynamical changes of precipitation extremes in projected future climates (Lavers et al., 2015; Espinoza et al., 2018; Benedict et al., 2019; Hsu and Chen, 2020; Sousa et al., 2020). For extensive 110   reviews on the link between intense moisture transport and EPEs, the reader is referred to Gimeno et al. (2016) and Liu et al. (2020).

Although the focus on intense moisture transport and the use of IVT is very prominent in studies of extreme precipitation, investigations of the large-scale circulation processes that drive the moisture transport often remain more in the background 115   and less understood. Few studies combined the use of PV and IVT, and linked Rossby wave breaking to moist air incursions into the Arctic (Liu and Barnes, 2015), and to the formation and landfall of atmospheric rivers (Payne and Magnusdottir, 2014, 2016; Hu et al., 2017; Zavadoff and Kirtman, 2020). Several extreme precipitation-related studies took Rossby wave breaking as a starting point, and linked this process to enhanced moisture transport (Martius et al., 2006; Ryoo et al., 2013; Moore et al., 2019), whereas others took the perspective of moisture transport, and identified some form of upper-level 120   forcing in the shape of upper-level troughs and/or cutoffs upstream, suggesting the occurrence of Rossby wave breaking (e.g., Neiman et al., 2008; Ralph et al., 2011; Mundhenk et al., 2016b). So far, however, a systematic evaluation of the individual and combined importance of Rossby wave breaking and intense moisture transport for EPEs is limited to the Middle East region (De Vries et al., 2018) and remains unexplored at the global scale.

This study presents for the first time a global and systematic climatological analysis of the individual and combined importance of Rossby wave breaking and intense moisture transport for EPEs. The motivations for this approach are threefold: (1) previous studies that linked Rossby wave breaking to EPEs were confined to specific regions and are here

complemented by a global analysis, (2) the combined application of PV and IVT flow features represent two key larger-scale (thermo)dynamic processes that can lead to the formation of EPEs, and (3) this setup facilitates a process-based evaluation of the forcing mechanism through which these two synoptic-scale processes can bring about EPEs. Note that the focus of this study is primarily on the synoptic-scale processes that shape the tropospheric environment in which these EPEs develop, whereas mesoscale processes that determine the precise timing, location and organization of convective storms are beyond the scope of this work.

This study addresses the following three research questions:

1) How often and where are EPEs linked to Rossby wave breaking, intense moisture transport, and their combined occurrence?
2) How do the characteristics of these two synoptic-scale processes relate to the extreme precipitation severity?
3) How do the tropospheric environments (e.g., forcing mechanism of upward motion) contribute to the formation of EPEs for different constellations of the two synoptic-scale processes?

To this end, we use reanalysis data and apply state-of-the-art object-based identification methods for (i) daily EPEs, (ii) stratospheric PV streamers as indicators of Rossby wave breaking, and (iii) structures of high IVT that represent intense moisture transport (Fig. 1). We qualitatively demonstrate the importance of these two synoptic-scale processes for EPEs with several infamous flood events in different parts of the world that were documented in previous case studies, and then proceed with a climatological quantification at the global scale. First, we explore the geographical distribution of EPEs that are linked to the separate and combined occurrence of Rossby wave breaking and intense moisture transport. Next, we relate the strength of the wave breaking and the intensity of the moisture transport to the extreme precipitation severity. Furthermore, we use composites to elucidate the tropospheric environments and forcing mechanism for ascent that contribute to the formation of EPEs in different regions and for different constellations of PV streamers and IVT structures that coincide with the EPEs. These processes are then summarized in a concept that reconciles longstanding meteorological principles with the importance of Rossby wave breaking and intense moisture transport for EPEs. The findings of this study also form the basis for a new classification of EPE-related weather regimes that reflect a large range of relevant weather systems across different climate zones. This new perspective on the atmospheric processes of EPEs can have large implications for the predictability of precipitation extremes and opens up new avenues to study their future changes in a warming climate.

The organization of the paper is as follows. The reanalysis data and object-based identification methods are described in section 2. Section 3 presents illustrative case studies. Section 4 presents the climatology of Rossby wave breaking and intense moisture transport, and then links these processes to EPEs. Section 5 elaborates on the relationship between the extreme precipitation severity and the characteristics of the PV and IVT structures. Section 6 uses composites of the

tropospheric circulation to elucidate the relevant meteorological processes that contribute to the formation of EPEs. Finally, section 7 presents a synthesis, followed by the conclusions in section 8.

## 2.	Methods and data

### 2.1	Reanalysis data and key diagnostics

We use the ERA-Interim reanalysis dataset from the European Centre for Medium-range Weather Forecasts (ECMWF) for the identification of daily EPEs, stratospheric PV streamers, and structures of high IVT. The reanalysis dataset provides a best estimate of the global atmospheric state based on a forecast model integration with the Integrated Forecast System (IFS, release Cy31r2) and the assimilation of several observation types (Dee et al., 2011). The ERA-Interim data is used on its native N128 Gaussian grid (512 and 256 grid points in longitudinal and meridional directions, respectively), 60 model levels

and 37 pressure levels in the vertical direction, and 6-hourly time intervals for a 40-year period from 1979 to 2018.

Over the last decades, PV has proven to be invaluable in the field of dynamical meteorology. PV is materially conserved under adiabatic and frictionless conditions and contains the full information of a balanced flow, based on the so-called "PV invertibility" principle (Hoskins et al., 1985). These properties qualify PV as very useful for studying meteorological

processes at synoptic-scale scales such as Rossby wave breaking. PV is proportional to the scalar product of absolute vorticity $\boldsymbol{\zeta}_a$ and the gradient of potential temperature $\theta$

$$PV = \rho^{-1}\boldsymbol{\zeta_a} \cdot \nabla\theta \qquad (1)$$

where $\rho$ is the density. We compute PV using the wind and temperature on model levels and surface pressure, and then interpolate the PV fields onto isentropic surfaces between 300 and 350 K with 5 K intervals.

IVT quantifies the horizontal atmospheric moisture transport in a Eulerian framework, and this vector is defined by (Newell et al., 1992)

$$\mathbf{IVT} = g^{-1}\int_{P_{\text{bottom}}}^{P_{\text{top}}} q\mathbf{v}\,\mathrm{d}p \qquad (2)$$

where $g$ is the gravitational acceleration, $q$ the specific humidity, v the horizontal wind, and $p$ pressure. The zonal and meridional components of this vector are readily available in the ERA-Interim dataset (Berrisford et al., 2011), and the IVT

magnitude follows from

$$IVT = \sqrt{IVT_{zonal}^2 + IVT_{meridional}^2} \qquad (3)$$

Furthermore, we adopt several other diagnostics to study the larger-scale meteorological environment in which the EPEs develop, including the atmospheric moisture content, static stability, and quasi-geostrophic vertical motion. The atmospheric moisture content, also known to as total column water (TCW) or precipitable water, provides the necessary fuel for precipitation generation and is defined by

$$TCW = g^{-1} \int_{P_{bottom}}^{P_{top}} q \, dp \qquad (4)$$

and is also readily available in the ERA-Interim reanalysis dataset.

Synoptic-scale processes, such as upper-level PV anomalies, can reduce the tropospheric stratification beneath the anomaly, and can thus favour convective overturning if the static stability is sufficiently reduced and some form of initial lifting is provided (Doswell, 1996; Funatsu and Waugh, 2008; Schlemmer et al., 2010). Static stability can be expressed in several forms (Gates, 1960). We follow the approach of Funatsu and Waugh (2008), and compute static stability as

$$S = -g \frac{\partial \theta}{\partial p} \qquad (5)$$

between the 850 and 500 hPa pressure levels, which represents the stratification of the lower troposphere.

Several case studies (Knippertz and Martin, 2005; Hart et al., 2010; Martius et al., 2013; De Vries et al., 2016) and climatological analyses (Agel et al., 2019; Moore et al., 2019; Nie and Fan, 2019; Dai and Nie, 2020) found that EPEs occur in regions with enhanced quasi-geostrophic (QG) forcing for ascent. QG ascent, also often referred to as dynamical lifting, large-scale ascent, or balanced ascent, can be quantified through inversion of the QG omega equation in the Q-vector form (Hoskins et al., 1978; Davies, 2015). The QG vertical motion can be decomposed in contributions due to the forcing by the upper and lower tropospheric circulations, respectively. Here, as in Graf et al. (2017) and Crespo et al. (2020), we use the QG vertical motion attributed to the upper-tropospheric circulation (< 600 hPa) with the specific purpose to quantify the contribution of the upper-level forcing to the QG ascent in the regions of EPEs.

## 2.2 Extreme precipitation objects

Previous studies adopted various approaches and criteria to define precipitation extremes. Common criteria, applied to daily or multiple-day precipitation amounts, are (i) absolute thresholds, (ii) percentile-based thresholds, and (iii) monthly or annual maxima (Zhang et al., 2011; Barlow et al., 2019). Also, with regard to their spatial characteristics, several approaches have been used to define extremes, including (1) individual grid points (e.g., Pfahl and Wernli, 2012), (2) area averages for specific regions of interest (Martius et al., 2006; De Vries et al., 2018), (3) spatially smoothed fields using a spatially-moving average within a defined radius (Skok et al., 2009; Raveh-Rubin and Wernli, 2015), and (4) spatially coherent objects that consist of multiple co-located extreme precipitation grid points (Moore et al., 2015, 2019; Mahoney et al., 2016).

In this study, we define EPEs as objects on daily timescales consisting of grid points that exceed a percentile-based threshold and that are located within a certain distance of another. This procedure consists of the following four steps, as demonstrated in Fig. 2 for a few EPEs that will be discussed in section 3. First, daily precipitation sums are constructed from 12 h forecasts from 00 and 12 UTC (Fig. 2a-d). Next, extreme precipitation is defined by grid points that exceed the annual 99[th] percentile of daily precipitation during the period of 1979-2018 (Fig. 2e-h). Then, all extreme precipitation grid points within a radius of 250 km are connected with another to obtain spatially coherent objects. Objects with a surface area < 50,000 km$^2$ are removed from the selection to focus on the larger-scale events (Fig. 2i-l). Finally, characteristics of the remaining objects are retrieved, such as the total precipitation volume and the mass-weighted centre (Fig. 2m-o).

The choice to use reanalysis data for defining extreme precipitation objects has several advantages, but also few limitations that deserve some consideration. Precipitation in the ERA-Interim reanalysis is a forecasted quantity and not directly constrained by observations in the data assimilation procedure. In this regard, precipitation may be interpreted as a modeling product with inherent uncertainties, rather than actual observed precipitation. Furthermore, as a result of a relatively coarse model resolution, reanalysis precipitation typically exhibits larger spatial patterns with longer durations and lower rainfall intensities as compared to precipitation in high-resolution observational products (Donat et al., 2014; White et al., 2017). Observational precipitation products; however, are subject to several other limitations. Rain gauge measurements, and gridded datasets based on these measurements, are confined to land areas, suffer from regions with a low-density station network, and can have inhomogeneous time series (e.g., Donat et al., 2014; Sun et al., 2017). Satellite-based estimates depend on algorithm retrievals with relatively large uncertainties over (semi)arid regions and high topography (Dinku et al., 2011; Zambrano-Bigiarini et al., 2017), typically span relatively short periods from 1998 onwards, and cover predominantly tropical and subtropical regions (Sun et al., 2017). The use of precipitation from reanalysis data has the highly desired advantage of a full spatial and temporal coverage, which facilitates a worldwide assessment of the link between EPEs and synoptic-scale processes for an extensive 40-year period. Moreover, the shortcomings of reanalysis precipitation have only a limited influence on our EPEs as they are based on normalized fields that generally show a better comparison to those in

other data product types than actual values (Donat et al., 2014), and are defined by relatively large spatiotemporal scales (daily, > 50.000 km$^2$) that makes the EPE objects less sensitive to the highly variable nature of precipitation.

## 2.3    Stratospheric PV streamers

Previous studies have followed two different approaches for detecting PV structures as indicators of Rossby wave breaking. One approach is based on the meridional reversal of the PV gradient on an isentropic surface (Postel and Hitchman, 1999). Similarly, Strong and Magnusdottir (2008) used meridionally overturning PV contours, which found application in aforementioned studies that linked Rossby wave breaking to landfalling atmospheric rivers (Payne and Magnusdottir, 2014, 2016; Mundhenk et al., 2016b; Hu et al., 2017; Zavadoff and Kirtman, 2020). Another method, introduced by Wernli and Sprenger (2007), searches for an elongated shape of PV contours, referred to as PV streamers, often leading to so-called PV cutoffs when these structures split off from the main PV reservoir. This approach was refined in Sprenger et al. (2013) and adopted in several studies that linked Rossby wave breaking to EPEs in the Alpine region, the Middle East, and North America (Martius et al., 2006; De Vries et al., 2018; Moore et al., 2019). In this study, we apply the algorithm from De Vries et al. (2018), that follows the approach of Wernli and Sprenger (2007), to both hemispheres. For the sake of clarity and completeness, we describe the algorithm briefly, with special emphasis on some new aspects and modifications performed for this study.

For the identification of PV streamers, we use +/–2 potential vorticity unit (PVU; 1 PVU = $10^{-6}$ K kg$^{-1}$ m$^2$ s$^{-1}$) contours on isentropic surfaces between 300 and 350 K with 5 K intervals. Note that PV has generally negative values in the Southern Hemisphere, and that we will use absolute PV values for consistency in both the text and figures hereafter. The 2 PVU surface is usually considered as the dynamical tropopause, which adequately reflects transient upper-tropospheric disturbances. In a first stage, we define the stratospheric body in each hemisphere by the most equatorward 2 PVU contour that encircles the pole. If such a contour does not exist, we appoint the longest 2 PVU contour as stratospheric body, provided that this contour extends over more than 180 degrees in the zonal direction and reaches at least partly poleward of 80° latitude. This study only considers stratospheric PV streamers (PV > 2 PVU) and ignores their tropospheric counterparts.

In a second stage, we evaluate for each pair of contour points on the 2 PVU contour that demarcates the stratospheric body if the (1) width $W < 1500$ km, (2) the length $L > 1000$ km, and (3) their ratio $r = L/W > 1$, see also the schematic in Fig. 1. If all three criteria are satisfied, then the surface between the two contour points A and B and the corresponding contour segment is considered as a PV streamer. The distances $W$ and $L$ are measured by the great-circle distances between points A and B, and C and D, respectively (Fig. 1). The chosen geometrical criteria are the strictest in the sensitivity tests of De Vries et al. (2018) and intend to limit the detection of PV streamers to those that exhibit a very elongated form. If more than 50 % of the stratospheric body is covered by stratospheric streamers, all PV streamers are removed at that particular time instance and

isentropic surface for that hemisphere, assuming that the 2 PVU contour is too distorted to realistically represent the dynamical tropopause. A further screening removes all PV streamers with a surface area $< 100,000$ km$^2$ to retain only structures with relevance to the larger-scale circulation. Our methodology does not distinguish between anticyclonic and cyclonic wave breaking (Martius et al., 2007), which is considered of secondary importance as both types of wave breaking can be associated with precipitation extremes (Ryoo et al., 2013; Hu et al., 2017; Moore et al., 2019). Figure 3 demonstrates the functionality of the algorithm with PV on the 335 K isentrope in the upper panel and the identified PV streamers in the lower panel.

In a last stage, we assign to each PV streamer an extended area of influence to facilitate the matching of PV streamers with EPEs. Previous studies showed that extreme precipitation typically forms downstream of PV streamers (e.g., Martius et al., 2006; De Vries et al., 2018; Moore et al., 2019). This motivates the use of such an extended area of influence, which is also in line with the notion that PV "acts" at a distance, or in other words, that the atmospheric flow "feels" a remote PV anomaly (Hoskins et al., 1985). We adopt a relatively simple and straightforward geometrical approach to define this extended area. All nearby grid points, within a radius $R$ of any grid point that is part of the PV streamer, are assigned to the extended area of this specific PV streamer (light blue shading in Figs. 1 and 3b). This radius $R$ is variable and proportional to the surface area of the PV streamer, and computed as follows. We determine the equivalent radius $R_{eqv}$ of the PV streamer, assuming a circular shape, and then retrieve the extension of $R_{eqv}$ needed to double the PV surface under a maintained circular shape ($R = 0.414 R_{eqv}$). Hereafter, we refer to the PV streamers with their extended areas as PV structures.

## 2.4    IVT structures

Moisture transport is typically very large over tropical and extratropical oceans where easterly trade winds and the storm tracks dominate. To avoid very frequent and extraordinarily large IVT structures over these regions, we use the following two IVT forms and criteria: (1) a static threshold of 150 kg m$^{-1}$ s$^{-1}$ on filtered IVT whereby the 91-day running mean is subtracted from the full IVT (IVT$_{\text{filt}}$), and (2) exceedances of the annual 95$^{\text{th}}$ percentile of 6-hourly full IVT over the period of 1979-2018 and a minimum threshold of 200 kg m$^{-1}$ s$^{-1}$ (IVT$_{\text{pct}}$). Regarding the latter form, IVT$_{\text{pct}}$ structures thus consists of grid points where full IVT values both exceed the annual 95$^{\text{th}}$ percentile and a lower limit of 200 kg m$^{-1}$ s$^{-1}$. The first form removes the seasonality of IVT and the second form focuses on unusually high IVT. Both forms aim to detect transient structures of high IVT with respect to a varying background IVT. We discuss both types of IVT structures in the first part of our analysis since they have substantially different climatologies, as will be shown in section 4.

Similar to the PV streamers, we only retain IVT structures with a surface area $> 100,000$ km$^2$ with the aim to eliminate the local-scale features and to focus on synoptic-scale processes. Furthermore, the maximum IVT value within the IVT structures is assigned as a characteristic to these structures as to provide a measure of the moisture transport intensity.

Finally, each IVT structure is provided with an extended area (light green colours in Fig. 3b), in the same manner as for the PV structures, to facilitate the matching of remote PV structures with the extreme precipitation objects, as further detailed in section 4. Figure 3 provides an example of an IVT field and the identification of IVT structures in the $IVT_{pct}$ form.

Our identification method for IVT structures is very similar to the approach used in several studies to identify atmospheric rivers, with the notable exception that we do not impose any specific geometrical criteria in terms of the length, width, and length-width ratio (e.g., Guan and Waliser, 2015; Mundhenk et al., 2016a). Our rationale is that structures with high IVT can potentially be relevant to extreme precipitation independent of their specific form. To provide further context for our choice of IVT forms, we briefly list different definitions of atmospheric rivers applied in previous studies. These include (1) full IVT fields with a static threshold ranging between 250 and 500 kg m$^{-1}$ s$^{-1}$ (Rutz et al., 2014; Mahoney et al., 2016; Pasquier et al., 2018), (2) anomalous IVT fields whereby the lowest frequency harmonics are removed to obtain structures of high anomalous IVT, applying a static threshold of 250 kg m$^{-1}$ s$^{-1}$ on these filtered fields (Mundhenk et al., 2016a), (3) percentile-based thresholds, typically the 85$^{th}$ percentile, on global fields with a lower limit of 100 kg m$^{-1}$ s$^{-1}$ (Guan and Waliser, 2015), or over coastlines where atmospheric rivers make landfall (Lavers et al., 2012; Ramos et al., 2015). In this regard, our two IVT forms, $IVT_{filt}$ and $IVT_{pct}$, are very similar to, and inspired by the aforementioned second and third methods to identify atmospheric rivers, respectively.

## 2.5    Matching of EPEs with Rossby wave breaking and intense moisture transport

In this study, we connect EPEs with Rossby wave breaking and intense moisture transport in two different ways (Table 1). One method adopts a so-called "inclusive" matching approach, used in sections 4.2 and 4.3, whereby an EPE can be assigned to both individual and combined synoptic-scale processes to quantify the global contribution of these processes to EPEs. Another method follows an "exclusive" matching approach, used in sections 6.2 and 7, whereby an EPE can only be assigned to one of five different constellations of synoptic-scale processes to isolate their contribution to EPE formation and the associated tropospheric environmental conditions. For both approaches, the matching of EPEs with synoptic-scale structures is defined as follows. Daily extreme precipitation objects match with the PV and/or IVT structures if they spatially overlap the extreme precipitation object with at least one grid point for at least three of the four 6-hourly time instances of the day. Furthermore, for PV structures, the multiple matches per day need to occur on the same isentrope, which has the intention to exclude 'random' matches of PV structures on different isentropes that are related to different weather systems, following the approach of De Vries et al. (2018). Matches of combined PV and IVT structures are allowed to occur at different time intervals of the day as long as both occur for at least three time instances of the day.

Results of the "inclusive" matching approach are evaluated for their statistical significance using a Monte Carlo test. Specifically, we test the null hypothesis that EPEs occur independently from PV and/or IVT structures, and hence the

observed matching fractions solely result from the climatological frequencies of EPEs and these structures. Statistical

significance is then assigned to grid points where this null hypothesis is rejected, using a two-step approach detailed hereafter. Note that rejections of this null hypothesis indicate that the matches occur more/less frequently than one would expect under the assumption of independence, and hence, we can infer from this test where significantly positive or negative associations arise between the two synoptic-scale processes and the occurrence of EPEs.

We first construct the null distribution of our test statistic (the matching fraction) by matching of the objects for 1000 Monte Carlo samples whereby the dates of EPEs are retained as in reality, and dates of PV and IVT structures are shuffled through selecting a random day within the same month of another year. This selection procedure removes potential sensitivities of the test to a strong seasonal influence, and provides at the same time a large sample of random days. The p-value for each grid point is then derived from comparing the observed matching fraction with the null distribution at the respective grid

point. Second, we apply the false discovery rate (FDR) test (Benjamini and Hochberg, 1995) with a maximum FDR of 0.1 to the resulting field of p-values to control the fraction of falsely rejected null hypotheses (Wilks, 2016).

To quantify spatial variation in the strength of the association between the EPEs and the PV and/or IVT structures we compare the observed odds of a match at any grid point with the respective odds under the above null hypothesis, by

365 computing odds ratios using the results from the Monte Carlo test, i.e.,

$$odds\ ratios = \frac{P(EPE|S_{obs})[1 - P(EPE|S_{MC})]}{P(EPE|S_{MC})[1 - P(EPE|S_{obs})]}. \quad (6)$$

Here, $P(EPE|S_{obs})$ is the observed matching fraction, which can also be interpreted as the probability of finding a match of

370 an EPE with an observed PV and/or IVT structure. $P(EPE|S_{MC})$ is the averaged matching fraction of the 1000 Monte Carlo samples, which may also be interpreted as the probability of finding a match under our null hypothesis. Odds ratio values > 1 indicate a positive association of EPEs and the PV and/or IVT structures (more matches than under independence), and conversely for odds ratio values < 1. These odds ratios provide a measure of the strength of the association between EPEs and the two synoptic-scale processes, and they are comparable across space as they take into account the spatially varying

occurrence frequencies of these synoptic-scale processes (see section 4.1). Odds ratios have also been used in previous studies to estimate the likelihood of precipitation extremes to occur in relation to several synoptic-scale weather systems (Röthlisberger et al., 2016; Lenggenhager and Martius, 2019; Moore et al., 2019).

## 3    Example cases of extreme precipitation and flooding

We demonstrate the importance of Rossby wave breaking and intense moisture transport for EPEs using a selection of
twelve notorious flood events that have been the subject of previous case studies (Table 2). Most of these events led to
immense catastrophic societal and economic impacts. Records of EM-DAT, also provided in Table 2, estimate that several of
these events resulted in a dozen up to thousands of fatalities and an estimated damage in the order of billions of US dollars.
These infamous flooding events include the great Colorado floods in September 2013 (Gochis et al., 2015; Eden et al.,
2016), the Jeddah flooding of November 2009 (Haggag and El-Badry, 2013; De Vries et al., 2016), the July 2010 Pakistan
floods (Hong et al., 2011; Martius et al., 2013), the Atacama Desert floods in March 2015 (Wilcox et al., 2015; Bozkurt et
al., 2016; Rondanelli et al., 2019), and the September 1987 Natal floods in South Africa (Triegaardt et al., 1988; Heerden
and Taljaard, 1998). It should be noted here that the dates of the EPEs, as listed in Table 2, are as mentioned in the cited case
studies, while the periods of the flood events and attributed socioeconomic impacts from the EM-DAT records can cover
much longer periods. For example, the EPE in India occurred on 17-18 June 2013 (Joseph et al., 2015; Kumar et al., 2016;
Vellore et al., 2016), whereas the 6,054 fatalities attributed to the flood were reported for the period of 12-27 June across
several provinces of India in the EM-DAT database. Thus, it is not guaranteed that all listed socioeconomic impacts in Table
2 exclusively stem from the EPEs under discussion.

All twelve cases clearly demonstrate the presence of a stratospheric PV streamer and an IVT structure on its downstream
flank that overlaps the extreme precipitation objects (Figs. 4 and S1). Most of these events depict a southwest-northeastward
(northwest-southeastward for events in the Southern Hemisphere) orientation of the PV streamer, which is typical of
anticyclonic Rossby wave breaking that follows the LC1 baroclinic wave life cycle of Thorncroft et al. (1993). Likely, this
type of wave breaking is very prominent in our selection of events as most of them took place in subtropical regions, which
are predominantly affected by "equatorward" Rossby wave breaking that is characterized by a backward-tilting and thinning
upper-level trough that moves equatorward (Thorncroft et al., 1993). It is noteworthy that several of our selected events
affected subtropical hot arid desert regions in Arizona, the Western Sahara, Arabia, the Atacama, and Namibia (Knippertz
and Martin, 2005; Muller et al., 2008; De Vries et al., 2016; Wilcox et al., 2016; Yang et al., 2019), which is not a coincide
as will be shown in our climatological analysis (section 4).

Interestingly, these cases also depict two different constellations of the positioning of the PV streamers with respect to the
EPEs. For most cases, the PV structures directly overlap a part of the extreme precipitation objects, as for example for the
events over North America, northwest Africa, the Alpine region, the Atacama Desert, and southern Africa (Fig.
4a,b,c,d,h,j,k). For few other cases, the PV structures are positioned farther away and do not directly overlap the extreme
precipitation object, although they connect indirectly with the extreme precipitation objects through overlap with the IVT
structures' extended area (Fig. 4e,f,g). These two different constellations imply that Rossby wave breaking can initiate heavy

rainfall through (1) solely steering the moist air flow toward the region of extreme precipitation, as for the latter mentioned cases, and (2) in addition to moisture transport also providing some form of upper-level forcing for upward motion, as for the former mentioned cases. Sections 6.2 and 7 will elaborate on this concept and provide climatological evidence for these two different forcing mechanisms.

## 4      Climatology

### 4.1      Rossby wave breaking and intense moisture transport

We now shift our focus from case studies to a climatological analysis for the ERA-Interim period of 1979-2018. Before addressing the link between EPEs and the synoptic-scale processes of interest, we briefly examine the climatology of Rossby wave breaking and intense moisture transport as to provide a climatological context for these synoptic-scale processes. Figure 5 depicts occurrence frequencies of stratospheric PV streamers and IVT structures. Previous studies have shown climatological frequencies of stratospheric PV streamers on single isentropic surfaces based on ERA-15 and ERA-40 reanalyses (Wernli and Sprenger, 2007; Martius et al., 2007). Here, we show the global ERA-Interim climatology of stratospheric PV streamers across multiple isentropes to account for the strong seasonality and latitudinal variability of the isentropic surfaces on which these structures preferentially appear (Röthlisberger et al., 2018). Figure 5a displays vertically aggregated fields whereby PV streamers are only counted once if they occur on multiple isentropes at a particular time step and grid point. The climatology of IVT structure frequencies is shown in Fig. 5b and Fig. 5d for the $IVT_{filt}$ and $IVT_{pct}$ forms, respectively, and Fig. 5c shows the IVT values that correspond to the annual 95[th] percentile, used as threshold for identification of $IVT_{pct}$ structures.

Rossby wave breaking occurs predominantly in the extratropics, and reduces towards the tropics and poles. PV streamers reach highest frequencies over the West and East Coasts of North America, the Mediterranean and southern Europe, the southern part of South America, along the southern Australian coast extending in an eastward direction across New Zealand and a large part of the South Pacific, and to a lesser extent near the South African coast. Values reach over 32 % in most of these regions and exceed 40 % over the Mediterranean (Fig. 5a). Interestingly, PV streamers occur most frequently in coastal regions near high topography, including the Rocky Mountains, the Andes, the Mediterranean Basin, the southern African and Australian coasts, and New Zealand.

Intense moisture transport, expressed in the $IVT_{filt}$ form, occurs predominantly over extratropical and tropical oceans, where storm tracks and tropical easterlies dominate (Fig. 5b). Furthermore, the climatology reflects the presence of well-known low-level jets, including those over the Great Plains in the USA, South America along the Andes, and the Caribbean (Gimeno et al., 2016). $IVT_{filt}$ structures are rarest over the polar regions and high topography where IVT is typically very

low, and over the subtropical eastern Pacific and Atlantic Oceans where semi-permanent subtropical anticyclones dominate. Switching focus to the $IVT_{pct}$ form, the annual 95$^{th}$ percentile of IVT values range from < 50 kg m$^{-1}$ s$^{-1}$ in polar regions up to > 850 kg m$^{-1}$ s$^{-1}$ over the western North Pacific (Fig. 5c). Frequencies of $IVT_{pct}$ structures are by design near 5 % as these

445 structures are defined by the annual 95$^{th}$ percentiles of IVT magnitudes (Fig. 5d). Note that these fractions can be lower than 5% as small IVT structures are removed from the selection, and the 95$^{th}$ percentile can fall below the lower limit of 200 kg m$^{-1}$ s$^{-1}$. Especially due to the latter reason, $IVT_{pct}$ structure frequencies remain very low over polar regions and elevated topography, i.e., the Rocky Mountains, the Andes, the Himalaya's, the Ethiopian Highlands, Greenland and Antarctica (Fig. 5c,d).

**4.2     EPEs linked to Rossby wave breaking and intense moisture transport**

In this section, we connect extreme precipitation objects with (1) PV structures, (2) IVT structures, (3) combined PV and IVT structures with a direct influence of the PV structures, and (4) combined PV and IVT structures, whereby also a remote influence of the PV structures is allowed. More specifically, the direct PV influence of category (3) requires the PV structure to have a direct overlap with the extreme precipitation object, whereas for category (4) the PV structure is also allowed to

455 connect with the extreme precipitation object through overlap with the IVT structures' extended area, provided that the IVT structure itself overlaps the extreme precipitation object. Examples of categories (3) and (4) were given by the two types of cases discussed in section 3, and are also exemplified in Fig. 3 by an EPE over the Middle East and the US West Coast, respectively. Figure 6 presents the fractions of EPEs that match with the PV and/or IVT structures for the four described categories. Hatching and stippling in Fig. 6a indicates where these values are statistically significantly larger and smaller,

respectively, than under assumed independence of the EPEs and the PV structures (section 2.5). Fractions in Fig. 6b-d are only plotted if statistically significantly larger than under assumed independence as grid points with statistically significantly fractions smaller than under assumed independence are virtually absent. In the following discussion, we focus primarily on IVT structures in the $IVT_{pct}$ form as this form shows a particularly high importance for EPEs over land, in contrast to the $IVT_{filt}$ form that has a relatively high relevance for EPEs over the large ocean basins (Fig. S2).

Rossby wave breaking is frequently associated with EPEs over subtropical and extratropical regions (Fig. 6a). Frequencies reduce substantially toward the poles and are negligible over the tropics. More specifically, regions where EPEs are strongly and significantly (crossed hatching in Fig. 6a) linked to PV structures include southwestern North America and the adjacent eastern Pacific upstream of the Rocky Mountains at subtropical latitudes, central North America downstream of the Rocky

Mountains at extratropical latitudes, the entire Mediterranean Basin and surrounding regions extending over the eastern Atlantic, North Africa, and a large part of the Middle East, and Northeast Asia with portions north of the Himalaya's and over the Sea of Okhotsk. In the Southern Hemisphere, we find the subtropical eastern Pacific and the western fringes of South America upstream of the Andes, the extratropical part of South America downstream of the Andes, the southern

African coast and adjacent seas, the southern margins of the Australian continent, and the Southern Ocean around the Antarctic continent. Fractions of EPEs that co-occur with PV structures exceed 80 % in all these regions, and reach over 90 % in parts of the Mediterranean and central North America. Interestingly, EPEs linked to PV structures also show statistically significant fractions larger than under assumed independence over the core polar regions where their values remain relatively low. In contrast, high values of EPE fractions over much of the extratropics remain statistically insignificant or even show statistically significant fractions lower than under assumed independence (dots in Fig. 6a), revealing a negative association with PV structures. This suggests that very frequent Rossby wave breaking in midlatitude storm track regions is not a decisive factor for EPE formation in these regions, whereas infrequent Rossby wave breaking poleward and at several locations equatorward of the midlatitude storm tracks contributes significantly to EPEs. Specific regions that were addressed in previous case studies and regional climatological analyses also clearly show up as important regions where Rossby wave breaking is relevant for EPEs; southwestern and central North America, the southern Alps, the Middle East, the western Himalaya's and Pakistan, northern Chile, and southern Africa (Massacand et al., 1998; Martius et al., 2006; Knippertz and Martin, 2007; Hart et al., 2010; Martius et al., 2013; Bozkurt et al., 2016; De Vries et al., 2016, 2018; Vellore et al., 2016; Moore et al., 2019). These regions are characterized by high topographic plateaus and mountain ranges, in line with a previously proposed mechanism through which a PV streamer forces and orients the low-tropospheric moist air flow towards a topographic barrier, supporting the formation of heavy rainfall (Schlemmer et al., 2010).

Intense moisture transport is important for EPEs throughout the globe and in particular over many coastal regions where vast oceans border continental land (Fig. 6b). These regions include the entire West Coast of North America, eastern North America stretching from the Caribbean up to the southwestern margins of Greenland, northwest Africa and parts of Western Europe, large parts of the East Asian coast that border the Pacific, and the southwestern and southeastern coastal zones of South America (Fig. 6b). Many of these coastal regions have also been identified as hotspots where atmospheric rivers bring heavy rainfall and flooding (Ralph et al., 2006; Lavers et al., 2013a; Guan and Waliser, 2015; Waliser and Guan, 2017). Other regions include almost entire Australia, large parts of the Middle East and the Indian Peninsula, and very large parts of the subtropical Indian and Pacific Oceans (Fig. 6b). The latter region very likely reflects signatures of tropical cyclones that occur relatively frequent over these warm subtropical oceans (Khouakhi et. al., 2017; Franco-Diaz et al., 2019). Fractions of EPEs related to intense moisture transport reach over 90 % in most of these aforementioned regions and locally exceed 95%. $IVT_{pct}$ structures have very limited relevance to EPEs across the tropics, where heavy rainfall typically occurs under the influence of local forcing, and over high topography where IVT values rarely exceed the lower limit of 200 kg m$^{-1}$ s$^{-1}$ to identify $IVT_{pct}$ structures. Interestingly, EPE matches with $IVT_{pct}$ structures are also remarkably low on the lee side of the Rocky Mountains and the Andes, the eastern part of the Iberian Peninsula, and the entire Mediterranean Basin. These are precisely the regions where EPEs are very strongly linked to PV structures, suggesting that the upper-level forcing alone

suffices to initiate their formation and does not require intense moisture transport. We will further discuss this in sections 6.2 and 7.

Hotspots where combined Rossby wave breaking and intense moisture transport are relevant for EPEs emerge predominantly
over subtropical regions and to a lesser degree over extratropical regions (Fig. 6c,d). These regions include southwestern North America and the adjacent eastern North Pacific, northeastern North America with centres over the Hudson Bay and Labrador Sea, northwest Africa and the neighbouring southwestern Iberian Peninsula and eastern North Atlantic, the Middle East, the northeastern part of the Asian continent including the Sea of Okhotsk, the western margins of South America and adjacent South Pacific, southern Africa and adjacent seas, the southern margins of Australia and parts of the neighbouring
ocean, and the coastal waters of the Southern Ocean around the Antarctic continent. Fractions of EPEs that are linked to combined PV and $IVT_{pct}$ structures with a direct PV influence (category 3) attain 60 % over all these regions, reach locally up to 70 % for most of these regions, and exceed 80 % near California (Fig. 6c). Turning to the combined occurrence of PV and $IVT_{pct}$ structures, whereby also a remote PV influence is allowed (category 4), EPE fractions demonstrate a very similar spatial distribution with about 10 to 20 % higher fractions that exceed 70 % in all aforementioned region, reach up to 80 % in
most of these regions, and exceed 90 % over California (Fig. 6d). Most of the aforementioned subtropical (extratropical) regions in the Northern Hemisphere are positioned near the equatorward end (poleward start) of the Pacific and Atlantic storm tracks where anticyclonic (cyclonic) wave breaking has been suggested to favour precipitation extremes (Röthlisberger et al., 2016). Taking both hemispheres into consideration, subtropical hotspots emerge near west coasts of continental land. These regions are typically characterized by hot (semi)arid desert climates where tropical-extratropical interactions
contribute to a large part of annual rainfall and heavy rainfall events (Favors and Abatzoglou, 2013; Pascale and Bordoni, 2016; Knippertz, 2003; Hart et al., 2013; De Vries et al., 2018; Sierks et al., 2020).

Repeating the same analysis with $IVT_{filt}$ instead of $IVT_{pct}$ shows a very similar spatial distribution and magnitude of fractions for combined PV and IVT structures with a direct PV influence (category 3; Fig. S2c). However, EPE matches with $IVT_{filt}$
structures (category 2; Fig. S2b) and with combined PV and IVT structures that allow a remote PV influence (category 4; Fig. S2d) show much higher fractions over oceans. For this reason, bearing in mind the destructive impacts of EPEs and flooding over land, we focus in the remaining analysis solely on the $IVT_{pct}$ form, hereafter simply referred to as IVT structures.

**4.3     Odds ratios of EPEs linked to Rossby wave breaking and intense moisture transport**

The previous section presented fractions of EPEs that can be attributed to Rossby wave breaking, intense moisture transport, and their combined occurrence. To provide a measure of the importance of the individual and combined synoptic-scale

processes for the formation of EPEs, we consider the odds of observing a match of an EPE with a PV and/or IVT structure
relative to the "matching-odds" expected under the assumption of independence between EPEs and these PV and/or IVT structures (see section 2.5). Figure 7 presents the resulting odds ratios of EPEs that match with these structures for the four described categories where their linkage is statistically significant.

Odds ratios of EPEs linked to the two individual and combined synoptic-scale processes highlight identical regions where
EPE fractions attributed to these categories are highest. Therefore, we refrain from a detailed analysis and focus on the overall emerging patterns. Rossby wave breaking is positively associated (odds ratios > 1) with EPEs over several subtropical regions and the poles, equatorward and poleward of the midlatitude storm tracks, respectively (Fig. 7a). In contrast, EPEs are negatively associated (odds ratios < 1) with Rossby wave breaking over the midlatitude storm track regions and few low-latitude regions where (extra)tropical cyclones dominate the EPE formation. Intense moisture transport
is positively associated with EPEs across the globe, especially in many coastal regions where atmospheric rivers and tropical cyclones frequently lead to EPEs. Combined Rossby wave breaking and intense moisture transport is highly associated with EPEs over a large part of the globe, in particular in the aforementioned (semi)arid subtropical regions where tropical-extratropical interactions govern (extreme) precipitation generation as well as over Greenland and Antarctica (Fig. 7c). Odds ratios of combined Rossby wave breaking and intense moisture transport that also allows a remote PV influence show a very
similar spatial distribution, with higher odds ratio values over several extratropical west coasts where atmospheric rivers frequently make landfall (Fig. 7d).

A general finding that emerges from Figure 7 is that odds ratios of EPEs linked to IVT structures are much higher, about a factor 10, than those linked to PV streamers (Figs. 7a,b). A plausible interpretation of this result is that intense moisture
transport is more tightly linked to EPEs, and hence conceivably also a better predictor for EPEs, than Rossby wave breaking. This notion has to be interpreted with caution as PV structures are typically much larger than IVT structures (e.g., see Fig. 4) and basic synoptic reasoning suggests that not in all parts of PV structures increased odds of EPEs are to be expected. Both of these aspects may be partly responsible for higher matching fractions in the Monte Carlo samples, and thus lower odds ratios, for PV structures compared to IVT structures. Thus, the higher importance of IVT structures for EPEs than PV
structures may in part be a consequence of the way we define these structures. An alternative PV streamer definition, for example, identifying only their downstream flank, might yield different results.

## 5 Extreme precipitation severity related to PV and IVT characteristics

Next, we explore the relationship between the extreme precipitation severity and the characteristics of Rossby wave breaking and intense moisture transport. To this end, we consider the number of isentropic surfaces with PV structures that coincide

with the extreme precipitation objects as a measure of the depth and strength of the wave breaking. Similarly, for the IVT structures, their maximum IVT values are used as an indicator of the moisture transport intensity. This part of the analysis focuses solely on EPEs that are linked to combined Rossby wave breaking and intense moisture transport with a direct PV influence (category 3 of the previous section), and is restricted to EPEs for six different regions in the Northern Hemisphere; the subtropics (20º-40º N), extratropics (40º-60º N), and polar regions (60º-80º N), over land and over water separately.

Figure 8 shows the statistical distributions of the depth of the PV structures and the averages of the moisture transport intensities for six clusters of EPEs, defined and ranked by their precipitation volumes. Across all regions, a clear relationship becomes immediately apparent. For larger precipitation volumes, we find larger IVT magnitudes, and higher numbers of isentropes with PV structures. The only exception appears over subtropical waters where the second and fourth precipitation

cluster shows a slightly lower number of isentropes (the average and 90th percentile) than the preceding precipitation clusters. For all other precipitation clusters and regions, the average depth of the PV structures steadily increases towards larger precipitation volumes. Note that the average precipitation volumes for the six clusters in all regions increase by construction due to the definition of the precipitation clusters.

Figure 8 reveals a few other interesting facts. The average IVT maximum is consistently larger over water than over land, and larger over the extratropics than over subtropical and polar regions. This finding is not too surprising as IVT is proportional to the product of specific humidity and wind, and considering that wind speeds are generally stronger at higher than at lower latitudes, and that tropospheric moisture amounts are generally higher over ocean than over land, and decrease towards polar regions. Interestingly, the two precipitation clusters with the largest precipitation volumes, found over

subtropical and extratropical waters (Fig. 8d,e), go along with the highest IVT maxima (> 1100 kg m$^{-1}$ s$^{-1}$) and the deepest PV structures (2.4 isentropes on average, and five isentropes for the 90th percentile), underlining the profound influence of the wave breaking strength and the moisture transport intensity on the extreme precipitation severity. Furthermore, we observe that the depth of the PV structures is marginally, but consistently, larger for EPEs over water than over land. This may suggest that the upper-level forcing plays a more vital role for EPEs over homogeneous oceans, where the forcing for

ascent must primarily come from upper levels, than for EPEs over land, where orographic lifting and strong surface heating can contribute to upward motion, which is consistent with findings of section 6.

In Fig. 9, we look at the relationship under discussion from a reversed perspective. We divide the 6-hourly time instances with extreme precipitation objects that match combined PV and IVT structures in different clusters, based on the IVT

maximum and the depth of the PV structures, and compute the corresponding average precipitation volumes for these clusters. One may expect increasing frequencies with increasing IVT maxima and an increasing depth of the PV structures. However, the frequency distribution of PV and IVT characteristics do not show such a pattern and adhere to a skewed

normal distribution. Interestingly; however, in all regions, the average precipitation volume increases with increasing depth of the PV structures and increasing IVT maxima. Thus, also when taking the PV and IVT characteristics as a starting point, the same conclusion emerges; the extreme precipitation severity is strongly influenced by the strength of the wave breaking and the intensity of the moisture transport.

Another interesting finding from Fig. 9 is that the average precipitation volume does not substantially increase with an increased depth of the PV structures for the lowest IVT maxima clusters ($< 400$ and $400$-$600$ kg m$^{-1}$ s$^{-1}$). This holds in all regions, and suggests that even when the upper-level forcing is very strong, the IVT intensity acts as a constraint on the precipitation volume and is thus crucial for regulating the extreme precipitation severity. This finding is in line with previous work that linked the IVT magnitude to precipitation extremes (Lavers et al., 2014; Froidevaux and Martius, 2016).

## 6    Tropospheric environments and mechanisms of EPEs

### 6.1    Tropospheric circulation in different regions

In the last part of this study, we return to the starting point of our endeavour, and examine the tropospheric environments and processes that contribute to the formation of the EPEs. To this end, we construct composites of daily means and normalized anomalies with a radius of 2000 km, centred on the mass-weighted centres of the EPEs. Normalized anomalies are computed using daily means and standard deviations based on a 21-day running time window, centred on the day of the EPEs, following De Vries et al. (2016; 2018). Thus, daily means reflect the "average" tropospheric environment in which the EPEs develop, whereas the normalized anomalies provide a measure for the deviation of this tropospheric state from normal conditions for the time of the year and the specific locations of the events.

Figure 10 presents these composites for EPEs that are linked to combined Rossby wave breaking and intense moisture transport in the subtropical, extratropical and polar regions of the Northern Hemisphere, over land and water separately. EPEs in all regions clearly demonstrate the formation of an upper-level trough and cyclogenetic processes near the surface, as indicated by the 500-hPa geopotential height and mean sea level pressure contours (Fig. 10a-f). The position of the corresponding normalized anomaly centres demonstrates a backward tilt with height (Fig. 10g-l), which is suggestive of baroclinic wave growth (Hoskins et al., 1985) and consistent with other global studies on EPEs (Dai and Nie, 2020). This circulation drives intense poleward and eastward moisture transport on the downstream flank of the upper-level trough (Fig. 10a-l), leading to enhanced tropospheric moisture amounts in the region of extreme precipitation, exceeding 1.5 to 2.0 standard deviation (SD) (Fig. 10g-l). Daily precipitation amounts and their normalized anomalies exhibit diagonally oriented patterns, reflecting a poleward and eastward movement of the precipitating systems (cf. Skok et al., 2009), most notable in the subtropics and the extratropics (Fig. 10a,b,d,e,g,h,j,k).

When comparing the tropospheric circulation and their deviation from climatology across the latitudinally varying regions, some interesting differences attract our attention. In the subtropics, pronounced 500-hPa normalized anomalies reach below –1.5 SD (Fig. 10g,j), which reflect the strong signatures of upper-level troughs that reach far into low latitudes, as opposed to much weaker anomalies over the extratropical and polar regions of –0.5 SD (Fig. 10h,i,k,l). In addition, mean sea level pressure contours for subtropical EPEs suggest that inverted surface troughs extend from the equatorial low-pressure zone in a poleward direction under the influence of the intruding upper-level troughs (Fig. 10a,d). These characteristics are very typical of tropical-extratropical interactions that have previously been associated with precipitation and extreme precipitation in several subtropical regions (Knippertz and Martin, 2005; Hart et al., 2010; Favors and Abatzoglou, 2013; Pascale and Bordoni, 2016; Vellore et al., 2016; De Vries et al., 2018). In contrast, the extratropical and polar regions demonstrate a more pronounced backward tilting cyclonic structure with height (Fig. 10h,i,k,l) and multiple closed mean sea level pressure contours (Fig. 10b,c,e,f), reflecting the signatures of developing surface cyclones that frequently lead to precipitation extremes in these regions (Pfahl and Wernli, 2012; Papritz et al., 2014).

## 6.2    Extreme precipitation mechanisms for different synoptic categories

Finally, we attempt to disentangle the different tropospheric processes and forcing mechanisms of upward motion for EPEs that are linked to different constellations of the two synoptic-scale processes under discussion, using the "exclusive" categorization (Table 1). More specifically, composites are constructed for EPEs that are linked to (1) *only* PV structures, (2) *only* IVT structures, (3) combined PV and IVT structures with a *direct* PV influence, (4) combined PV and IVT structures, *only* with a *remote* PV influence, and (5) *neither* PV structures *nor* IVT structures. These categories are similar to those of sections 4.2 and 4.3; however, the assignment of EPEs to the categories is now exclusive, meaning that a specific EPE only belongs to one category and that matching with multiple categories is ruled out. Composites are complemented with 500-hPa QG ascent and low-tropospheric static stability, and only include EPEs for the period of 1979-2016 due to the availability of QG vertical motion for only these years. The composites are discussed for the Northern Hemispherical subtropical regions over land and water, where the largest differences manifest between the five categories (Figs. 11 and 12). The findings presented in this section also hold to a large extent in the extratropics (Figs. S3 and S4), while the differences between the categories largely disappear in the polar regions (not shown). Regions in the Southern Hemisphere show generally similar patterns, although the unequal distribution of land and water leave a noticeable imprint on these patterns. For this reason, we only include the composites of EPEs over water in the subtropical and extratropical Southern Hemisphere in the supplement (Figs. S5 and S6).

Composites of subtropical EPEs that are linked to combined Rossby wave breaking and intense moisture transport, with a direct PV influence (category 3), show strong 500-hPa QG ascent near the centre of the EPEs, both over land and water

(Figs. 11c,h, and 12c,h). Daily mean QG ascent exceeds 0.1 (0.075) Pa s$^{-1}$ and their normalized anomalies surpass 1.5 (2.0) SD over land (water). In addition, the low-tropospheric static stability is reduced by about 0.25 SD over a large part of the extreme precipitation region over water, reaching 0.5 SD along its margins (Fig. 12h), whereas no substantial reductions are observed over land (Fig. 11h). This suggests that ascent induced by the balanced flow has an important influence on the formation of EPEs over both land and ocean, whereas the relevance of reduced static stability is weaker and confined to EPEs over oceans.

EPEs that are linked to combined PV and IVT structures, exclusively with a remote PV influence (category 4), show remarkable differences compared to those with a direct PV influence (category 3). The upper-level forcing appears in the form of a wider, open-trough like structure that covers a larger spatial extent and is positioned farther away from the extreme precipitation centre in a poleward-westward direction (Figs. 11d,i and 12d,i). Importantly, the centre of QG ascent is located farther away from the extreme precipitation centre and has a lower magnitude that reaches about 0.05 Pa s$^{-1}$ and 1 SD over both land and water (Figs. 11d,i and 12d,i). Also, the region of reduced static stability for EPEs over water is positioned far away from its centre, and lacks any spatial overlap with the extreme precipitation imprint (Fig. 12i). These findings suggests that EPEs of this category arise from the upper-level forcing that steers more zonally-oriented moisture transport towards the region of extreme precipitation, consistent with IVT daily means and normalized anomaly vectors (Figs. 11d,i and 12d,i), whereby the direct influence of the upper-level forcing on ascent has only a reduced or negligible influence. Presumably, other forcing mechanism for ascent play a dominant role here, such as (1) orographic lifting of the moist air masses for EPEs over land, (2) the potential effects of condensational heating on the vertical stability, and (3) other mesoscale processes that shape the organization of convective storms.

EPEs that are exclusively linked to Rossby wave breaking (category 1) exhibit similar characteristics as those linked to combined PV and IVT structures with a direct PV influence (category 3), except for few remarkable differences. By definition, daily mean and anomalous IVT remains very low (Figs. 11a,f and 12a,f). Interestingly, the 500-hPa geopotential height normalized anomalies reflect the signatures of upper-level troughs that reach far equatorward and are positioned very close to the centre of the EPEs (Figs. 11f and 12f). The direct influence of the upper-level forcing is manifested by strong QG ascent that exceeds 0.075 Pa s$^{-1}$ and 1.5 SD over both land and water, and reduced static stability of about 0.25 SD for a marginal region over land (Fig. 11f) and of more than 0.5 SD over water across almost the entire extreme precipitation imprint (Fig. 12f). Destabilization of the lower troposphere by the upper-level PV anomaly is of particular importance for EPEs of this category over water. As previously suggested, these events are likely found over warm subtropical waters, such as the Mediterranean Sea. Presumably, stratospheric PV streamers reach into a moist tropospheric environment, where the upper-level forcing suffices to trigger deep moist convection, without requiring intense moisture transport.

EPEs that are exclusively linked to intense moisture transport (category 2) display a very similar tropospheric circulation (Figs. 11b and 12b) as those linked to combined PV and IVT structures with exclusively a remote PV influence (category 4). Dynamical lifting is substantially weaker, as indicated by normalized anomalies in QG ascent of about 0.5 SD over both land and water (Figs. 11g and 12g). Finally, EPEs that occur in absence of both PV and IVT structures (category 5) exhibit very weak (over water) or virtually absent (over land) deviations in 500-hPa geopotential height and mean sea level pressure (Figs. 11j and 12j). This implies that the large-scale forcing is weak and that the EPEs occur in a moist tropospheric environment (TCW normalized anomalies attain +1.5 SD over both water and land) under the influence of local forcing mechanisms, which include destabilization of the troposphere over water (Fig. 12j), and presumably other processes such as orographic lifting and surface heating over land.

## 7    Towards a classification of EPE-related weather systems

Naturally, the question arises as to where these five categories of synoptic-scale processes are important for EPEs, and whether they have an importance influence on the extreme precipitation severity. Figure 13 presents the spatial distribution of EPEs that are linked to the five exclusive categories, their share in the total number of EPEs, and their partition in different precipitation volume clusters. These categories demonstrate a distinct geographical distribution (see also Fig. S7), consistent with previous results and the process-based understanding derived from the composite analysis in the previous section. In the following, we summarize the main characteristics of these five categories, and place these in context of well-known weather phenomena and existing literature. EPEs that are linked to:

1) **only Rossby wave breaking** are confined to distinct regions over central North America, southeastern South America, the Mediterranean and Caspian Seas, including the eastern Iberian Peninsula, and several local areas, such as the eastern portions of Iceland, Ireland, the United Kingdom, and furthermore, the Balkans, parts of Germany and Poland, the Caucasus, north of the Himalaya's, the western margins of northern Chili, and parts of Antarctica. Interestingly, most of these regions are located poleward and eastward of high topographic barriers and vast deserts (e.g., the Sahara, Rocky Mountains, Andes, Alps, Taurus and Zagros Mountains, and the Himalaya's), suggesting that these regions are deprived from remote moisture supplies. EPEs arise when Rossby wave breaking provides sufficient upper-level forcing through reduction of low-tropospheric stratification and dynamical lifting to initiate the extreme precipitation, as shown by the composites. Likely, these events form when PV streamers reach in a relatively moist tropospheric environment and/or when strong evaporation from warm sea surfaces, such as the Mediterranean Sea, provides sufficient moisture at the local scale (Winschall et al., 2014; Raveh-Rubin and Wernli, 2016);

2) **only intense moisture transport** reach around the globe at higher tropical and lower subtropical latitudes, away from the inner tropics, but beyond the reach of the extratropical upper-level forcing. EPEs of this category are most

likely associated with (i) tropical cyclones over the western and eastern North Pacific near East Asia and Central America, respectively, the southern Indian Ocean near Madagaskar and the Maritime Continent near north Australia, and the Gulf of Mexico and the southeastern USA (Prat and Nelson, 2016; Khouakhi et. al., 2017; Franco-Diaz et al., 2019), (ii) tropical easterly waves over North Africa, the North Atlantic, the Carribean, and Central America (Ladwig and Stensrud, 2009; Cretat et al., 2015; Vigaud and Robertson, 2017), and (iii) monsoon lows/depression over India and neighbouring regions in South Asia, the eastern North Pacific near Central America, and north Australia (Hurley and Boos, 2015; Berry and Reeder, 2016);

3) **combined Rossby wave breaking and intense moisture transport** with a direct PV influence dominate over the larger part of the globe, from subtropical to polar latitudes. This category is particularly relevant in (semi)arid subtropical regions where tropical-extratropical interactions are of central importance for (heavy) rainfall events (Wright 1997; Knippertz, 2003; Favors and Abatzoglou, 2013; Hart et al., 2013; Pascale and Bordoni, 2016; De Vries et al., 2018; Sierks et al., 2020). These events arise when the midlatitude forcing reaches into low latitudes and couples with the tropical low-level circulation, initiating a poleward moisture incursion into these dry regions;

4) **combined Rossby wave breaking and intense moisture transport, exclusively with a remote PV influence** emerge over large parts of the extratropical west coasts of North and South America, and also over western margins of Iceland, Norway, and New Zealand. Remote Rossby wave breaking steers intense moisture transport towards these mountainous coastal zones where orographic lifting provides forcing for upward motion, whereas the direct influence of the upper-level forcing for ascent through dynamical lifting or reduced static stability is weak or negligible. This mechanism is very typical of Rossby wave breaking that goes along with the formation and landfall of atmospheric rivers (Payne and Magnusdottir, 2014, 2016; Mundhenk et al., 2016b; Hu et al., 2017; Zavadoff and Kirtman, 2020). In addition, events of this category also emerge over subtropical regions in transition zones between categories 2 and 3 over Australia and East Asia. Presumably, the PV streamers reach into low latitudes and initiate poleward transport of tropical moisture, but are positioned too far away to exert a direct forcing for ascent, consistent with the Jeddah, Indian and Pakistan flooding events as discussed in section 3;

5) **neither Rossby wave breaking nor intense moisture transport** are predominantly found over the inner tropics, high topography at lower latitudes (the Himalaya's, the Ethiopian Highlands, the Mexican Plateau, and the Altiplano in the Andes), and the core polar regions. The limited relevance of the two synoptic-scale processes over these regions can be explained by the strong local forcing that gives rise to EPEs over the tropics (e.g., Maranan et al., 2018) and high topography. In addition, air at high altitudes and latitudes can only hold very low moisture amounts, resulting in relatively low IVT that rarely exceeds 200 kg $m^{-1}$ $s^{-1}$. Also, over the poles, the detection of PV streamers can be questionable as the stratospheric bodies are centred over these regions.

This analysis suggests that different combinations of Rossby wave breaking and intense moisture transport can represent various EPE-related weather systems. Interestingly, regions where none of the five exclusive categories dominate (the white regions in Fig. 13) coincide to a large extent with those where cyclones and warm conveyor belts have the highest relevance for precipitation extremes (Pfahl and Wernli, 2012; Pfahl et al., 2014). Although these well-known weather systems typically go along with high IVT values, these do not necessarily attain the unusually high IVT values (>95$^{th}$ percentile) used for our definition of intense moisture transport. Whereas cyclones and warm conveyor belts, focus of numerous studies, have a particular relevance for precipitation and extreme precipitation in wet higher latitude regions, combined Rossby wave breaking and intense moisture transport is very important for EPEs in regions that received relatively little scientific attention, such as the (semi)arid subtropics. Accordingly, we propose that different combinations of Rossby wave breaking and intense moisture transport, complemented with cyclones, can introduce a new classification of EPE weather regimes across climate zones.

Although EPEs associated with neither PV nor IVT structures comprise the larger part of worldwide total events (36.7 %; Fig. 13b), their share reduces drastically toward higher extreme precipitation volumes (Fig. 13c). Likewise, percentages of EPEs linked to only Rossby wave breaking reduce drastically from the lowest (28.2 %) to the highest (3.3 %) precipitation volume clusters. In contrast, the share of EPEs linked to all three synoptic categories that involve IVT structures increase dramatically toward higher precipitation volumes. For example, the synoptic categories of only intense moisture transport (category 2) and combined Rossby wave breaking and intense moisture transport with a direct PV influence (category 3) almost quadruple from the lowest to the highest precipitation volume clusters; from 9.1 % to 35.0 %, and from 8.0 % to 29.4 %, respectively (Fig. 13c). These findings underscore once more that IVT can act as a constraint on the extreme precipitation volume and is highly associated with the extreme precipitation severity.

## 8      Conclusions

This study presented for the first time qualitative and quantitative evidence on the role of Rossby wave breaking and intense moisture transport for the formation of EPEs at the global scale. The importance of these synoptic-scale processes was first demonstrated by a number of previously documented flood events with catastrophic socioeconomic impacts in different parts of the world, and then complemented by a comprehensive climatological analysis. To this end, we applied state-of-the-art objects-based identification methods for (i) daily EPEs, (ii) stratospheric PV streamers, and (iii) IVT structures on the 40-year ERA-Interim reanalysis (1979-2018). The climatological analysis addressed three major aspects (1) the geographical distribution of EPEs that are linked to Rossby wave breaking, intense moisture transport, and their combined occurrence, (2) the influence of the strength of the wave breaking (i.e., the upper-level forcing) and the intensity of the moisture transport on the extreme precipitation severity, and (3) the nature of the tropospheric environment and mechanisms of upward motion that

contribute to the formation of EPEs. Accordingly, the introduction stated three research questions that are addressed in the following three paragraphs.

Rossby wave breaking is frequently associated with EPEs (> 80 %) in subtropical and extratropical regions, in particular in the vicinity of high topography and over the Mediterranean. Intense moisture transport is strongly linked to EPEs (up to 95 %) across the globe, especially over coastal zones where continental land borders vast ocean, consistent with findings of atmospheric-river related studies (Ralph et al., 2006; Lavers et al., 2013a; Waliser and Guan, 2017). Combined Rossby wave

breaking and intense moisture transport contribute substantially to EPEs (> 60 %) in subtropical regions and to a lesser degree in extratropical regions. Subtropical regions that stand out include southwestern North America, northwest Africa, the Middle East, the western flanks of southern South America, and the southern coasts of Africa and Australia. According to previous studies, and in line with our composites, these subtropical (semi)arid regions typically receive precipitation and extreme precipitation from tropical-extratropical interactions (Knippertz, 2003; Hart et al., 2013; Favors and Abatzoglou,

2013; Pascale and Bordoni, 2016; De Vries et al., 2018). Hereby, the midlatitude forcing reaches into low latitudes in the form of upper-level troughs that initiate poleward excursions of tropical moisture into these dry subtropical regions (Knippertz, 2007). These meteorological processes are very well represented by the combination of stratospheric PV streamers and IVT structures that reflect such transient tropical-extratropical interactions at synoptic scales (De Vries et al., 2018). In addition, EPEs also frequently co-occur with surface cyclones (Pfahl and Wernli, 2012) and PV cutoffs (Favre et

al., 2013; Abatzoglou, 2016; Al-Nassar et al., 2019; Barbero et al., 2019). These components are not explicitly included in our analysis and may complement the findings of this study in future research.

One unique aspect of this study is the combined application of PV and IVT. PV has been of invaluable importance in dynamic meteorology during the last decades (Hoskins et al., 1985) and has previously been used to diagnose (i) the

815 influence of the upper-level forcing on precipitation generation (Funatsu and Waugh, 2008; Schlemmer et al., 2010; Martius et al., 2013) and (ii) Rossby wave breaking as a key driver of EPEs (Massacand et al., 1998; Martius et al., 2006; De Vries et al., 2018; Moore et al., 2019). IVT became rapidly popular during the last decade, not only as an essential diagnostic for atmospheric rivers, but also as a general characteristic of the tropospheric environment that supports the development of extreme precipitation (Moore at al., 2015; Froidevaux and Martius, 2016). We use both variables to identify two key larger-

820 scale (thermo)dynamic processes, and present conclusive evidence on their relationship with the extreme precipitation severity. Whether taking the extreme precipitation volume or the characteristics of the two synoptic-scale processes as a starting point, leads to the same conclusion: the deeper the PV streamers, and the higher the IVT maxima, the larger the extreme precipitation volumes. Thus, the strength of the wave breaking and intensity of the moisture transport are inherently linked to the extreme precipitation severity. Another interesting finding, based on odds ratios, suggest that intense moisture

transport is stronger associated with EPE occurrences, and thus potentially a more skilful predictor of EPEs, than Rossby wave breaking.

Based upon our detailed composites, and well-established principles in dynamic meteorology, we present a concept of the meteorological processes that lead to the formation of EPEs (Fig. 14). This concept involves (1) the upper-level forcing in
the form of a stratospheric PV streamer that steers intense moisture transport towards the region of extreme precipitation, (2) baroclinic wave growth whereby anomalies in the upper and lower tropospheric circulation interact and mutually amplify each other, and (3) various mechanism through which the upper-level forcing can favour ascent and deep moist convection, including (i) reduced static stability beneath the upper-level PV anomaly, (ii) dynamical lifting on its downstream flank, and (iii) interaction of the larger-scale flow with topography (slowing down of the circulation and lee cyclogenesis), which can
orient and anchor a low-tropospheric moist air flow towards the topographic barrier (not depicted in the schematic). This model builds upon longstanding concepts in the literature. Hoskins et al. (1985) explained baroclinic wave growth in a PV perspective by the mutual interaction and strengthening of PV anomalies in the upper and lower troposphere. Furthermore, they also envisaged the potential of PV to study intrusions of extratropical air masses into low latitudes in relation to tropical-extratropical interactions. More recently, several case studies proposed combinations of mechanism through which
an upper-level PV anomaly can initiate ascent, deep moist convection and precipitation generation (Funatsu and Waugh, 2008; Schlemmer et al., 2010; Martius et al., 2013). Our study presents climatological evidence on the contribution of these processes to EPEs, apart from orographic lifting, for different constellations of PV and/or IVT structures and for different regions in which these EPEs occur. QG ascent is particularly enhanced in regions of subtropical and extratropical EPEs that co-occur with PV structures over both land and water, whereas reduced static stability is predominantly confined to these
EPEs over water.

Going beyond the three central research questions of this study, we propose that different combinations of Rossby wave breaking and intense moisture transport may perfectly serve for a new definition and classification of weather regimes for EPEs across climate zones (Fig. 13). Over the inner tropics and high topography at low latitudes, EPEs arise predominantly
under the influence of local forcing, and both synoptic-scale processes are irrelevant. At the outer bands of the tropics and lower subtropics, IVT structures are highly associated with the formation of EPEs, reflecting well-known weather phenomena such as tropical cyclones, tropical easterly waves, and monsoon lows (Prat and Nelson, 2016; Cretat et al., 2015; Hurley and Boos, 2015). Slightly beyond these latitudes, the midlatitude upper-level forcing comes into play, and across the larger part of the globe, from subtropical to polar latitudes, EPEs are governed by combined Rossby wave breaking and
intense moisture transport, including dry desert regions that can be affected by tropical-extratropical interactions (see above). Several extratropical coastal regions, on western flanks of continental land, frequently experience EPEs when remote, far upstream Rossby wave breaking steers a moist air flow towards these mountainous coasts, conform previous studies that

linked the formation and landfall of atmospheric rivers to wave breaking (Payne and Magnusdottir, 2014, 2016; Mundhenk et al., 2016b; Hu et al., 2017; Zavadoff and Kirtman, 2020). Other specific extratropical regions are deprived from remote moisture transport by high mountain barriers or vast deserts, and EPEs occur when the upper-level forcing appears in the form of Rossby wave breaking, and reaches into a relatively moist tropospheric environment or when evaporation from local water bodies provides sufficient moisture. This new classification may have reduced relevance over the core polar regions, where the identification of PV streamers can be questionable. Seasonal aspects of this new classification are not addressed in this work and are planned as a part of a follow-up study.

Before concluding, we list three limitations of this study. First, the EPEs are defined on daily timescales and large spatial scales ($>50,000$ km$^2$), and thus, our findings may have reduced relevance for precipitation extremes at hourly and multi-day timescales as well as local flash floods over complex topography. Second, the EPEs are derived from reanalysis precipitation with inherent limitations, and although these may only weakly affect our EPEs due to their relatively large spatiotemporal scales, it would be interesting to extend the analysis of this study to other types of precipitation observational datasets. Third, the findings of this study may be sensitive to several choices in the methodology, such as the PV streamer geometry, the IVT form and threshold, the extreme precipitation definition, the minimum surface areas of the objects, and the criteria for the matching of the extreme precipitation objects with the PV and IVT structures.

In summary, this study shows that Rossby wave breaking and intense moisture transport are of central importance for the formation of EPEs. This relation holds globally except for the inner tropics and core polar regions. These findings may contribute to an improved understanding of the atmospheric processes that lead to EPEs, and may benefit medium-range weather prediction and early warning systems that can reduce flood risk and alleviate their socioeconomic impacts. Also, the presented methodology can be applied to climate model simulations for the past and projected future to provide a new perspective on how and why EPEs will change in a warming climate. Moreover, such an endeavour may provide new insights into the strengths and limitations of climate models in representing key synoptic-scale (thermo)dynamic processes of EPEs.

**Code and data availability**

ERA-Interim data from ECMWF is available from (https://www.ecmwf.int/), and the codes and data from this study can be provided by the author upon request.

**Competing interest**

The author declares no competing interests.

**Author contribution**

AV formed the ideas, prepared the data, designed the methodology, developed the computer algorithms, analysed the data, created the figures, and wrote the manuscript.

**Acknowledgements**

The author thanks Raphael Portmann (ETH Zürich), Michael Riemer (University of Mainz), and Stephan Pfahl (Freie Universität Berlin) for their comments on an early version of the manuscript. Deep gratitude goes to Heini Wernli (ETH
Zürich) for inspiring discussions and detailed feedback on this manuscript, and to Matthias Röthlisberger (ETH Zürich) for his invaluable help with the statistical significance test and the computation and interpretation of odds ratios based on the Monte Carlo simulations. Comments of Frederico Grazzini and an anonymous reviewer helped to improve the quality of the manuscript. Also, the author would like to thank Huug Ouwersloot for contributing to the algorithm for detection of EPEs, and Klaus Klingmüller (Max Planck Institute for Chemistry) for his help to run the algorithm for PV streamers on the
supercomputer Mistral from DKRZ. Furthermore, thanks go to Michael Sprenger (ETH Zürich) for providing global QG vertical motion, and to Jonathan Vigh (NCAR) for drawing the author's attention to the great Colorado floods in September 2013. The National Center for Atmospheric Research (NCAR) command language (NCL), version 6.3.0, has been used for the identification of PV streamers, and version 6.5.0, for identification of other object-based features, all computations, and the visualization of the results (http://dx.doi.org/10.5065/D6WD3XH5).

**Financial support**

The author acknowledges support from the PIRE funding scheme, via the Swiss National Science Foundation (SNSF) grant nr. 177996.

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

## Figure captions

**Figure 1.** Schematic representation of Rossby wave breaking (stratospheric PV streamer in blue) and intense moisture transport (IVT structure in green) as synoptic-scale processes of EPEs (extreme precipitation object in red), adapted from De Vries et al., (2018). The light blue and light green shadings illustrate the area of influence of the synoptic features, and the symbols W and L refer to the geometrical criteria of PV streamers, as further detailed in section 2.

**Figure 2.** Examples of the algorithm for identification of EPEs for (first column) Arizona, USA, 27 September 2014, (second column) northwest Africa, 10 January 2002, (third column) Jeddah, Saudi Arabia, 25 November 2009, and (fourth column) Pakistan, 28 July 2010. In (a)-(d) daily rainfall amounts, (e)-(h) grid points that exceed the annual >99[th] (green), 99.5[th] (yellow), and 99.9[th] (red) percentiles, (j)-(l) extreme precipitation objects with a surface > 50.000 km$^2$ and consisting of connected grid points within R < 250 km, and (m)-(p) extreme precipitation objects illustrated by random colours, with their mass-weighted centres denoted by the markers, and their volumes (m$^3$) stated in the text.

**Figure 3.** Example of the algorithm (a) input and (b) output for stratospheric PV streamers and IVT$_{pct}$ structures at 12 UTC 17 November 1996. In (a) $|PV|$ (PVU) on 335 K in yellow/red colours, and full IVT (kg m$^{-1}$ s$^{-1}$) in green/blue colours, and IVT vectors in black, only plotted where the full IVT magnitude > 250 kg m$^{-1}$ s$^{-1}$. Contour lines denote the +/–2 PVU contours (black), the 250 kg m$^{-1}$ s$^{-1}$ full IVT (blue), and the annual 95[th] IVT percentile with a lower threshold of 200 kg m$^{-1}$ s$^{-1}$ (red). In (b) the identified stratospheric PV streamers, IVT structures in the IVT$_{pct}$ form, and their areas of influence, as indicated by the colours in the legend. Also, (b) shows the 99[th], 99.5[th] and 99.9[th] percentile exceedances of daily ERA-Interim precipitation.

**Figure 4.** Illustrative cases of extreme precipitation and flooding accompanied by stratospheric PV streamers and IVT structures in (a) Arizona, USA, 00 UTC 27 September 2014, (b) Colorado, USA, 12 UTC 10 September 2013, (c) northwest Africa, 06 UTC 10 January 2002, (d) the Alpine region, 06 UTC 15 October 2000, (e) the Middle East, Jeddah, 12 UTC 25 November 2009, (f) Pakistan, 18 UTC 28 July 2010, (g) India, 00 UTC 17 June 2013, (h) the Atacama Desert in Chile, 12 UTC 25 March 2015, (i) Argentina and Uruguay, 18 UTC 28 March 2007, (j) Namibia, 00 UTC 20 April 2006, (k) South Africa, 00 UTC 27 September 1987, and (l) Australia, 00 UTC 13 January 2011. All meteorological variables and identified objects correspond to those as in Fig. 3b with the exception of extreme precipitation for which only the objects instead of all percentile-exceeding grid points are shown.

**Figure 5.** Climatological frequencies, based on 6-hourly occurrences for the period of 1979-2018, of (a) vertically aggregated stratospheric PV streamers between 300-350 K, (b) IVT structures in the IVT$_{filt}$ form, and (d) IVT structures in the IVT$_{pct}$ form. In (c) IVT values (kg m$^{-1}$ s$^{-1}$) that correspond to the annual 95[th] percentiles of full IVT magnitudes used as threshold for the identification of IVT$_{pct}$ structures.

**Figure 6.** Fractions (%) of EPE matches with (a) PV structures, (b) IVT$_{pct}$ structures, (c) combined PV and IVT$_{pct}$ structures with a direct PV influence, and (d) combined PV and IVT$_{pct}$ structures whereby also a remote PV influence is allowed, see the text for details. The climatologies are based on the ERA-Interim period of 1979-2018. All matches that involve PV structures are based on isentropic surfaces between 300-350 K with 5 K intervals. In (a) the crossed hatching and stippling indicates statistically significant fractions that are larger and smaller, respectively, than expected under independence of EPEs and PV structures (see the text for details), whereas in (b)-(d) only statistically significant fractions are plotted that are larger than expected under independence of EPEs and the structures.

**Figure 7.** Odds ratios of EPE matches with (a) PV structures, (b) IVT$_{pct}$ structures, (c) combined PV and IVT$_{pct}$ structures with a direct PV influence, and (d) combined PV and IVT$_{pct}$ structures whereby also a remote PV influence is allowed, i.e., observed matching odds divided by the matching odds expected under independence of EPEs and the PV and/or IVT structures. Odds ratios are plotted transparent where the link between EPEs and these structures is not statistically significant.

**Figure 8.** Characteristics of PV and IVT structures for EPEs, linked to combined wave breaking and intense moisture transport, divided and ranked in 6 clusters based on their precipitation volumes for the Northern Hemispheric (left) subtropics (20°-40° N), (middle) extratropics (40°-60° N), and (right) polar regions (60°-80° N), over (top) land and (bottom) water. The number of isentropes with PV structures is shown by the 10[th] to 90[th] percentile range and the average indicated by the dot, and the average of the IVT maxima (kg m$^{-1}$ s$^{-1}$) is shown by the colours of the bars. The height of the bars corresponds to the average precipitation volumes (m$^3$) of each EPE cluster. The number of EPEs for each region is included in the figure titles, and the numbers above the colour bars indicate the fractions (%) of 6-hourly time instances with combined PV and IVT structures during the daily EPEs.

**Figure 9.** Average precipitation volumes of the EPEs in the circles, as denoted by the legend, for clusters defined by the number of isentropes with PV structures and the IVT maxima (kg m$^{-1}$ s$^{-1}$). As in Fig. 7, the plot is based on EPEs, linked to combined wave breaking

and intense moisture transport, for identical regions. The colours of the squares reflect the frequencies (%) of 6-hourly time instances with PV and IVT structures that fall in the corresponding clusters relative to the total number of 6-hourly time instances during all EPEs that are linked to combined wave breaking and intense moisture transport. Circles for average precipitation volumes are not plotted where frequencies are below 0.1 %.

**Figure 10.** Composites of the tropospheric circulation for EPEs linked to combined Rossby wave breaking and intense moisture transport with (a-f) daily means and (g-l) normalized anomalies. The composites have a radius of 2000 km and are centred on the mass-weighted centres of the EPEs for subtropical (20°-40° N), extratropical (40°-60° N), and polar (60°-80° N) regions in the Northern Hemisphere over (first and third row) land and (second and fourth row) water during the period of 1979-2018. Shading in (a)-(f) shows daily precipitation (mm) and in (g)-(l) their normalized anomalies (SD). The contour lines in (a-f) show daily means of the 500-hPa geopotential height (gpm, in blue), mean sea level pressure (hPa; in black), and the IVT magnitude (150 kg m$^{-1}$ s$^{-1}$ contour in red) with IVT vectors where the IVT magnitude exceeds 150 kg m$^{-1}$ s$^{-1}$. The contour lines in (g-l) show normalized anomalies, in colours as in (a-f), in standard deviation (SD) for intervals of +/–0.5 SD starting at +/–0.5 SD, with only contours for negative values of 500-hPa geopotential height and mean sea level pressure, and for positive values of total column water (in green). Also, in (g-l), IVT vectors constructed from their zonal and meridional components, and only displayed where their vector length exceeds 1 SD.

**Figure 11.** Composites, as in Fig. 10, but for EPEs over subtropical land, for the period of 1979-2016, and for five exclusive categories of EPEs that match (a,f) *only* PV structures, (b,g) *only* IVT structures, (c,h) combined PV and IVT structures with a *direct* PV influence, (d,i) combined PV and IVT structures with *only* a *remote* PV influence, and (e,j) *neither* PV structures *nor* IVT structures, as detailed in the text and Table 1. The composites also include in (a)-(e) daily mean contours of QG vertical motion (Pa s$^{-1}$, in grey, only ascent is shown, denoted by negative values) and static stability (K m$^2$ kg$^{-1}$, in orange), and in (f)-(j) their normalized anomalies in identical colours, whereby those of static stability are displayed for intervals of –0.25 SD starting at –0.25 SD. Different from Fig. 10, the IVT contours in (a)-(e) correspond to a 200 kg m$^{-1}$ s$^{-1}$ value with IVT vectors shown where the IVT magnitude exceeds 200 kg m$^{-1}$ s$^{-1}$.

**Figure 12.** As Figure 11, but for EPEs over subtropical water.

**Figure 13.** Climatology of EPEs linked to the five exclusive categories, as detailed in section 6.2 and Table 1, as indicated by the legend from left to right: combined PV and IVT structures with a *direct* PV influence (PV & IVT, in purple), combined PV and IVT structures with *only* an *remote* PV influence (remote PV & IVT, in red), *only* PV structures (only PV, in blue), *only* IVT structures (only IVT, in green), and *neither* PV structures *nor* IVT structures (none, in brown-grey). In (a) their geographical distribution in fractions (%), (b) their share in the total number of EPEs, and (c) their share in six EPE clusters defined and ranked based on the precipitation volumes (%). Fractions in (a) are always shown for the category with the highest values, as shown in the legend, except for category 4 (remote PV & IVT), which are given advantage over category 3 (PV & IVT), provided the category 4 fractions exceed 30 %, as indicated by the legend, with the aim to emphasize the regions where this category is relatively important.

**Figure 14.** Schematic representation of the meteorological processes, including Rossby wave breaking and intense moisture transport, that lead the formation of EPEs (adapted from De Vries, 2017). This figure depicts (1) intense poleward and eastward moisture transport (green arrow) on the downstream flank of the stratospheric PV streamer (in blue), (2) an upper-level PV anomaly in blue and theta anomaly near the surface in red, that interact and mutually strengthen each other, consistent with baroclinic wave growth, and (3) forcing of upward motion induced by the upper-level PV anomaly; reduced static stability beneath the anomaly (in orange) and dynamical lifting (grey arrow) ahead. This model is based on well-established and longstanding concepts in the meteorology, including baroclinic wave growth in a PV perspective from Hoskins et al., (1985), their figures 15 and 21, and forcing mechanism of upward motion induced by an upper-level PV anomaly from Funatsu and Waugh (2008), their figure 1, the introduction section of Schlemmer et al. (2010), and section 6 of Martius et al. (2013). This study presents climatological evidence for these processes, based on the composites in Figs. 11 and 12, and integrates the role of Rossby wave breaking and intense moisture transport for the formation of EPEs into this model.

## Appendices

**Figure S1.** As Fig. 4, but with the PV and IVT fields, as depicted in Fig. 3a, used as input for the algorithm for PV and IVT structures.

**Figure S2.** As Fig. 6, but for the IVT$_{filt}$ form.

**Figure S3.** As Fig. 11, but for EPEs over extratropical land.

**Figure S4.** As Fig. 11, but for EPEs over extratropical water.

**Figure S5.** As Fig. 11, but for EPEs in the Southern Hemisphere over subtropical water.

**Figure S6.** As Fig. 11, but for EPEs in the Southern Hemisphere over extratropical water.

505 **Figure S7.** As Fig. 13a, but with EPE fractions in individual panels showing the full range of fractions. This Figure is analogue to Fig. 6, but with EPE fractions for the five exclusive categories.

**Table 1.** Methodology for matching EPEs with synoptic-scale processes

| Link to EPEs | Rossby wave breaking | Intense moisture transport | Rossby wave breaking & intense moisture transport (direct PV influence) | Rossby wave breaking & intense moisture transport (remote PV influence) | NONE |
|---|---|---|---|---|---|
| "inclusive" categories (sections 4.2 and 4.3) | All PV structures | All IVT structures | PV (direct overlap) & IVT structures | PV (direct & indirect overlap) & IVT structures | |
| "exclusive" categories (sections 6.2 and 7) | Only PV structures | Only IVT structures | PV (direct overlap) & IVT structures | PV (only indirect overlap) & IVT structures | Neither PV nor IVT structures |

⌊510

**Table 2.** Example cases of EPEs, flooding, and their socioeconomic impacts (EM-DAT)[a]

| Nr | Year | Month | Days | Country | Location[b] | Deaths | Affected people | Damage US$ (M) | Literature |
|---|---|---|---|---|---|---|---|---|---|
| 1 | 2014 | 9 | 27 | USA | Arizona | | | | Yang et al., 2019 |
| 2 | 2013 | 9 | 9-16 | USA | *Colorado, New Mexico* | 9 | 21,900 | 1,900 | Gochis et al., 2015; Eden et al., 2016 |
| 3 | 2002 | 1 | 9-11 | Mauritania Senegal | *Trarza, Brakna, Gorgol* *St. Louis, Matam, Louga* | 25 28 | 27,500 179,000 | 41 | Knippertz and Martin, 2005 |
| 4 | 2000 | 10 | 15 | Switzerland Italy | *Valais* *Lombardia, Piemonte* | 16 25 | 1,500 43,000 | 330 8,000 | Froidevaux and Martius, 2016; Lenggenhager et al., 2019 |
| 5 | 2009 | 11 | 25 | Saudi Arabia | Jeddah, Mecca cities | 161 | 10,000 | 900 | Haggag and El-Badry, 2013; De Vries et al., 2016; |
| 6 | 2010 | 7 | 19-22 28-30 | Pakistan | *Balochistan* *Balochistan*[c] | 60 1,985 | 4,000, 20,356,550 | - 9,500 | Martius et al., 2013; Hong et al., 2011 |
| 7 | 2013 | 6 | 17-18 | India | *Uttarakhand*[c] | 6,054 | 504,473 | 1,100 | Joseph et al., 2015; Kumar et al., 2016; Vellore et al., 2016 |
| 8 | 2015 | 3 | 24-26 | Chile | *Atacama, Amtofagasta, Coquimbo* | 178 | 193,881 | 1,500 | Wilcox et al., 2016; Bozkurt et al., 2016; Rondanelli et al., 2019 |
| 9 | 2007 | 3 | 25-31 | Argentina Uruguay | *Santa Fe, Entre Rios Colonia*[c] | 5 | 70,000 | 10 | Cavalcanti, 2012 |
| 10 | 2006 | 4 | 16-23 | Namibia | | | | | Muller et al., 2008 |
| 11 | 1987 | 9 | 26-29 | South Africa | Natal, Kwazulu | 506 | 65,000 | 765 | Triegaardt et al., 1988 van Heerden and Taljaard, 1998 |
| 12 | 2011 | 1 | 9-14 | Australia | | | | | Whelan & Frederiksen, 2017 |

[a]the reported socioeconomic impacts include the disaster (sub)types flash floods, riverine floods, and landslides.

[b]text in italic refers to provinces

[c]only one of several affected provinces reported is listed

⌊515

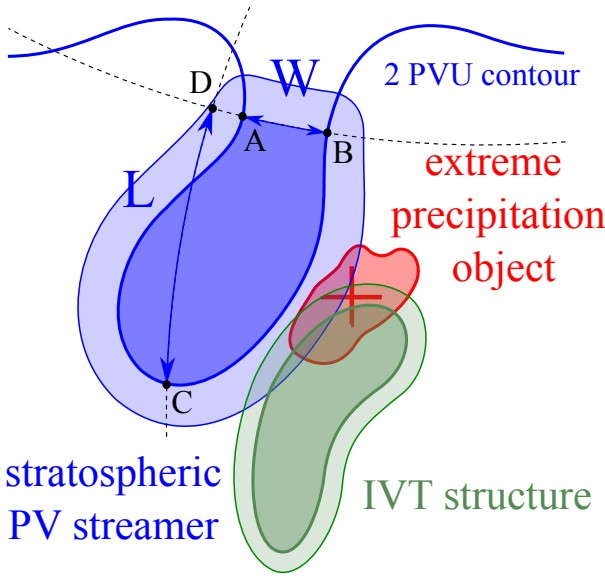

Figure 1. Schematic representation of Rossby wave breaking (stratospheric PV streamer in blue) and intense moisture transport (IVT structure in green) as synoptic-scale processes of EPEs (extreme precipitation object in red), adapted from De Vries et al., (2018). The light blue and light green shadings illustrate the area of influence of the synoptic features, and the symbols W and L refer to the geometrical criteria of PV streamers, as further detailed in section 2.

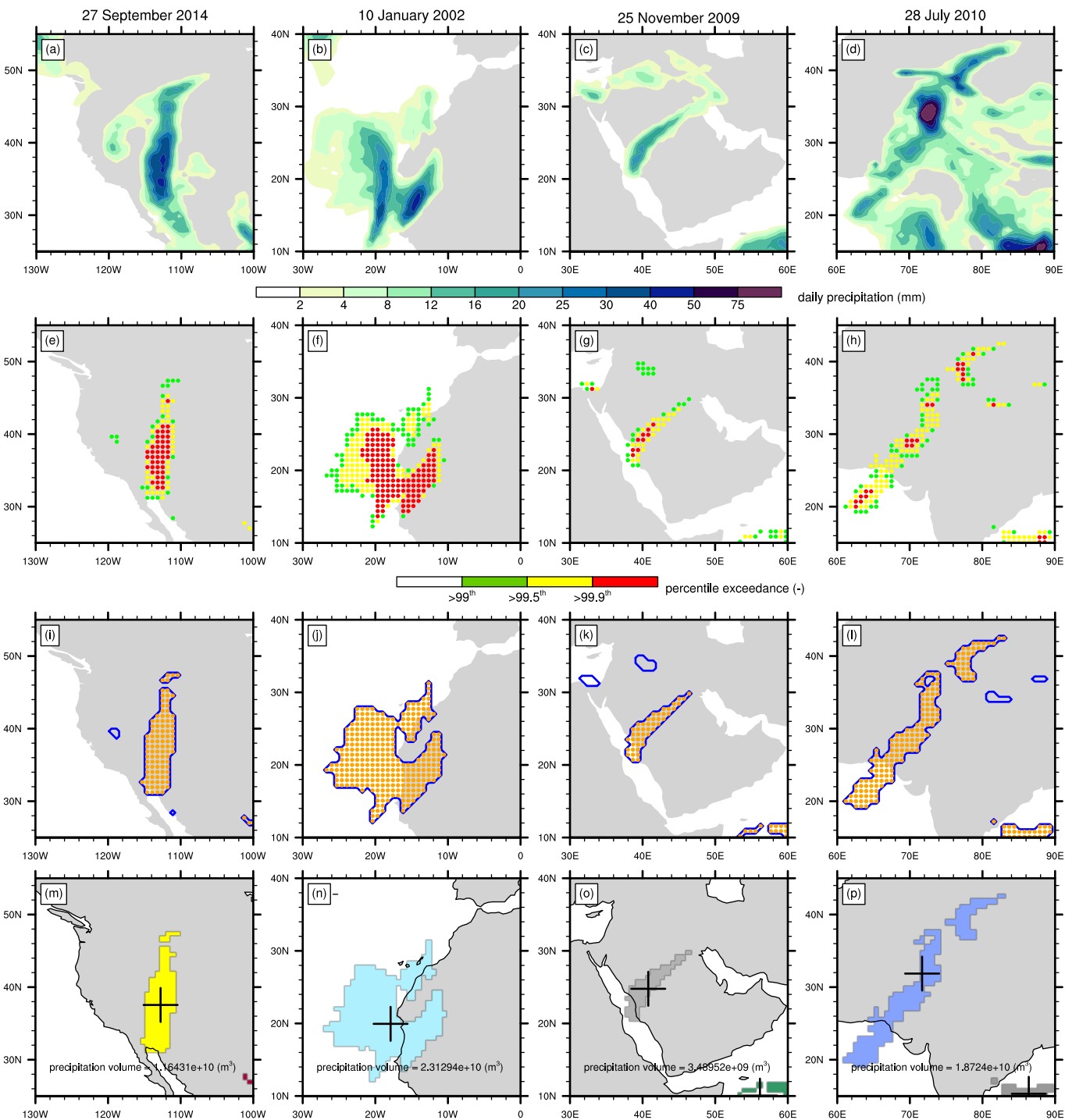

Figure 2. Examples of the algorithm for identification of EPEs for (first column) Arizona, USA, 27 September 2014, (second column) northwest Africa, 10 January 2002, (third column) Jeddah, Saudi Arabia, 25 November 2009, and (fourth column) Pakistan, 28 July 2010. In (a)-(d) daily rainfall amounts, (e)-(h) grid points that exceed the annual >99th (green), 99.5th (yellow), and 99.9th (red) percentiles, (j)-(l) extreme precipitation objects with a surface > 50.000 km2 and consisting of connected grid points within R < 250 km, and (m)-(p) extreme precipitation objects illustrated by random colours, with their mass-weighted centres denoted by the markers, and their volumes (m3) stated in the text.

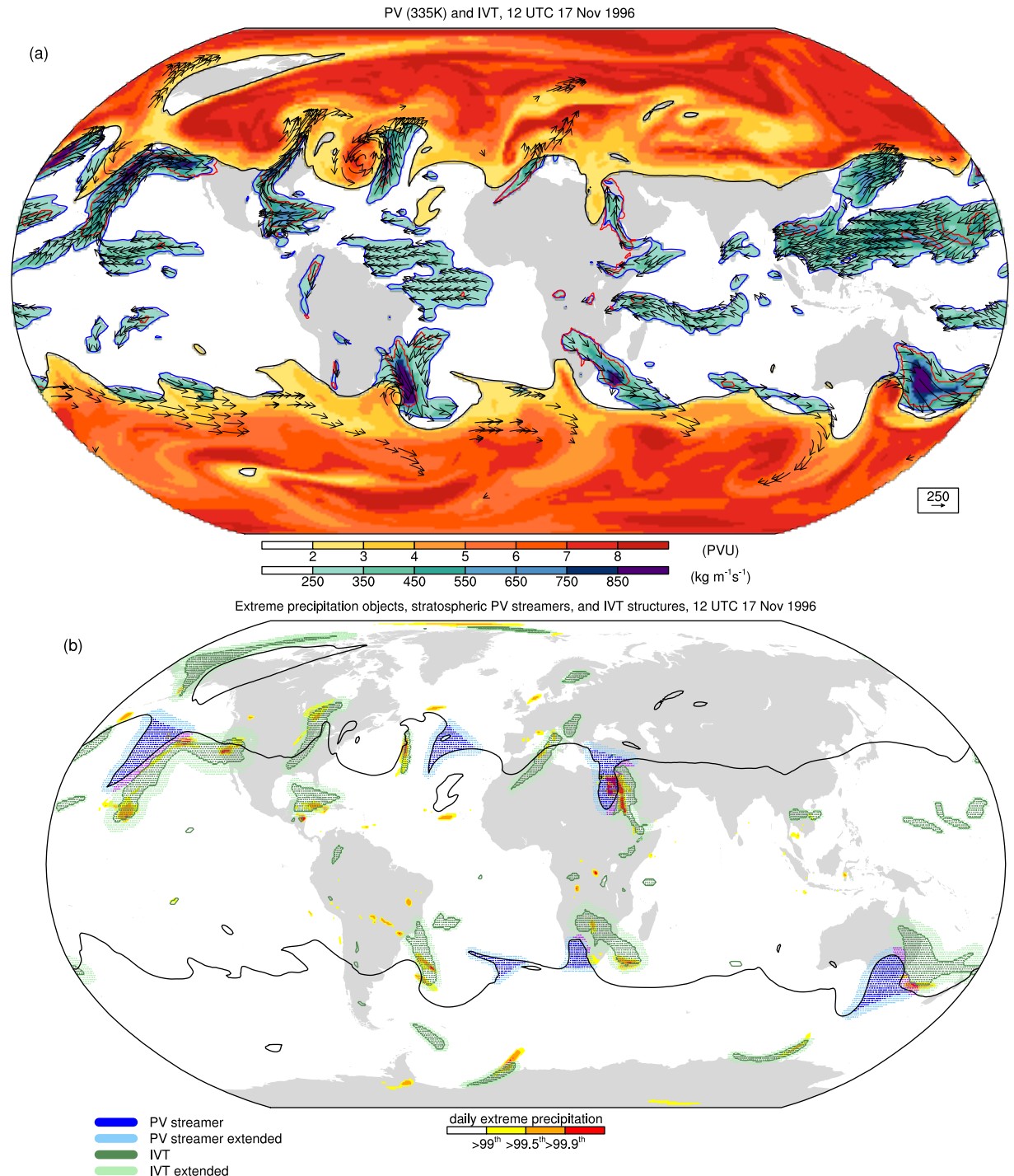

Figure 3. Example of the algorithm (a) input and (b) output for stratospheric PV streamers and IVTpct structures at 12 UTC 17 November 1996. In (a) PV (PVU) on 335 K in yellow/red colours, and full IVT (kg m–1 s–1) in green/blue colours, and IVT vectors in black, only plotted where the full IVT magnitude > 250 kg m–1 s–1. Contour lines denote the +/–2 PVU contours (black), the 250 kg m–1 s–1 full IVT (blue), and the annual 95th IVT percentile with a lower threshold of 200 kg m–1 s–1 (red). In (b) the identified stratospheric PV streamers, IVT structures in the IVTpct form, and their areas of influence, as indicated by the colours in the legend. Also, (b) shows the 99th, 99.5th and 99.9th percentile exceedances of daily ERA-Interim precipitation.

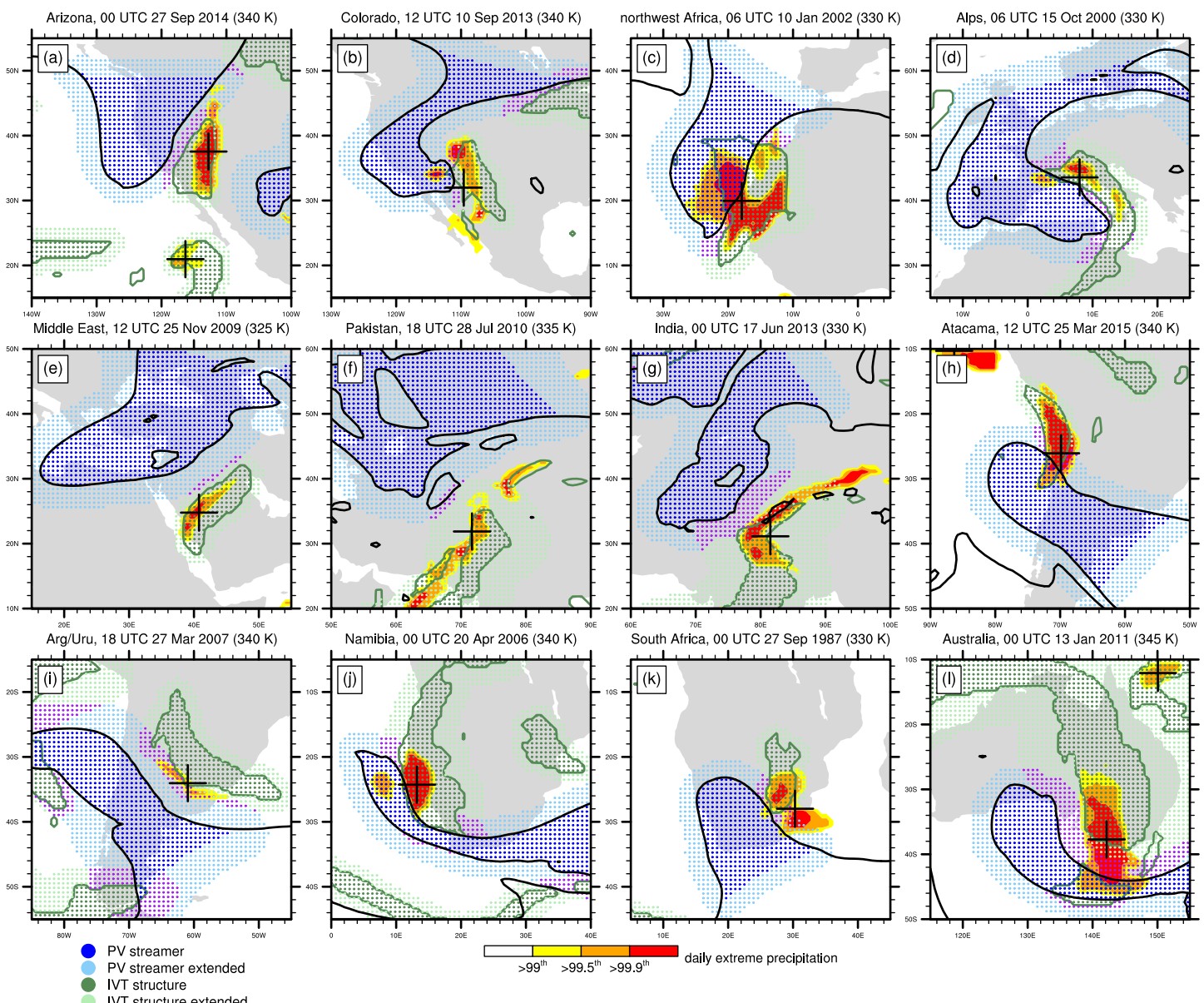

Figure 4. Illustrative cases of extreme precipitation and flooding accompanied by stratospheric PV streamers and IVT structures in (a) Arizona, USA, 00 UTC 27 September 2014, (b) Colorado, USA, 12 UTC 10 September 2013, (c) northwest Africa, 06 UTC 10 January 2002, (d) the Alpine region, 06 UTC 15 October 2000, (e) the Middle East, Jeddah, 12 UTC 25 November 2009, (f) Pakistan, 18 UTC 28 July 2010, (g) India, 00 UTC 17 June 2013, (h) the Atacama Desert in Chile, 12 UTC 25 March 2015, (i) Argentina and Uruguay, 18 UTC 28 March 2007, (j) Namibia, 00 UTC 20 April 2006, (k) South Africa, 00 UTC 27 September 1987, and (l) Australia, 00 UTC 13 January 2011. All meteorological variables and identified objects correspond to those as in Fig. 3b with the exception of extreme precipitation for which only the objects instead of all percentile-exceeding grid points are shown.

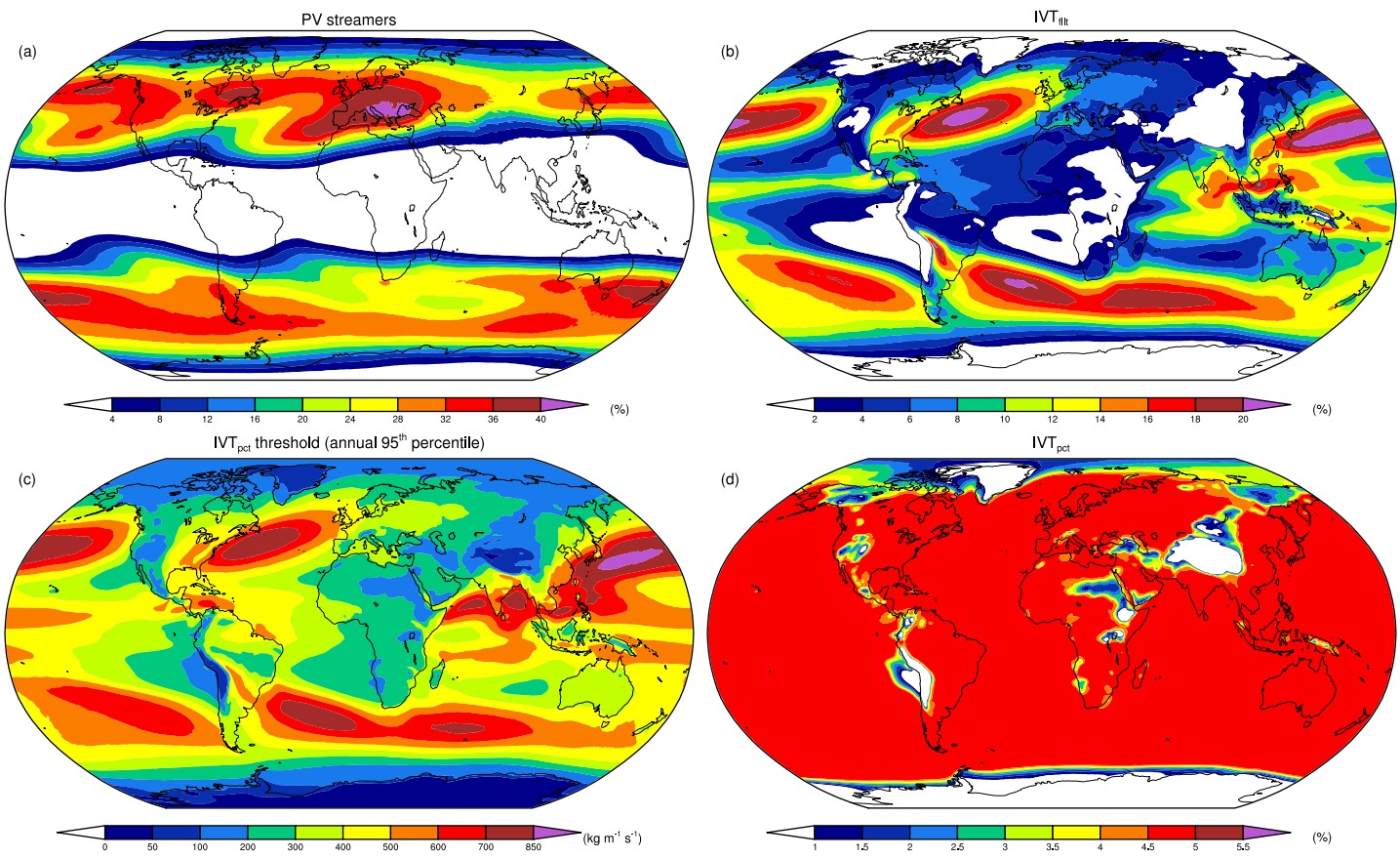

Figure 5. Climatological frequencies, based on 6-hourly occurrences for the period of 1979-2018, of (a) vertically aggregated stratospheric PV streamers between 300-350 K, (b) IVT structures in the IVTfilt form, and (d) IVT structures in the IVTpct form. In (c) IVT values (kg m–1 s–1) that correspond to the annual 95th percentiles of full IVT magnitudes used as threshold for the identification of IVTpct structures.

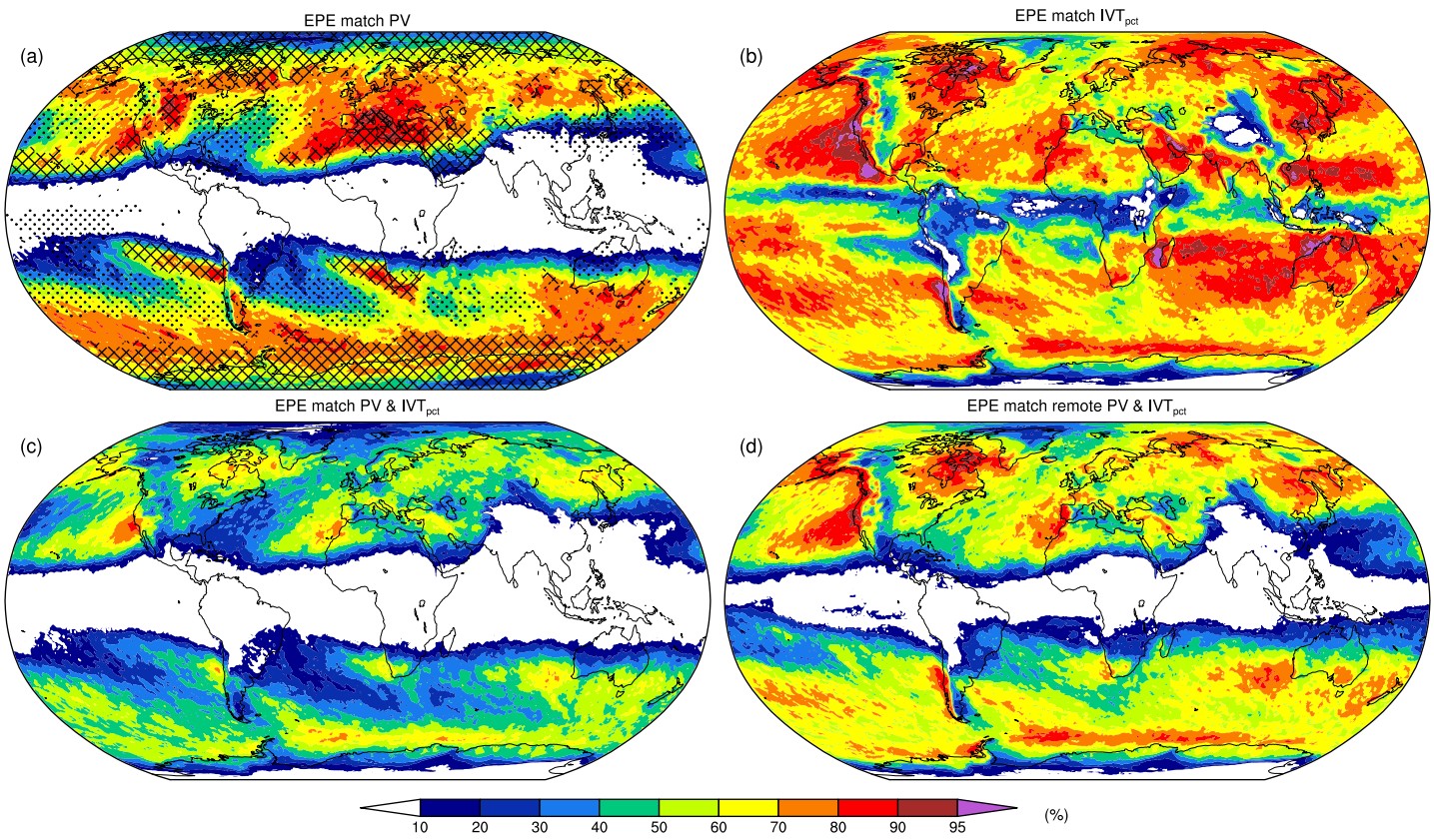

Figure 6. Fractions (%) of EPE matches with (a) PV structures, (b) IVTpct structures, (c) combined PV and IVTpct structures with a direct PV influence, and (d) combined PV and IVTpct structures whereby also a remote PV influence is allowed, see the text for details. The climatologies are based on the ERA-Interim period of 1979-2018. All matches that involve PV structures are based on isentropic surfaces between 300-350 K with 5 K intervals. In (a) the crossed hatching and stippling indicates statistically significant fractions that are larger and smaller, respectively, than expected under independence of EPEs and PV structures (see the text for details), whereas in (b)-(d) only statistically significant fractions are plotted that are larger than expected under independence of EPEs and the structures.

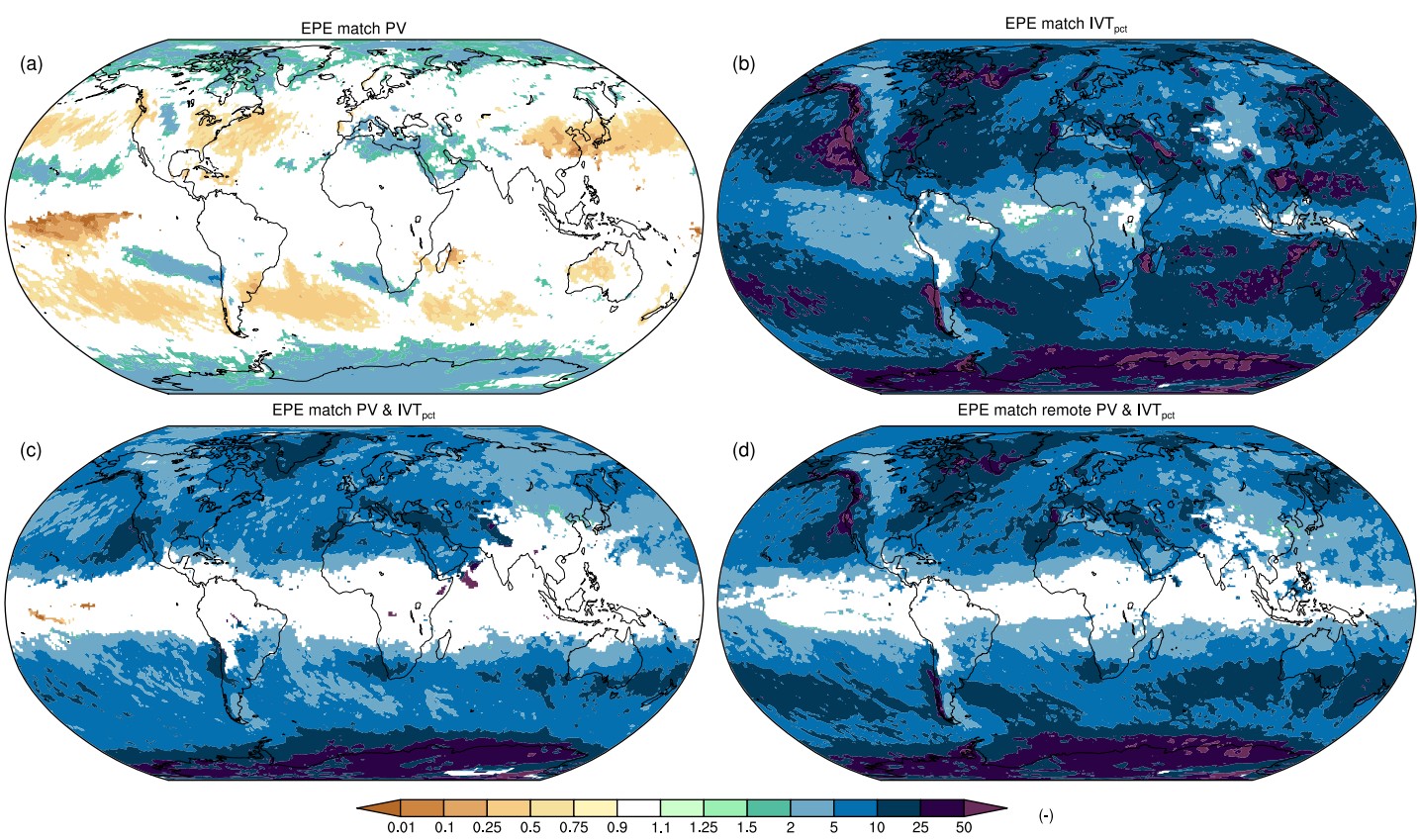

Figure 7. Odds ratios of EPE matches with (a) PV structures, (b) IVTpct structures, (c) combined PV and IVTpct structures with a direct PV influence, and (d) combined PV and IVTpct structures whereby also a remote PV influence is allowed, i.e., observed matching odds divided by the matching odds expected under independence of EPEs and the PV and/or IVT structures. Odds ratios are plotted transparent where the link between EPEs and these structures is not statistically significant.

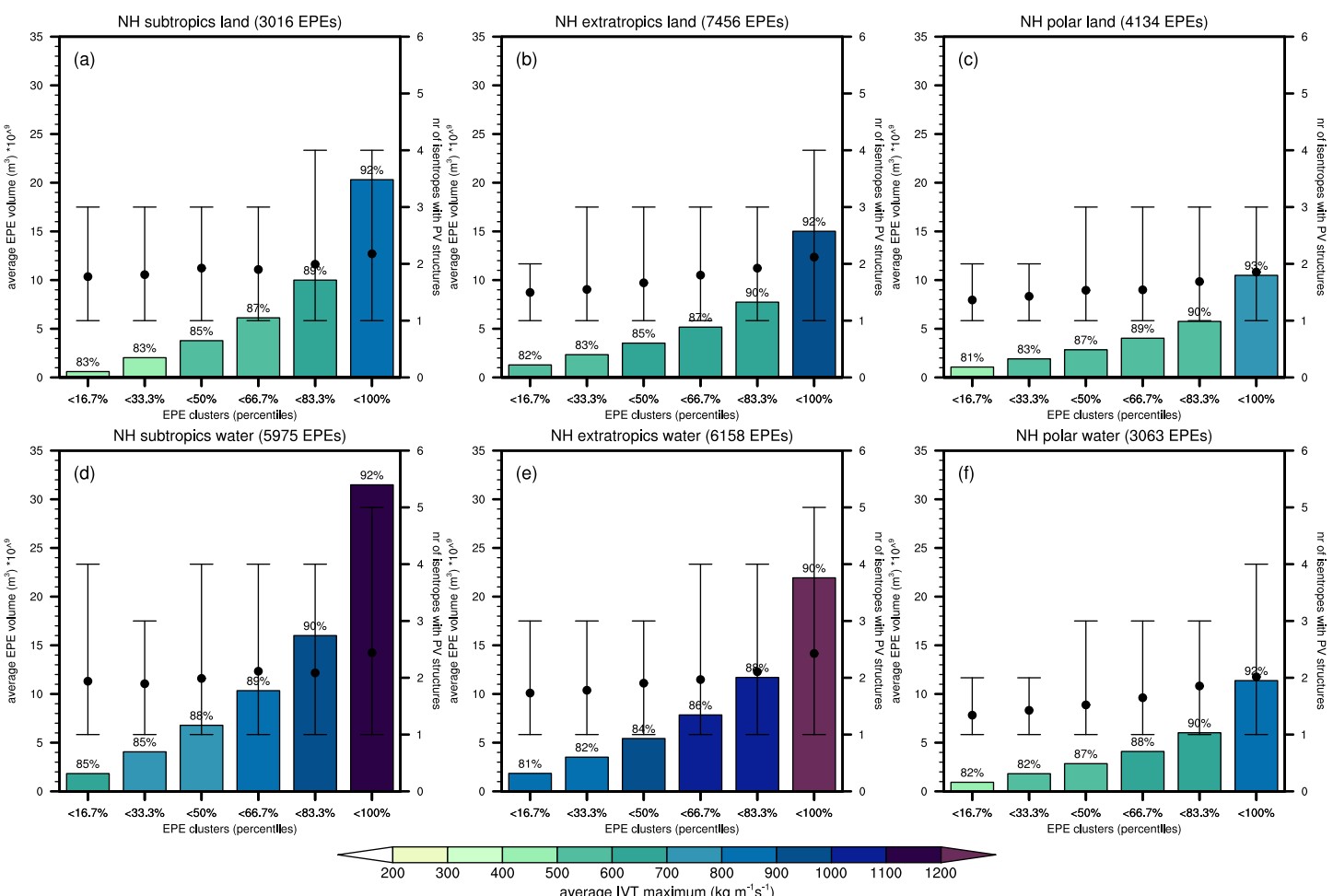

Figure 8. Characteristics of PV and IVT structures for EPEs, linked to combined wave breaking and intense moisture transport, divided and ranked in 6 clusters based on their precipitation volumes for the Northern Hemispheric (left) subtropics (20°-40° N), (middle) extratropics (40°-60° N), and (right) polar regions (60°-80° N), over (top) land and (bottom) water. The number of isentropes with PV structures is shown by the 10th to 90th percentile range and the average indicated by the dot, and the average of the IVT maxima (kg m–1 s–1) is shown by the colours of the bars. The height of the bars corresponds to the average precipitation volumes (m3) of each EPE cluster. The number of EPEs for each region is included in the figure titles, and the numbers above the colour bars indicate the fractions (%) of 6-hourly time instances with combined PV and IVT structures during the daily EPEs.

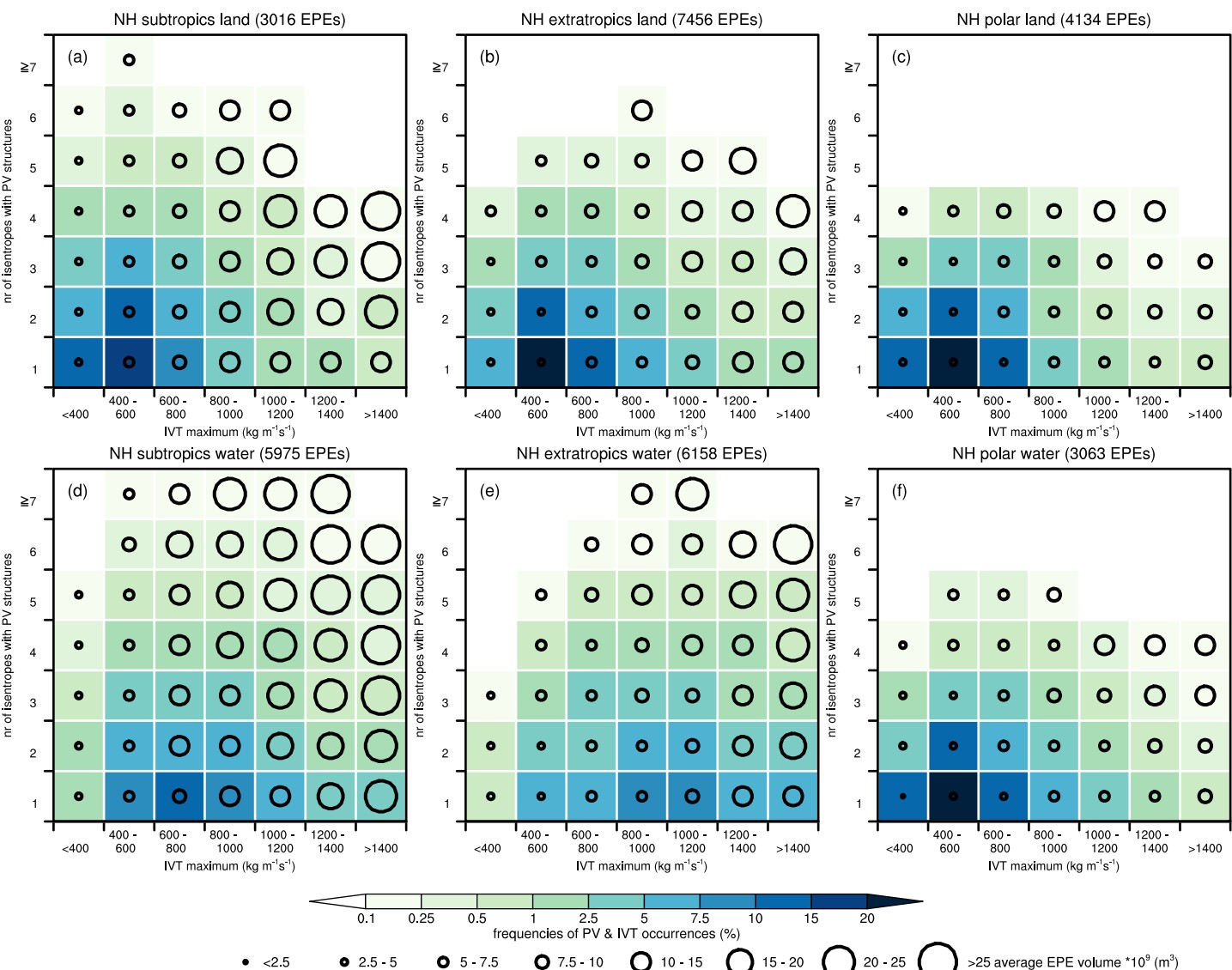

Figure 9. Average precipitation volumes of the EPEs in the circles, as denoted by the legend, for clusters defined by the number of isentropes with PV structures and the IVT maxima (kg m–1 s–1). As in Fig. 7, the plot is based on EPEs, linked to combined wave breaking and intense moisture transport, for identical regions. The colours of the squares reflect the frequencies (%) of 6-hourly time instances with PV and IVT structures that fall in the corresponding clusters relative to the total number of 6-hourly time instances during all EPEs that are linked to combined wave breaking and intense moisture transport. Circles for average precipitation volumes are not plotted where frequencies are below 0.1 %.

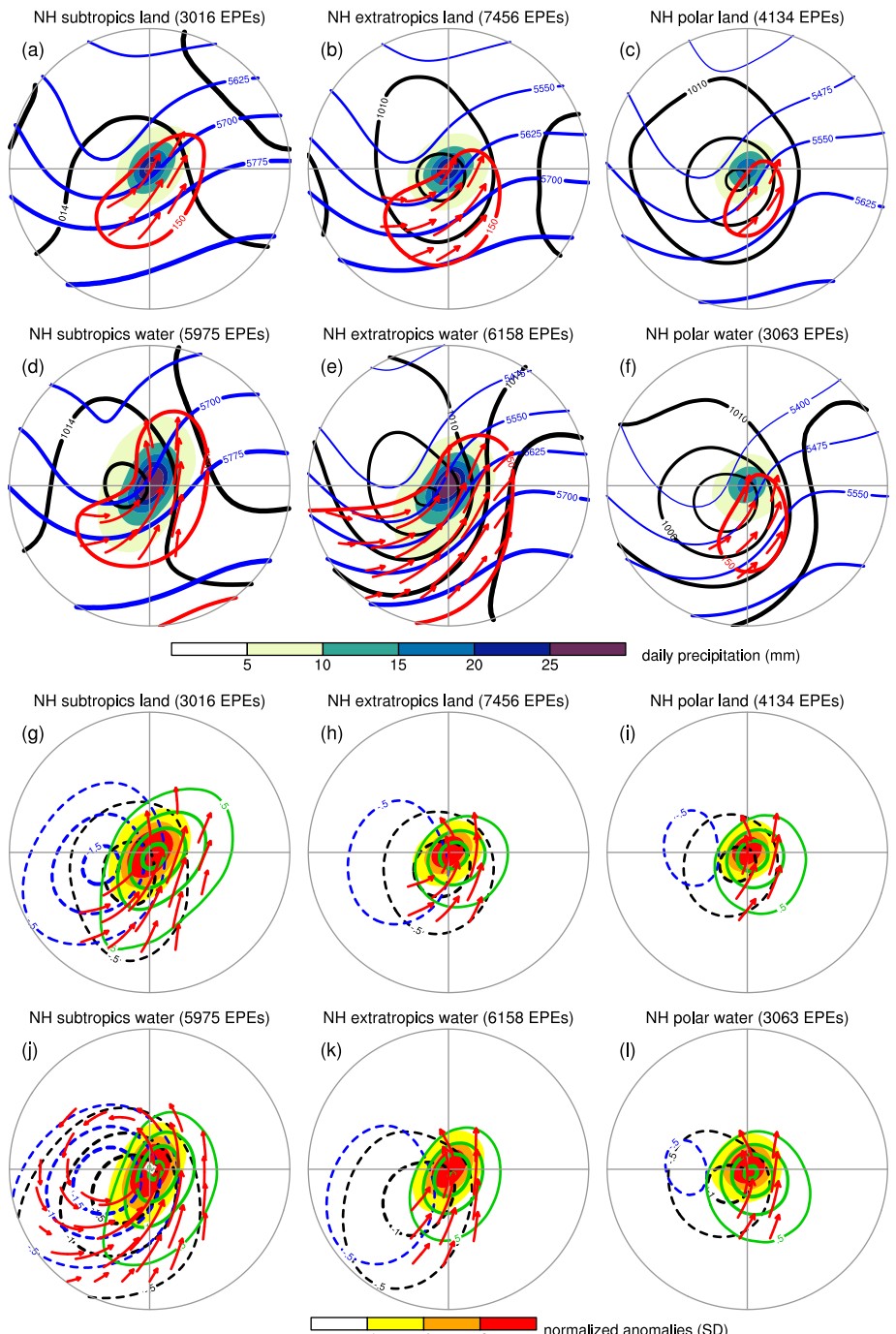

Figure 10. Composites of the tropospheric circulation for EPEs linked to combined Rossby wave breaking and intense moisture transport with (a-f) daily means and (g-l) normalized anomalies. The composites have a radius of 2000 km and are centred on the mass-weighted centres of the EPEs for subtropical (20°-40° N), extratropical (40°-60° N), and polar (60°-80° N) regions in the Northern Hemisphere over (first and third row) land and (second and fourth row) water during the period of 1979-2018. Shading in (a)-(f) shows daily precipitation (mm) and in (g)-(l) their normalized anomalies (SD). The contour lines in (a-f) show daily means of the 500-hPa geopotential height (gpm, in blue), mean sea level pressure (hPa; in black), and the IVT magnitude (150 kg m–1 s–1 contour in red) with IVT vectors where the IVT magnitude exceeds 150 kg m–1 s–1. The contour lines in (g-l) show normalized anomalies, in colours as in (a-f), in standard deviation (SD) for intervals of +/–0.5 SD starting at +/–0.5 SD, with only contours for negative values of 500-hPa geopotential height and mean sea level pressure, and for positive values of total column water (in green). Also, in (g)-(l), IVT vectors constructed from their zonal and meridional components, and only displayed where their vector length exceeds 1 SD.

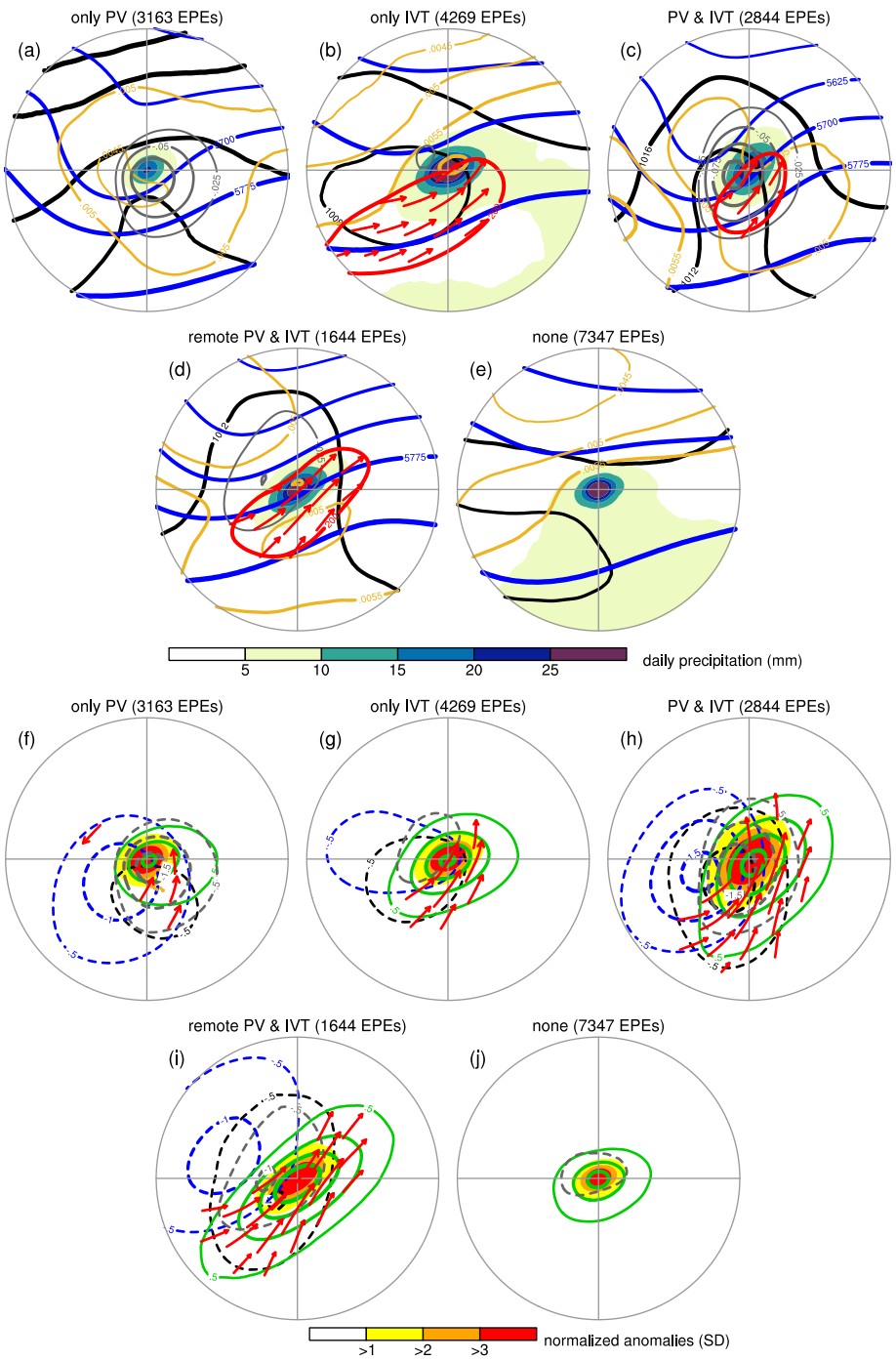

Figure 11. Composites, as in Fig. 10, but for EPEs over subtropical land, for the period of 1979-2016, and for five exclusive categories of EPEs that match (a,f) only PV structures, (b,g) only IVT structures, (c,h) combined PV and IVT structures with a direct PV influence, (d,i) combined PV and IVT structures with only a remote PV influence, and (e,j) neither PV structures nor IVT structures, as detailed in the text and Table 1. The composites also include in (a)-(e) daily mean contours of QG vertical motion (Pa s–1, in grey, only ascent is shown, denoted by negative values) and static stability (K m2 kg-1, in orange), and in (f)-(j) their normalized anomalies in identical colours, whereby those of static stability are displayed for intervals of –0.25 SD starting at –0.25 SD. Different from Fig. 10, the IVT contours in (a)-(e) correspond to a 200 kg m–1 s–1 value with IVT vectors shown where the IVT magnitude exceeds 200 kg m–1 s–1.

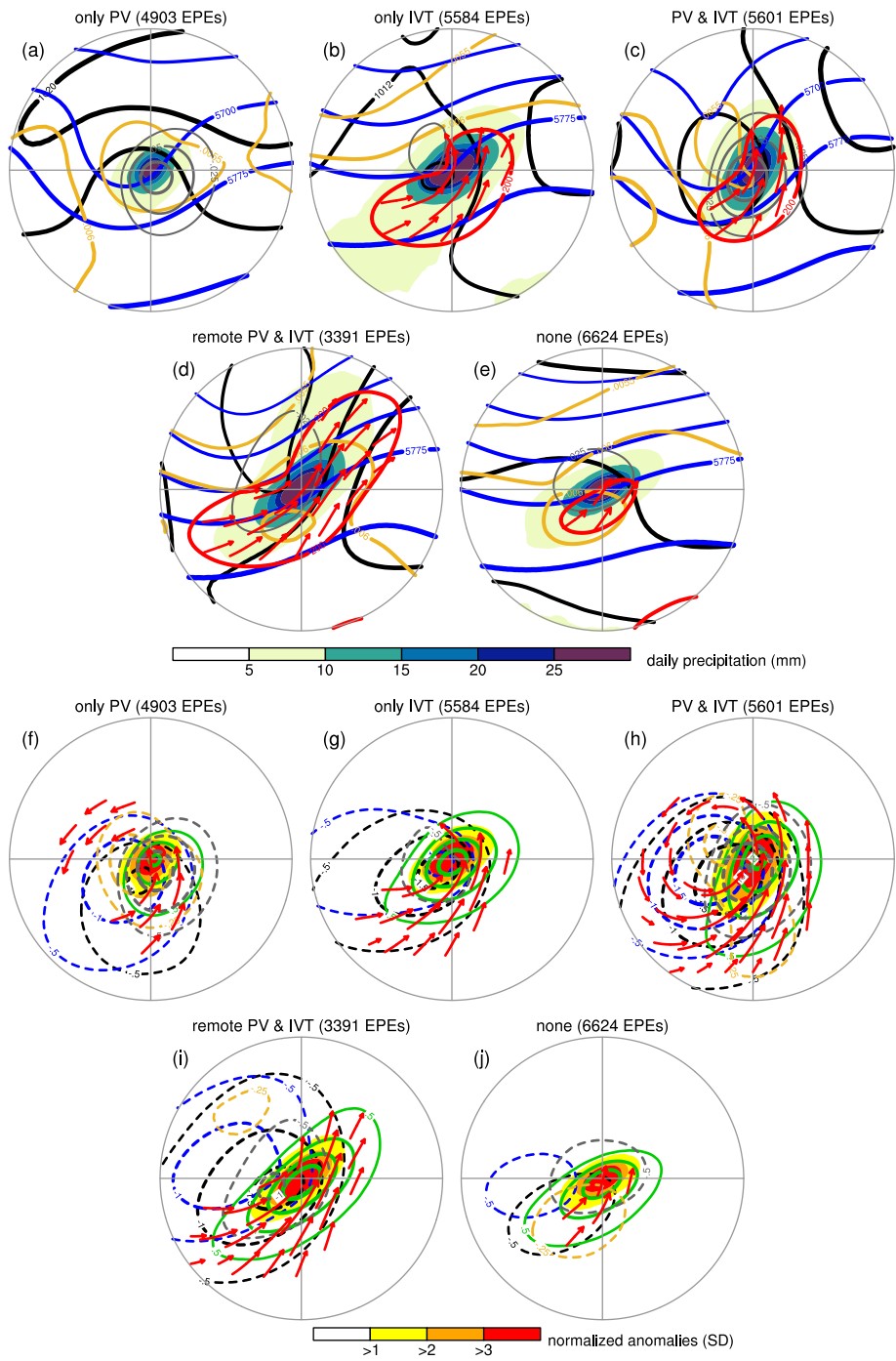

Figure 12. As Figure 11, but for EPEs over subtropical water.

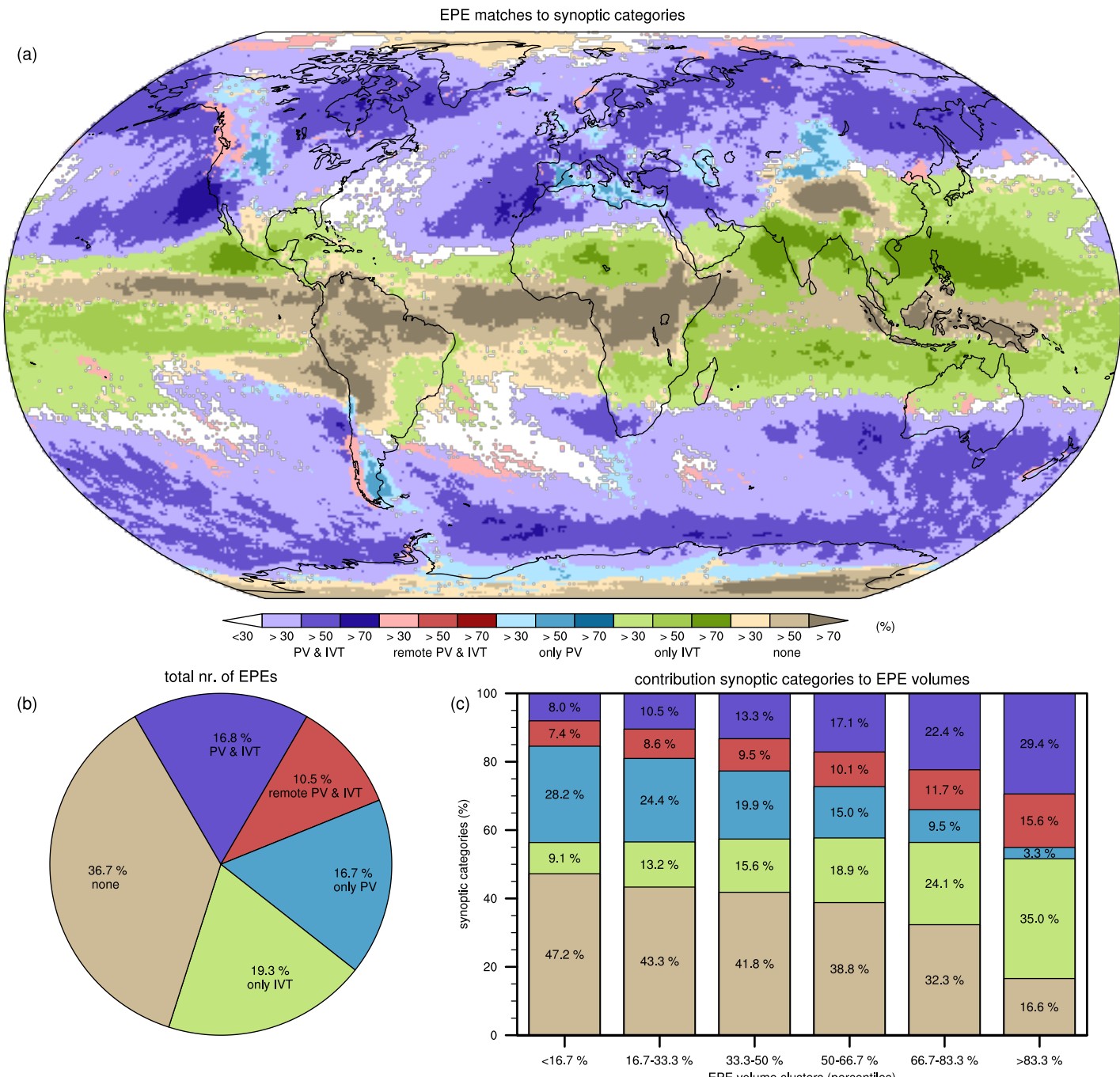

Figure 13. Climatology of EPEs linked to the five exclusive categories, as detailed in section 6.2 and Table 1, as indicated by the legend from left to right: combined PV and IVT structures with a direct PV influence (PV & IVT, in purple), combined PV and IVT structures with only an remote PV influence (remote PV & IVT, in red), only PV structures (only PV, in blue), only IVT structures (only IVT, in green), and neither PV structures nor IVT structures (none, in brown-grey). In (a) their geographical distribution in fractions (%), (b) their share in the total number of EPEs, and (c) their share in six EPE clusters defined and ranked based on the precipitation volumes (%). Fractions in (a) are always shown for the category with the highest values, as shown in the legend, except for category 4 (remote PV & IVT), which are given advantage over category 3 (PV & IVT), provided the category 4 fractions exceed 30 %, as indicated by the legend, with the aim to emphasize the regions where this category is relatively important.

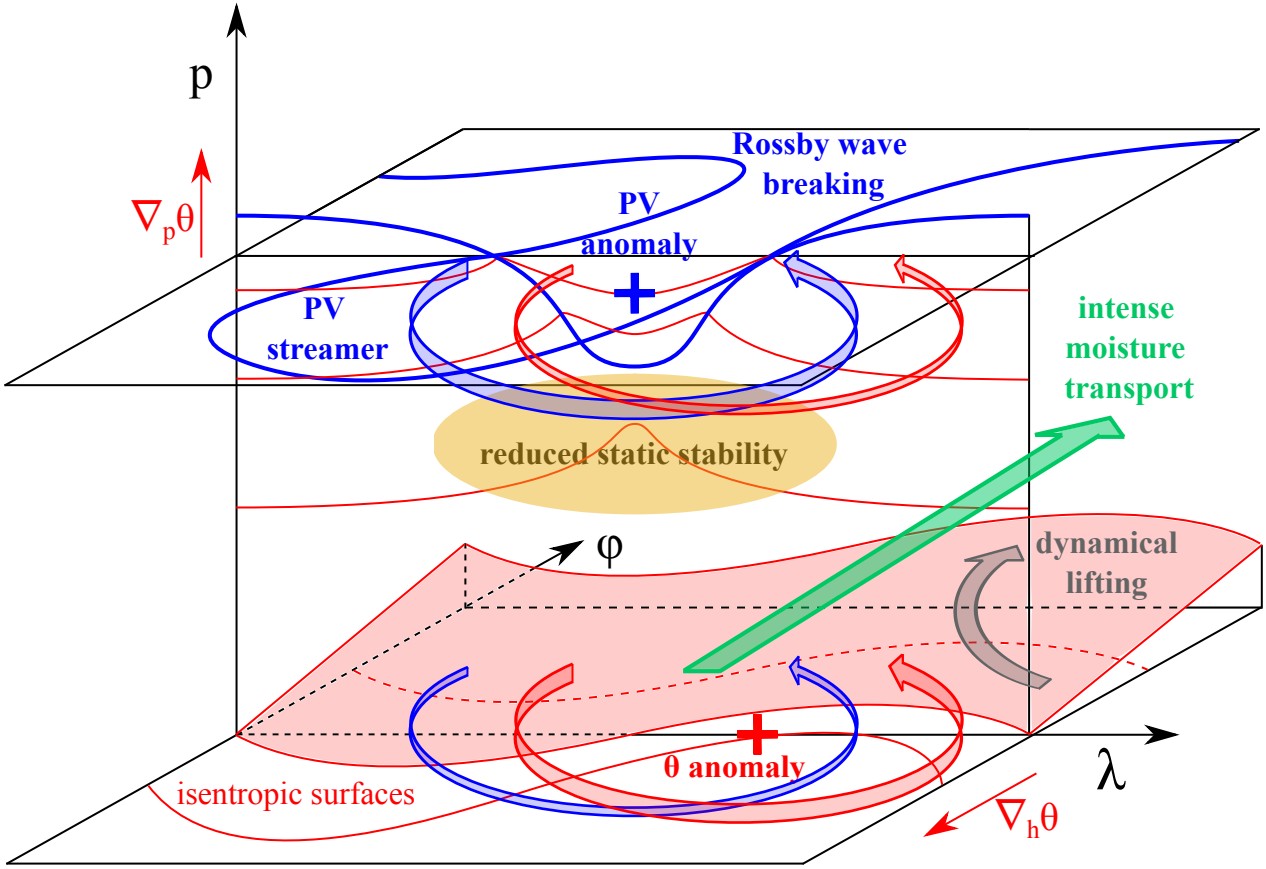

Figure 14. Schematic representation of the meteorological processes, including Rossby wave breaking and intense moisture transport, that lead the formation of EPEs (adapted from De Vries, 2017). This figure depicts (1) intense poleward and eastward moisture transport (green arrow) on the downstream flank of the stratospheric PV streamer (in blue), (2) an upper-level PV anomaly in blue and theta anomaly near the surface in red, that interact and mutually strengthen each other, consistent with baroclinic wave growth, and (3) forcing of upward motion induced by the upper-level PV anomaly; reduced static stability beneath the anomaly (in orange) and dynamical lifting (grey arrow) ahead. This model is based on well-established and longstanding concepts in the meteorology, including baroclinic wave growth in a PV perspective from Hoskins et al., (1985), their figures 15 and 21, and forcing mechanism of upward motion induced by an upper-level PV anomaly from Funatsu and Waugh (2008), their figure 1, the introduction section of Schlemmer et al. (2010), and section 6 of Martius et al. (2013). This study presents climatological evidence for these processes, based on the composites in Figs. 11 and 12, and integrates the role of Rossby wave breaking and intense moisture transport for the formation of EPEs into this model.