# Peer review of "A global climatological perspective on the importance of Rossby wave breaking and intense moisture transport for extreme precipitation events"

_Weather and Climate Dynamics, 2020_

## Referee Comment (RC1) · Anonymous Referee #1 · 27 Oct 2020

A global climatological perspective on the importance of Rossby wave breaking and intense moisture transport for extreme precipitation events

General comments

In this paper diagnostics are used to identify PV streamers, extreme precipitation events and high IVT structures in reanalysis data. The relationship between these features are examined and relative importance of the different features contributing to extreme precipitation events quantified. Events are partitioned into 5 synoptic categories summarising the main influences allowing a climatology of categories and their geographical locations to be produced. Overall the paper is well written with careful analysis of the data and interpretation of the results. 60% of the extreme precipitation events can be attributed to one of the 5 categories and over 85% of the most intense extreme precipitation events demonstrating the importance of these combined features. The subject of this paper is suitable for publication in WCD although before publication some minor changes to the text are required (see below).

Major comments

1.  Line 11, line 129 and elsewhere: There is little motivation for this work in the abstract and introduction. I agree that previous studies have considered RWB and IVT separately, but what is the motivation for considering them together? Why is this a good idea? I assume that the author regards the fact that ascent (or cooling) is necessary to convert water vapour into liquid water and further precipitation is unnecessary to include. However, given the number of studies that use IVT as a proxy for precipitation and EPE I think this is a key point to include to motivate the work.

2.  Introduction: A vast amount of literature is covered in the introduction section. To the extent that it is quite difficult to read the paragraphs due to the very large number of references. I appreciate that the statements included in the introduction need to be supported by published literature, but is it necessary to include well over 100 references? This would be suitable for a review paper but not for a paper containing new science. The material included in the introduction needs to be more focused to identify the knowledge gaps in the published literature and motivate the study presented in the paper.

3.  Figures: Most of the figures use a rainbow colour bar. This made it difficult to view the figures in black and white and is also very difficult for red-green colour-blind people. Please consider using a different colour bar in your figures.

Minor comments

1.  Line 8: 'Frequently' should be 'frequent' I think.
2.  Line 17: The statistics presented here are rather one-sided. For example, if > 90% of EPEs are associated with RWB, how many RWB events are not associated with EPE's.
3.  Line 21: I'm not familiar with the term 'outer tropics', is this the same region as the subtropics?
4.  Line 58: What is the difference between a tropical moisture export and an atmospheric river?
5.  Line 121: What does the author mean by 'the precise connection between the driving synoptic processes and the generation of extreme precipitation often remains more in the background and less understood'. This is a bit cryptic. Are they referring to microphysical processes?

6. Line 286: What do the modulus sign represent. Is the 2 PVU definition of the tropopause appropriate in the tropics? E.g. Wilcox et al. (2011)
Wilcox, L.J., Hoskins, B.J. and Shine, K.P., 2011. A global blended tropopause based on ERA data. Part 1: Climatology. *QJR Meteorol. Soc.*
7. Line 328: Given that the climatologies are so different when different thresholds are used, does this mean that there is no robust definition of this feature?
8. Line 330: Why is an area of 10,000 $km^2$ used? How sensitive are the results used to this threshold?
9. Line 333: Why are the IVT structures 'provided with an extended area'? How was this area chosen? Are the results sensitive to the size of this area?
10. Lines 392-400: There are 12 case studies presented in figure 4 but only 1 paragraph devoted to their description. Given that they represent a variety of time evolving processes, it seems simplistic to group them into 2 different constellations. More evidence is needed to support this binary grouping.
11. Line 421: The PV streamers are 'relatively frequent' compared to what?
12. Line 430-434: I'm afraid I didn't follow the argument regarding the $IVT_{pct}$. Could this section be made clearer?
13. Line 459: Here the statistics are rather one-sided. How many PV structures are not associated with EPEs?
14. Pages 15, 16 and elsewhere: Here and elsewhere the author makes use of lists to analyse the figures. While this can be a useful scientific writing tool, overuse becomes a bit repetitive after a while and I found myself skipping to the end of the list without reading each point.
15. Line 498: Why are many regions of combined RWB and IVT positioned at the equatorward exit regions and poleward entrance regions of the extratropical storm tracks?
16. Line 532. I'm not sure figure 7 is necessary. What does it add to the arguments already stated in the paper?
17. Line 634 and elsewhere: Here the author refers to 'category 1' and later to categories 2-5. Perhaps I missed this, but where are these different categories defined? Are these the categories in the synthesis section? If so, they shouldn't be referred to in section 6.
18. Line 740: What do the numbers in brackets mean? >90% of what?

---

## Referee Comment (RC2) · Anonymous Referee #2 · 27 Oct 2020

General comments:

A global systematic analysis of the main processes contributing to EPE genesis is presented. In particular, the occurrence of two main drivers (in tandem and isolation) is considered for the first time on a global scale; Rossby wave breaking and IVT. To this end, this represents a very original contribution to the understanding of EPE generation.

The analysis builds on detailed and careful analysis of the data, on a large number

of variables. I find figures excellent and well-illustrating results. However, the text sometimes is a bit too dense and difficult to follow due to overuse of bullet (numbered) lists, which breaks the flow of the text.

My suggestion, for example, is to reduce the abstract which I found far too long and detailed in the description of the results. See the attached pdf for detailed comments also on the abstract.

Major comments:

Seasonality is not explicitely considered in this analysis. I wonder how the repartition of these five categories in the space could change in the seasons.

Lines 323-325: The definition of IVTpct is not totally clear: Do you subtract to the full IVT the 95° percentile of the annual distribution at each grid point ? If yes, state this.

Lines 585-594: The difference between subtropical and extratropical circulation around EPE it is not very clear. Perhaps commenting more on the anomaly panels, one could see better the difference in baroclinicity with a more pronounced back-tilted cyclones in the extratropics.

Lines 710-713 . I found weird the explanation of white areas in Fig.12 . It is stated that these areas coincides with those where cyclones and warm-conveyor belts have the highest relevance. Correct. But the same areas should also display substantial IVT values, like over the western Atlantic. If you have cyclones and WCB, usually there are also high IVT values (at least), in addition to upper level waves. I wonder if this lack of classification in this region it is not due to the definition of IVTpct objects which set to a too high treshold for this region in which very frequently IVT is very high. This is not clear to me and it deserves a clarification.

In addition to these few major points I have noted few technical corrections on the annoted pdf attached below.

Please also note the supplement to this comment:
https://wcd.copernicus.org/preprints/wcd-2020-44/wcd-2020-44-RC2-supplement.pdf
* * *
[Figure]

**Supplement:**

[revised manuscript text omitted]

---

## Author Comment (AC1) · 27 Nov 2020

**Author response to referee comments on manuscript WCD-2020-44**

**Title: "A global climatological perspective on the importance of Rossby wave breaking and intense moisture transport for extreme precipitation events"**

**Author: A. J. de Vries**

The author would like to thank the two anonymous reviewers for their helpful comments. Each of the comments was carefully considered for improving the manuscript. Most of the comments led to changes in the text with the aim to improve the clarity and readability of the manuscript. The most substantial changes are

- Replacement of a very long detailed sentence in the abstract by a concise one
- Include a motivation in the introduction for the study of the individual and combined importance of wave breaking and moisture transport by describing the mechanisms of how PV streamers can facilitate extreme precipitation formation
- Removal of numbered lists in the text where different regions are discussed
- Removal of a few references in the introduction and placing others at one location in the sentence instead of scattered over different parts
- Rephrasing the definition of the $IVT_{pct}$ structures

The text below (in blue) responds to each of the reviewer's comments and indicates what changes in the text will be made, answers to questions and concerns raised, and at few locations, provides arguments why we prefer to refrain from specific suggested revisions.

***Authors response to the comments of Reviewer 1***

Major comments
1. Line 11, line 129 and elsewhere: There is little motivation for this work in the abstract and introduction. I agree that previous studies have considered RWB and IVT separately, but what is the motivation for considering them together? Why is this a good idea? I assume that the author regards the fact that ascent (or cooling) is necessary to convert water vapour into liquid water and further precipitation is unnecessary to include. However, given the number of studies that use IVT as a proxy for precipitation and EPE I think this is a key point to include to motivate the work.

*Previous studies have mostly investigated the role of Rossby wave breaking and intense moisture transport for the formation of extreme precipitation events separately. Also, previous studies that addressed the linkage between wave breaking and EPEs were confined to specific regions. This study brings for the first time together these two synoptic-scale processes as drivers of EPEs at the global scale. This motivation is explicitly articulated in the original version of the manuscript, more specifically, the lines 9-12 on page 1 in the abstract, and lines 119-129 and 131-137 in the introduction. An important reason to combine these two (thermo)dynamic processes is that Rossby wave breaking can initiate extreme precipitation events through (i) steering intense moisture transport towards the regions of heavy rainfall, (ii) providing forcing mechanisms for upward motion via reducing the static stability beneath the upper-level PV anomaly and inducing dynamical lifting ahead. Thus, the analysis that links EPEs to PV and IVT structures, separately and combined, can give a process-based understanding of the relevance of these different processes for the formation of EPEs, supported by the composites in section 6. Although the ideas behind this approach were articulated in lines 135-137 and 145-146 (the third research question), these different forcing mechanism through which Rossby wave breaking (i.e., an upper-level PV anomaly) can support heavy rainfall were only explicitly discussed in the conclusions of the original manuscript (lines 769-785). For this reason, we will include information on the mechanism of how Rossby wave breaking can bring about EPEs in the introduction to further strengthen the motivation for the approach of this study.*

2. Introduction: A vast amount of literature is covered in the introduction section. To the extent that it is quite difficult to read the paragraphs due to the very large number of references. I appreciate that the statements included in the introduction need to be supported by published literature, but is it necessary to include well over 100 references? This would be suitable for a review paper but not for a paper containing new science. The material included in the introduction needs to be more focused to identify the knowledge gaps in the published literature and motivate the study presented in the paper.

*We thank the reviewer for noticing that the many references interfere with the readability of the manuscript. We remove several references that are not vital to support the statements in the introduction, and also change the structure of citing studies, for example, in lines 94-99, we place all citations at the end of the sentence instead of after each individual region. At the same time, we also would like to motivate and clarify the reason for the many references in this manuscript. This study connects several very relevant meteorological aspects of extreme precipitation events, for example, weather systems of these events throughout the globe, other important processes such as wave breaking and intense moisture transport, and furthermore, also builds upon many previous regional studies that deserve appropriate consideration in our global analysis. We also would like to mention that, to the best of the author's knowledge, there is no set limit on the number of cited studies.*

3. Figures: Most of the figures use a rainbow colour bar. This made it difficult to view

the figures in black and white and is also very difficult for red-green colour-blind people. Please consider using a different colour bar in your figures.

*We will change the colour bars of Figures 7 and 8 by one that aids colour-blindness. For Figures 5 and 6, we realized that the use of such colour bars, that mostly use variations of one or two colours, reduce the visibility of detailed spatial patterns and make it difficult to accurately retrieve specific values of the fractions, and thus impede the interpretation of the results. Furthermore, these colour bars limit the visual aesthetics of the carefully designed figures. Please, see the accompanying Figure (Fig. 1) for an impression. For these reasons, we prefer to keep the rainbow colour bars in Figures 5 and 6 as they are, and hope this is acceptable, also with the thought in mind that many articles are published with figures using green-red colours.*

Minor comments
1. Line 8: 'Frequently' should be 'frequent' I think.

*In our understanding, "frequently" is an adverb in this sentence as it relates to the verb "cause" which then should end on "-ly". If this is incorrect, we would be more than happy to change this. Perhaps, the editing service, in case of acceptance of the manuscript, could handle this.*

2. Line 17: The statistics presented here are rather one-sided. For example, if > 90% of EPEs are associated with RWB, how many RWB events are not associated with EPE's.

*It is indeed correct that the presented analysis is "one-sided" as this study investigates the link of extreme precipitation events to Rossby wave breaking and/or intense moisture transport. This aim is accordingly articulated in research question #1 in the introduction. During the preparations of the manuscript, we considered a reversed perspective, and also analysed the fractions of e.g., Rossby wave breaking events that were linked to EPEs relative to the all Rossby wave breaking occurrences. However, this analysis appeared not to be useful as the PV and IVT structures are typically very large, especially PV structures near the Poles, and IVT structures in and near the tropics. For this reason, these fractions of PV and/or IVT structures linked to EPEs relative to all PV and/or IVT occurrences show a latitudinal gradient, with highest fractions for PV structures near the poles and highest fractions of IVT structures near the tropics. Accordingly, in our opinion, this analysis is not helpful for providing physical insights into the dynamics of EPEs. Therefore, we decided not to include this analysis in the manuscript. At present, we explore the possibility to include a short additional analysis that conveys information on this link from a reversed perspective using odds ratios.*

3. Line 21: I'm not familiar with the term 'outer tropics', is this the same region as the subtropics?

*This term refers to the poleward margins of the tropics, as opposed to the inner tropics. Following a suggestion of Reviewer #2, this entire sentence is removed from the abstract, and thus the term is removed.*

4. Line 58: What is the difference between a tropical moisture export and an atmospheric river?

*Tropical moisture export is, according to the cited studies, defined by poleward moving trajectories that originate from latitudes below 20 degrees and reach over 40 degrees within 7 days, provided that the air parcels have a moisture flux of 100 g kg$^{-1}$ m s$^{-1}$ from the start point of their (backward) trajectory calculations. Atmospheric rivers are typically defined by narrow and elongated structures of high vertically integrated vapor transport. Thus, tropical moisture exports are defined using a Lagrangian method whereas atmospheric rivers are typically*

*defined in a Eulerian framework. These methodologies used for identification of these features are briefly described in lines 81-84, and for more detailed information we refer to the cited studies. We wish not to go into more detail in the manuscript on this aspect as it is not the main focus of the study.*

5. Line 121: What does the author mean by 'the precise connection between the driving synoptic processes and the generation of extreme precipitation often remains more in the background and less understood'. This is a bit cryptic. Are they referring to microphysical processes?

*We rephrase the sentence for more clarity. We meant to refer to the synoptic-scale processes that initiate the moisture transport. Also, we hinted at the role of these synoptic processes for the forcing of ascent (dynamical lifting, reduced static stability), but this link is out of context in the present paragraph, and therefore, the rephrased sentence only refers to the large-scale circulation processes that drive the moisture transport.*

6. Line 286: What do the modulus sign represent. Is the 2 PVU definition of the tropopause appropriate in the tropics? E.g. Wilcox et al. (2011)
Wilcox, L.J., Hoskins, B.J. and Shine, K.P., 2011. A global blended tropopause based on ERA data. Part 1: Climatology. QJR Meteorol. Soc.

*The modulus sign refers to the 2 PVU definition of the dynamical tropopause in both the Northern and Southern Hemispheres. PV is, due to the planetary vorticity, negative in the Southern Hemisphere (planetary vorticity changes sign at the Equator). In our analysis, there is no need for a blended of the tropopause using thermal and dynamical properties as Rossby waves occur and break beyond the tropics, and thus, the use of the dynamical tropopause suffices. We also refer to the many cited studies in the first paragraph of section 2.3 that solely use the dynamical tropopause for the identification of wave breaking.*

7. Line 328: Given that the climatologies are so different when different thresholds are used, does this mean that there is no robust definition of this feature?

*The climatologies are indeed different when using different forms of IVT (but results are less sensitive to different thresholds). For this reason, atmospheric river studies used different IVT forms and thresholds to identify these features, as elaborated on in lines 337-349, which all have their advantages and disadvantages. In this context, it can indeed be said there is no consensus on a specific criterion for atmospheric rivers or structures of intense moisture transport. The fact that climatologies of intense moisture transport differ for the two IVT forms, motivated the choice to include figures for both IVT forms (see Figures 6 and S1 in the supplement) to give the reader as much insight as possible into the influence of the two different IVT forms.*

8. Line 330: Why is an area of 10,000 km2 used? How sensitive are the results used to this threshold?

*The study applies a minimum threshold of 100,000 $km^2$ on the IVT (and PV) structures to focus on synoptic-scale/larger-scale processes and remove the smallest-scale/local-scale features. This surface area is the equivalent of about 20 grid points of ERA-Interim at its native N128 Gaussian grid; see for an impression Fig. 4 of the manuscript the typical number of grid points of IVT structures which are well above this threshold. To clarify the motivation for applying this threshold in the study, we rephrase the sentence and mention explicitly the intention to eliminate local-scale structures and to focus at synoptic-scale structures.*

*The choice for this specific threshold is based on expert judgement that follows from detailed sensitivity analyses in a previous study on the linkage between PV and IVT structures and*

*extreme precipitation events in the Middle East (De Vries et al., 2018). These sensitivity test in this study were done for different geometrical criteria for PV streamers and IVT thresholds (see Tables 1 and 2 of De Vries et al., 2018). From these sensitivity tests we learned that stricter criteria reduce so-called false alarms (PV and/or IVT structures without occurrence of extreme precipitation), but also deteriorate hit ratios (extreme precipitation events coinciding with PV and/or IVT structures), and vice versa. In preparation of the Middle East study, we also applied different thresholds on the minimum surface area of IVT structures (e.g., 150.000 km$^2$ and 200.000 km$^2$) and found here a similar, but relatively weak effect on skill scores. For this reason, we did not include a sensitivity test for different surface area thresholds on PV and/or IVT structures in the current study. Considering the large number of choices in our current study for our objects and their matches (subjective, but based on rich expertise), we refrained from sensitivity tests on choices with relatively weak influences, but included sensitivity tests for choices with a relatively large influence; i.e., the IVT form (see the response on the previous comment, and Figures 6 and S1), the range of isentropes for PV structures (e.g., 300-350K and 310-350K; not shown) and IVT thresholds for IVT$_{pct}$ structures (200 and 0 kg m$^{-1}$ s$^{-1}$; not shown) and for IVT$_{rmean}$ structures (150 and 200 kg m$^{-1}$ s$^{-1}$; not shown). These subjective choices and their potential influence on the results are explicitly stated in the manuscript on lines 805-812 as potential limitations of the study.*

9. Line 333: Why are the IVT structures 'provided with an extended area'? How was this area chosen? Are the results sensitive to the size of this area?

*As written in the same sentence, the IVT structures are provided with an extended area of influence to facilitate the matching of remote PV structures with extreme precipitation objects. This extended area for IVT structures is only used for category 4 of the 'inclusive' and 'exclusive' matching approaches as described in the first paragraphs of sections 4.2 and 6.2. Also, as written in the same sentence, this extended area is defined in an identical way to the PV streamers, using a radius that corresponds to the objects surface area assuming a circular shape (see lines 308-319).*

*For the question on the sensitivity of the results to the size of this area we refer to the response of the previous comment nr. 8. It is also noteworthy to keep in mind that the extended surface area of the IVT structures only potentially affects the results of category 4 in Figures 6d and 12.*

10. Lines 392-400: There are 12 case studies presented in figure 4 but only 1 paragraph devoted to their description. Given that they represent a variety of time evolving processes, it seems simplistic to group them into 2 different constellations. More evidence is needed to support this binary grouping.

*The primary purpose of section 3 and Figure 4 is to demonstrate the importance of Rossby wave breaking and intense moisture transport for extreme precipitation events using catastrophic flood events from the literature. For this reason, we wish not to elaborate more on these events and their temporal evolution. The idea behind the two mechanisms that follow from these case studies is not solely based on these events, but stems from well-known previous studies as we now explicitly address in the revised introduction (see response to major comment nr. 1) and as was already mentioned in paragraph 4 of section 8. These previous studies described how PV streamers and/or cutoffs can favor extreme precipitation through inducing enhanced moisture transport and providing forcing for upward motion via reduced static stability beneath the upper-level PV anomaly and dynamical lifting ahead (Funatsu and Waugh, 2008; Schlemmer et al., 2010; Martius et al., 2013). The requested evidence to support these two groups are actually provided in the climatological analysis that follows in sections 4.2, 6.2 and 7. Briefly, to summarize, extreme precipitation events linked directly to PV and IVT structures dominate the larger part of the globe and in particular (semi)arid subtropical regions where tropical-extratropical interactions are relevant, whereas*

*remote PV structures, linked to extreme precipitation events via the IVT structures, resemble Rossby wave breaking linked to landfalling atmospheric rivers as well as events whereby wave breaking occurs too far away to provide forcing for ascent and only acts to deliver moisture supplies. For EPEs that are linked directly to PV structures, dynamical lifting (water and land) and reduced static stability (only over water) is important, whereas for EPEs that are linked to remote wave breaking and intense moisture transport, the dynamical lifting is weaker and reduced static stability absent, and PV streamers only act to induce moisture transport. Please, see further lines 686-702 and section 6.2 of the original manuscript and Figures 10 and 11 with the composites. To tell the reader upfront, in section 3, that a climatological evidence that will follow, we add a sentence at the end of section 3.*

11. Line 421: The PV streamers are 'relatively frequent' compared to what?

*We meant relatively frequent in spatial terms, compared to other regions. We rephrase the sentence for more clarity.*

12. Line 430-434: I'm afraid I didn't follow the argument regarding the IVTpct. Could this section be made clearer?

*We agree that the current description of the $IVT_{pct}$ structures was not clear. We rephrase the text in lines 430-434 as well as the text in section 2.4, lines 323-327, of the original manuscript to improve the clarity of the definition of IVT structures in the $IVT_{pct}$ form. This (sub)section, lines 430-434, describes the climatological distribution of IVT structures based on the $IVT_{pct}$ form. Since these features are defined by the $95^{th}$ percentile of IVT magnitudes at each grid point, their frequencies are by definition around 5%, taking into account that fractions can be smaller as (i) IVT structures with a surface area below 100.000 $km^2$ are removed from the selection, and (ii) the $95^{th}$ percentile of the IVT magnitude can be below the threshold of 200 kg m-1 s-1, which is used as a lower limit for the detection of the $IVT_{pct}$ structures. Especially due to the last-mentioned criterion, their frequency occurrences can be much lower over high topography and the Poles (Fig.6d) where the annual $95^{th}$ percentile IVT magnitudes are well below 200 kg $m^{-1}$ $s^{-1}$ (see Fig. 6c).*

13. Line 459: Here the statistics are rather one-sided. How many PV structures are not associated with EPEs?

*Please, see the response to comment nr. 2.*

14. Pages 15, 16 and elsewhere: Here and elsewhere the author makes use of lists to analyse the figures. While this can be a useful scientific writing tool, overuse becomes a bit repetitive after a while and I found myself skipping to the end of the list without reading each point.

*We remove the listing in the text when we refer to the regions when analysing Figures 6 and 12, in sections 4.2 and 7 (including pages 15, 16, 22 and 23). This comment was also made by reviewer #2. With these changes we hope to improve the readability of the manuscript.*

15. Line 498: Why are many regions of combined RWB and IVT positioned at the equatorward exit regions and poleward entrance regions of the extratropical storm tracks?

*Most likely because here Rossby wave breaking occurs relatively frequent in these regions and because other weather systems, such as extratropical cyclones and warm conveyor belts and recurving tropical cyclones are typically not controlling the extreme precipitation events in these regions, in contrast to the extratropical storm tracks. We rephrase the sentence for more clarity.*

16. Line 532. I'm not sure figure 7 is necessary. What does it add to the arguments already stated in the paper?

*Figure 7 shows that, taking the extreme precipitation volumes as starting point, the depth of the Rossby wave breaking and the intensity of the moisture transport increase with larger precipitation volumes (see lines 524-544). These findings are highly relevant as this link has not been shown in any previous study, to the best of the author's knowledge. In short, we show here for the first time that the deeper (i.e., stronger) the wave breaking and the more intense the moisture transport, the more severe the extreme precipitation. Figure 8 and lines 546-559 make this point from a reversed perspective, taking the depth of the wave breaking and intensity of the moisture transport as a starting point, and linking these characteristics to the average precipitation volumes, thus being a complementary analysis.*

17. Line 634 and elsewhere: Here the author refers to 'category 1' and later to categories 2-5. Perhaps I missed this, but where are these different categories defined? Are these the categories in the synthesis section? If so, they shouldn't be referred to in section 6.

*These five categories are actually explicitly described in the first paragraph of section 6.2, lines 595-601, of the original manuscript. This 'exclusive' matching approach, as opposed to the 'inclusive' matching approach used in section 4.2, are used in both sections 6.2 and 7. In order to make the reader in an early stage aware of the two different matching techniques, the manuscript contains a brief description of these two matching approaches in section 2.5, supported by Table 1, stating the differences and in which sections which method is used. Also, prior to the description of the relevant results we explicitly describe these category definitions, see lines 436-443 for the inclusive matching, and as mentioned, lines 595-601 for the exclusive matching.*

18. Line 740: What do the numbers in brackets mean? >90% of what?

*These numbers refer to the percentages of EPEs that are linked to Rossby wave breaking and/or intense moisture transport. For more clarity, we replace these fractions from the end of the sentences to directly after the "EPE" phrase to which the fraction values relate.*

***Authors response to the comments of Reviewer 2***

The analysis builds on detailed and careful analysis of the data, on a large number of variables. I find figures excellent and well-illustrating results. However, the text sometimes is a bit too dense and difficult to follow due to overuse of bullet (numbered) lists, which breaks the flow of the text.
My suggestion, for example, is to reduce the abstract which I found far too long and detailed in the description of the results. See the attached pdf for detailed comments also on the abstract.

*We removed the detailed description on the five exclusive categories in the abstract and added instead a short sentence reflecting the essence; that five categories with different combinations of Rossby wave breaking and intense moisture transport can reflect a large range of EPE related weather systems across various climate zones.*

*In addition, also consistent with the comments of reviewer #1, we removed the use of numbered lists to improve the readability of the text in sections 4.2 and 7 where results are described for different regions (pages 15-16, 22-23). At other locations, for example the description of the categories for matching EPEs with PV and/or IVT structures, we keep the*

*numbered lists as they refer directly to the category numbers mentioned in Table 1 and at other locations in the text.*

Major comments:

Seasonality is not explicitely considered in this analysis. I wonder how the repartition of these five categories in the space could change in the seasons.

*This is indeed a good and very interesting question. We also thought of addressing this point during preparation of the manuscript. However, since the large amount of material and the currently already very long manuscript, we decided not to address this aspect in the manuscript. Central focus of the manuscript is to show the importance of wave breaking and moisture transport for extreme precipitation events using two very powerful meteorological variables; PV and IVT. However, we plan to address the seasonality in a follow-up study with a specific focus on the partition in five categories (Figure 12). It would be very interesting to see how the EPE fractions attributed to the different categories will change with the seasons. One matter that may complicate the analysis here is the definition of the extreme precipitation events that is currently based on annual percentile-based thresholds. A seasonal analysis may also require percentile-based thresholds with reference to the specific seasons. In short, a seasonal analysis may become quite complex, and as the current analysis is already very complex and comprehensive, we prefer to address the seasonal aspect in a separate study. We include a sentence in the conclusions that acknowledges that the seasonality of this analysis is not included and states the intention for this in a follow-up study.*

Lines 323-325: The definition of IVTpct is not totally clear: Do you subtract to the full IVT the 95_ percentile of the annual distribution at each grid point ? If yes, state this.

*Not precisely. We define the IVT structures for the $IVT_{pct}$ form as follows; (1) we calculate at each grid point the $95^{th}$ percentile value of the IVT magnitude from 6-hourly fields throughout the period under consideration (1979-2018). Then, for detection of $IVT_{pct}$ structures, we only consider the grid points where IVT values exceed this $95^{th}$ percentile, and (2), we use a threshold of 200 kg $m^{-1}$ $s^{-1}$ as a lower limit on these IVT fields (note that these are not subtracted IVT fields, but simply only consider IVT values that exceed the $95^{th}$ percentiles). This means that $IVT_{pct}$ structures include grid points where IVT magnitude values exceed the annual $95^{th}$ percentile and exceed a lower limit of 200 kg $m^{-1}$ $s^{-1}$. Technically, we first placed IVT values in 6-hourly fields to zero values when they didn't exceed the annual $95^{th}$ percentiles, and then defined the IVT structures by IVT values that exceed 200 kg $m^{-1}$ $s^{-1}$. We rephrase the text for more clarity, and also correct a minor mistake in the text; "of daily mean IVT" in line 323 is incorrect; the annual $95^{th}$ percentile values are computed based on 6-hourly fields. We thank the reviewer for commenting on the unclarity of the $IVT_{pct}$ definition in the text.*

Lines 585-594: The difference between subtropical and extratropical circulation around EPE it is not very clear. Perhaps commenting more on the anomaly panels, one could see better the difference in baroclinicity with a more pronounced back-tilted cyclones in the extratropics.

*We rephrase a part of the text and refer here explicitly to stronger back-tilted cyclones in the extratropics based on the anomaly panels.*

Lines 710-713: I found weird the explanation of white areas in Fig.12 . It is stated that these areas coincides with those where cyclones and warm-conveyor belts have the highest relevance. Correct. But the same areas should also display substantial IVT values, like over the western Atlantic. If you have cyclones and WCB, usually there are also high IVT values (at least), in addition to upper level waves. I wonder if this lack of classification in this region it is not due to the definition of IVTpct objects which set to

a too high treshold for this region in which very frequently IVT is very high. This is not clear to me and it deserves a clarification.

*It is indeed correct that $IVT_{pct}$ structures are not very dominant in relation to EPEs in these regions due to their definition. For comparison, the $IVT_{rmean}$ structures are stronger associated with EPEs than $IVT_{pct}$ structures in these 'white' regions from Figure 12, see Figures 6b and S1b from the original manuscript. The point we want to make here is that different combinations of intense moisture transport (>95$^{th}$ percentile) and/or wave breaking are of reduced importance in these regions as EPE formation is primarily controlled here by cyclones and/or warm conveyor belts that, admittedly, may have high IVT values, but not the intense moisture transport in which we are interested. Instead, different combinations of wave breaking and intense moisture transport show up as very important for EPE formation in many other regions such as the (semi)arid subtropics. In the revised manuscript we will include a statement that makes the reader aware that warm conveyor belts and extratropical cyclones usually go along with IVT values but not necessarily the unusually high IVT values (>95$^{th}$ percentile) as used for our definition of intense moisture transport.*

In addition to these few major points I have noted few technical corrections on the annotated pdf attached below.

***Author's response to the comments of Reviewer 2 as provided in the pdf***

*1. We removed the long and detailed sentence in the abstract and replaced it by a short sentence, see also the response above.*

*2. the typo "discussing" on line 445 is corrected.*

*3. we revise the sentence on lines 462-464 and explicitly say "… at some locations equatorward …".*

*4. Figure 3 and caption. As far as I understand the comment, "daily" in the figure is spelled correctly which is visible when zooming in. For more clarity, we will replace "IVT structures" by "$IVT_{pct}$ structures" in the caption.*

***References***

*De Vries A. J., Ouwersloot, H. G., Feldstein, S. B., Riemer, M., El Kenawy, A. M., McCabe, M. F., and Lelieveld, J.: Identification of tropical-extratropical interactions and extreme precipitation events in the Middle East based on potential vorticity and moisture transport, J. Geophys. Res. Atmos., 123, 861-881, 2018.*

*Funatsu, B. M. and Waugh, D. W.: Connections between potential vorticity intrusions and convection in the Eastern Tropical Pacific, J. Atmos. Sci., 65, 987–1002, 2008.*

*Schlemmer, L., O. Martius, M. Sprenger, C. Schwierz, and A. Twitchett (2010), Disentangling the forcing mechanisms of a heavy precipitation event along the alpine south side using potential vorticity inversion, Mon. Weather Rev., 138, 2336–2353.*

*Martius, O., Sodemann, H., Joos, H., Pfahl, S.,Winschall, A., Croci-Maspoli, M., Graf, M., Madonna, E., Mueller, B., Schemm, S., Sedlácek, J., Sprenger, M., and Wernli, H.: The role of upper-level dynamics and surface processes for the Pakistan flood of July 2010, Q. J. Royal Meteorol. Soc., 139, 1780–1797, https://doi.org/10.1002/qj.2082, 2013.*

---

## Author Response (AR1)

**Response to referee comments on manuscript WCD-2020-44**

**Title: "A global climatological perspective on the importance of Rossby wave breaking and intense moisture transport for extreme precipitation events"**

**Author: A. J. de Vries**

The author would like to thank the two anonymous reviewers for their helpful and constructive comments and the editor for handling the peer review process. Each of the reviewers' comments was carefully considered for improving the manuscript. Most of the comments led to changes in the text. The most substantial changes are

- Replacement of a very long detailed sentence in the abstract by a concise one;
- Include a motivation in the introduction for the study of the individual and combined importance of wave breaking and moisture transport by describing the mechanisms of how PV streamers can contribute to extreme precipitation formation;
- Removal of numbered lists in the text where different regions are discussed;
- Removal of a few references in the introduction and replacing others at one location in the sentence instead of scattered over different parts;
- Rephrasing the definition of the $IVT_{pct}$ structures;
- Inclusion of odds ratios of extreme precipitation events linked to the synoptic-scale processes to provide a complementary analysis on the importance of the two processes for extreme precipitation events (section 4.3 and Figure 7 of the revised manuscript).

In addition to the suggestion from the referees, the author also implemented another change, inspired by discussions with a colleague. The first version of the manuscript used a simple statistical significance test that is now replaced by the false discovery rate test, which is according to many scientists seen as the new standard of statistical significance testing in atmospheric science (Wilks, 2016). Also, the author applied several additional minor changes in the text with the sole purpose to improve the quality of the manuscript. These changes are all shown by tracked changes in the manuscript. Finally, a brief clarification on the formulas in the manuscript; I removed those for the omega equation as these are also shown in the cited studies, while I rewrote the other ones because of a layout issue (they show up as tracked changes, but are identical to those in the original manuscript).

The text below (in blue) responds to each of the reviewer's comments and indicates what changes have been made in the text, answers to questions and concerns raised, and at few locations, provides arguments why we prefer to refrain from specific suggested revisions.

***Authors response to the comments of Referee 1***

Major comments
1. Line 11, line 129 and elsewhere: There is little motivation for this work in the abstract and introduction. I agree that previous studies have considered RWB and IVT separately, but what is the motivation for considering them together? Why is this a good idea? I assume that the author regards the fact that ascent (or cooling) is necessary to convert water vapour into liquid water and further precipitation is unnecessary to include. However, given the number of studies that use IVT as a proxy for precipitation and EPE I think this is a key point to include to motivate the work.

*Previous studies have mostly investigated the role of Rossby wave breaking and intense moisture transport for the formation of extreme precipitation events separately. Also, previous studies that addressed the linkage between wave breaking and EPEs were confined to specific regions. This study brings for the first time together these two synoptic-scale processes as drivers of EPEs at the global scale. This motivation is explicitly articulated in the original version of the manuscript, more specifically, the lines 9-12 on page 1 in the abstract, and lines 119-129 and 131-137 in the introduction. An important reason to combine these two (thermo)dynamic processes is that Rossby wave breaking can initiate extreme precipitation events through (i) steering intense moisture transport towards the regions of heavy rainfall, (ii) providing forcing mechanisms for upward motion via reducing the static stability beneath the upper-level PV anomaly and inducing dynamical lifting ahead. Thus, the analysis that links EPEs to PV and IVT structures, separately and combined, can give process-based insights into the relevance of these different processes for the formation of EPEs, supported by the composites in section 6. Although the ideas behind this approach were articulated in lines 135-137 and 145-146 of the original manuscript (the third research question), these different forcing mechanism through which Rossby wave breaking (i.e., an upper-level PV anomaly) can support heavy rainfall were only explicitly discussed in the conclusions of the original manuscript (lines 769-785). For this reason, we will include information on the mechanism of how Rossby wave breaking can bring about EPEs in the introduction to further strengthen the motivation for the approach of this study. In the introduction, we add the following sentence in the paragraph that introduces Rossby wave breaking as a driver of extreme precipitation events (lines 86-88 of the revised manuscript);*

*"Such PV streamers and cutoffs can favour EPE formation through inducing enhanced moisture transport and providing forcing for ascent via reduced static stability beneath the upper tropospheric positive PV anomaly and dynamical lifting ahead (Funatsu and Waugh, 2008; Schlemmer et al., 2010)."*

2. Introduction: A vast amount of literature is covered in the introduction section. To the extent that it is quite difficult to read the paragraphs due to the very large number of references. I appreciate that the statements included in the introduction need to be supported by published literature, but is it necessary to include well over 100 references? This would be suitable for a review paper but not for a paper containing new science. The material included in the introduction needs to be more focused to identify the knowledge gaps in the published literature and motivate the study presented in the paper.

*We thank the reviewer for noticing that the many references interfere with the readability of the manuscript. We remove several references that are not vital to support the statements in the introduction, and also change the structure of citing studies. For example, in lines 94-99 of the original manuscript (lines 91-92 of the revised manuscript), we place all citations at the end of the sentence instead of after each individual region. At the same time, we also would like to motivate and clarify the reason for the many references in this manuscript. This study*

*connects several very relevant meteorological aspects of extreme precipitation events, for example, several weather systems of these events across various climate zones, including important processes such as wave breaking and moisture transport. Furthermore, this study also builds upon many previous regional studies that deserve appropriate consideration in our global analysis. We also would like to mention that, to the best of the author's knowledge, there is no set limit on the number of cited studies.*

3. Figures: Most of the figures use a rainbow colour bar. This made it difficult to view the figures in black and white and is also very difficult for red-green colour-blind people. Please consider using a different colour bar in your figures.

*We change the colour bars of Figures 7 and 8 of the original manuscript (Figures 8 and 9 of the revised manuscript) by one that aids colour-blindness. For Figures 5 and 6, we realized that the use of such colour bars, that mostly contain variations of one or two colours, reduce the visibility of detailed spatial patterns and make it difficult to accurately retrieve specific values of the fractions, and thus hinder the interpretation of the results. Furthermore, these colour bars limit the visual aesthetics of the carefully designed figures. Please, see the below Figure R1 for an impression. For these reasons, we prefer to keep the rainbow colour bars in these two figures, and hope this is acceptable, also with the thought in mind that many articles are published with figures using green-red colours.*

[Figure]

*Figure R1: Identical to Figure 6 of the original and revised manuscript with a colour-blindness friendly colour bar.*

Minor comments
1. Line 8: 'Frequently' should be 'frequent' I think.

*In our understanding, "frequently" is an adverb in this sentence as it relates to the verb "cause" which then should end on "-ly". If this is incorrect, we would be more than happy to change this. Perhaps, the editing service, in case of acceptance of the manuscript, could handle this. We replace the adverb before the verb "…frequently cause…".*

2. Line 17: The statistics presented here are rather one-sided. For example, if > 90% of EPEs are associated with RWB, how many RWB events are not associated with

EPE's.

*It is indeed correct that the presented analysis in the original manuscript is "one-sided" as this study investigates the link of extreme precipitation events to Rossby wave breaking and/or intense moisture transport. This aim is accordingly articulated in research question #1 in the introduction. During the preparations of the manuscript, we considered a reversed perspective, and also analysed the fractions of e.g., Rossby wave breaking events that were linked to EPEs relative to the all Rossby wave breaking occurrences. However, this analysis appeared not to be useful as the PV and IVT structures are typically very large, especially PV structures near the Poles, and IVT structures in and near the tropics. For this reason, these fractions of PV and/or IVT structures linked to EPEs relative to all PV and/or IVT occurrences show a latitudinal gradient, with highest fractions for PV structures near the poles and highest fractions of IVT structures near the tropics, and lack any interesting regional features (see Figure R2 below). Accordingly, this analysis is not considered helpful for providing physical insights into the dynamics of EPEs. Therefore, we decided not to include this analysis in the manuscript.*

[Figure]

*Figure R2: Fractions of (top) PV structures and (below) IVT structures that match with EPEs relative to all PV structure and IVT structure occurrences, respectively.*

*To provide a complementary analysis on the importance of wave breaking and moisture transport for the occurrence of EPEs, we include a new analysis based on odds ratios. We compute the odds of EPE linked to PV and/or IVT structures relative to odds under assumed independence of the EPEs and the PV and/or IVT structures, based on average fractions from the 1000 Monte Carlo sets (see section 2.5 of the revised manuscript). This analysis presents insight into what extent these two individual and combined processes are relevant for the formation of EPEs. This analysis is included a new section, section 4.3, and supported by the new Figure 7 of the revised manuscript. This new analysis is also briefly summarized in one sentence in the abstract (lines 22-24 of the revised manuscript) and two sentences in the conclusions (lines 808-812 of the revised manuscript).*

3. Line 21: I'm not familiar with the term 'outer tropics', is this the same region as the

subtropics?

*This term refers to the poleward margins of the tropics, as opposed to the inner tropics. Following a suggestion of Reviewer #2, this entire sentence is removed from the abstract, and thus the term is removed here.*

4. Line 58: What is the difference between a tropical moisture export and an atmospheric river?

*Tropical moisture export is, according to the cited studies, defined by poleward moving trajectories that originate from latitudes below 20 degrees and reach over 40 degrees within 7 days, provided that the air parcels have a moisture flux of 100 g kg$^{-1}$ m s$^{-1}$ from the start point of their (backward) trajectory calculations. Atmospheric rivers are typically defined by narrow and elongated structures of high vertically integrated vapor transport. Thus, tropical moisture exports are defined using a Lagrangian method whereas atmospheric rivers are typically defined in a Eulerian framework. These methodologies used for identification of these features are briefly described in lines 81-84 of the original manuscript, and for more detailed information we refer to the cited studies. We wish not to go into more detail on this aspect in the manuscript as it is not the main focus of the study.*

5. Line 121: What does the author mean by 'the precise connection between the driving synoptic processes and the generation of extreme precipitation often remains more in the background and less understood'. This is a bit cryptic. Are they referring to microphysical processes?

*We rephrase the sentence for more clarity. We meant to refer to the synoptic-scale processes that initiate the moisture transport. Also, we hinted at the role of these synoptic processes for the forcing of ascent (dynamical lifting, reduced static stability), but this link is out of context in the present paragraph, and therefore, the rephrased sentence only refers to the large-scale circulation processes that drive the moisture transport (lines 113-115 of the revised manuscript);*

*"Although the focus on intense moisture transport and the use of IVT is very prominent in studies of extreme precipitation, investigations of the large-scale circulation processes that drive the moisture transport often remain more in the background and less understood."*

6. Line 286: What do the modulus sign represent. Is the 2 PVU definition of the tropopause appropriate in the tropics? E.g. Wilcox et al. (2011)
Wilcox, L.J., Hoskins, B.J. and Shine, K.P., 2011. A global blended tropopause based on ERA data. Part 1: Climatology. QJR Meteorol. Soc.

*The modulus sign referred to the 2 PVU definition of the dynamical tropopause in both the Northern and Southern Hemispheres. PV is, due to the planetary vorticity, negative in the Southern Hemisphere (planetary vorticity changes sign at the Equator). In our analysis, there is no need for a blended of the tropopause using thermal and dynamical properties as Rossby waves occur and break beyond the tropics, and thus, the use of the dynamical tropopause suffices. We also refer to the many cited studies in the first paragraph of section 2.3 that solely use the dynamical tropopause for the identification of wave breaking.*

*For more clarity, we remove the modulus sign in the text (but add it in the caption of Figure 3 for correctness), and include a sentence that explicitly articulates the generally negative PV values in the Southern Hemisphere, and the statement that we only use absolute PV values in this study (lines 270-271 of the revised manuscript):*

*"Note that PV has generally negative values in the Southern Hemisphere, and that we will use absolute PV values for consistency in both the text and figures."*

7. Line 328: Given that the climatologies are so different when different thresholds are used, does this mean that there is no robust definition of this feature?

*The climatologies are indeed different when using different forms of IVT (but results are less sensitive to different thresholds). For this reason, atmospheric river studies used different IVT forms and thresholds to identify these features, as elaborated on in lines 337-349 of the original manuscript, each with their advantages and disadvantages. In this context, it can indeed be said there is no consensus on a specific criterion for atmospheric rivers or structures of intense moisture transport. The fact that climatologies of intense moisture transport differ for the two IVT forms, motivated the choice to include figures for both IVT forms (see Figures 6 and S2 in the supplement of the revised manuscript) to give the reader as much insight as possible into the influence of the two different IVT forms.*

8. Line 330: Why is an area of 10,000 km2 used? How sensitive are the results used to this threshold?

*The study applies a minimum threshold of 100,000 $km^2$ for the IVT (and PV) structures to focus on synoptic-scale/larger-scale processes and remove the smallest-scale/local-scale features. This surface area is the equivalent of about 20 grid points of ERA-Interim at its native N128 Gaussian grid; see for an impression Fig. 4 of the manuscript the typical number of grid points of IVT structures which are well above this threshold. To clarify the motivation for applying this threshold in the study, we rephrase the sentence and mention explicitly the intention to eliminate local-scale structures and to focus at synoptic-scale structures (lines 314-315 of the revised manuscript);*

*"…with the aim to eliminate the local-scale features and to focus on synoptic-scale processes."*

*The choice for this specific threshold is based on expert judgement that follows from detailed sensitivity analyses in a previous study on the linkage between PV and IVT structures and extreme precipitation events in the Middle East (De Vries et al., 2018). These sensitivity test in this study were done for different geometrical criteria for PV streamers and IVT thresholds (see Tables 1 and 2 of De Vries et al., 2018). From these sensitivity tests we learned that stricter criteria reduce so-called false alarms (PV and/or IVT structures without occurrence of extreme precipitation), but also deteriorate hit ratios (extreme precipitation events coinciding with PV and/or IVT structures), and vice versa. In preparation of the Middle East study, we also applied different thresholds on the minimum surface area of IVT structures (e.g., 150.000 $km^2$ and 200.000 $km^2$) and found here a similar, but relatively weak effect on skill scores. For this reason, we did not include a sensitivity test for different surface area thresholds on PV and/or IVT structures in the current study. Considering the large number of choices in our current study for our objects and their matches (subjective, but based on rich expertise), we refrained from sensitivity tests on choices with relatively weak influences, but included sensitivity tests for choices with a relatively large influence; i.e., the IVT form (see the response on the previous comment, and Figures 6 and S2 of the revised manuscript), the range of isentropes for PV structures (e.g., 300-350K and 310-350K; not shown) and IVT thresholds for $IVT_{pct}$ structures (200 and 0 kg $m^{-1}$ $s^{-1}$; not shown) and for $IVT_{filt}$ structures (150 and 200 kg $m^{-1}$ $s^{-1}$; not shown). These subjective choices and their potential influence on the results are explicitly stated in the original manuscript on lines 805-812 as potential limitations of the study.*

9. Line 333: Why are the IVT structures 'provided with an extended area'? How was this area chosen? Are the results sensitive to the size of this area?

*As written in the same sentence, the IVT structures are provided with an extended area of influence to facilitate the matching of remote PV structures with extreme precipitation objects. This extended area for IVT structures is only used for category 4 of the 'inclusive' and 'exclusive' matching approaches as described in the first paragraphs of sections 4.2 and 6.2. Also, as written in the same sentence, this extended area is defined in an identical way to the PV streamers, using a radius that is proportional to the objects surface area assuming a circular shape (see lines 308-319 of the original manuscript).*

*For the question on the sensitivity of the results to the size of this area we refer to the response of the previous comment nr. 8. It is also noteworthy to keep in mind that the extended surface area of the IVT structures only potentially affects the results of category 4.*

10. Lines 392-400: There are 12 case studies presented in figure 4 but only 1 paragraph devoted to their description. Given that they represent a variety of time evolving processes, it seems simplistic to group them into 2 different constellations. More evidence is needed to support this binary grouping.

*The primary purpose of section 3 and Figure 4 is to demonstrate the importance of Rossby wave breaking and intense moisture transport for extreme precipitation events using catastrophic flood events from the literature. For this reason, we wish not to elaborate more on these events and their temporal evolution. The idea behind the two mechanisms that follows from these case studies is not solely based on these events, but stems from concepts in well-known previous studies as we now explicitly address in the revised introduction (see response to major comment nr. 1) and as was already mentioned in paragraph 4 of section 8. These previous studies described how PV streamers and/or cutoffs can favour extreme precipitation through inducing enhanced moisture transport and providing forcing for upward motion via reduced static stability beneath the upper-level PV anomaly and dynamical lifting ahead (Funatsu and Waugh, 2008; Schlemmer et al., 2010; Martius et al., 2013). The requested evidence to support these two groups are actually provided in the climatological analysis that follows in sections 6.2 and 7. Briefly, to summarize, extreme precipitation events linked directly to PV and IVT structures dominate the larger part of the globe and in particular (semi)arid subtropical regions where tropical-extratropical interactions are relevant, whereas remote PV structures, linked to extreme precipitation events via the IVT structures, resemble Rossby wave breaking linked to landfalling atmospheric rivers as well as events whereby wave breaking occurs too far away to provide forcing for ascent and only acts to deliver moisture supplies. For EPEs that are linked directly to PV structures, dynamical lifting (water and land) and reduced static stability (only over water) is important, whereas for EPEs that are linked to remote wave breaking and intense moisture transport, the dynamical lifting is weaker and reduced static stability absent, and PV streamers only act to steer moisture transport. Please, see sections 6.2 and 7 and Figures 10 and 11 of the original manuscript. To tell the reader upfront, in section 3, that climatological evidence for these two mechanisms will follow, we add a sentence at the end of section 3;*

*"Sections 6.2 and 7 will elaborate on this concept and provide climatological evidence for these two different forcing mechanisms."*

11. Line 421: The PV streamers are 'relatively frequent' compared to what?

*We meant relatively frequent in spatial terms, compared to other regions. We rephrase the sentence for more clarity (Line 434 of the revised manuscript);*

*"…PV streamers occur most frequently …"*

12. Line 430-434: I'm afraid I didn't follow the argument regarding the IVTpct. Could this section be made clearer?

*We agree that the description of the $IVT_{pct}$ structures in the original manuscript was not clear, and thank the reviewer for pointing this out. This (sub)section, lines 430-434, describes the climatological distribution of IVT structures based on the $IVT_{pct}$ form. Since these features are defined by the 95th percentile of IVT magnitudes at each grid point, their frequencies are by definition around 5%, taking into account that fractions can be smaller as (i) IVT structures with a surface area below 100.000 $km^2$ are removed from the selection, and (ii) the 95th percentile of the IVT magnitude can be below the threshold of 200 kg m-1 s-1, which is used as a lower limit for the detection of the $IVT_{pct}$ structures. Especially due to the last-mentioned criterion, their frequency occurrences can be much lower over high topography and the poles (Fig.6d) where the annual 95th percentile IVT magnitudes are well below 200 kg $m^{-1}$ $s^{-1}$ (see Fig. 6c).*

*We rephrase the text in lines 430-434 (lines 444-449 of the revised manuscript);*

*"Frequencies of $IVT_{pct}$ structures are by design near 5 % as these structures are defined by the annual 95th percentiles of IVT magnitudes (Fig. 5d). Note that these fractions can be lower than 5% as small IVT structures are removed from the selection, and the 95th percentile can be below the lower limit of 200 kg $m^{-1}$ $s^{-1}$. Especially due to the latter reason, $IVT_{pct}$ structure frequencies remain very low over polar regions and elevated topography, i.e., the Rocky Mountains, the Andes, the Himalaya's, the Ethiopian Highlands, Greenland and Antarctica (Fig. 5c,d).".*

*and revise the text and add to the first paragraph of section 2.4 to improve the clarity of the definition of IVT structures in the $IVT_{pct}$ form (lines 305-309 of the revised manuscript);*

*"…we use the following two IVT forms and criteria: (1) a static threshold of 150 kg $m^{-1}$ $s^{-1}$ on filtered IVT whereby the 91-day running mean is subtracted from the full IVT ($IVT_{filt}$), and (2) exceedances of the annual 95th percentile of 6-hourly full IVT over the period of 1979-2018 and a minimum threshold of 200 kg $m^{-1}$ $s^{-1}$ ($IVT_{pct}$). Regarding the latter form, $IVT_{pct}$ structures thus consists of grid points where full IVT values both exceed the annual 95th percentile and a lower limit of 200 kg $m^{-1}$ $s^{-1}$.".*

13. Line 459: Here the statistics are rather one-sided. How many PV structures are not associated with EPEs?

*Please, see the response to comment nr. 2.*

14. Pages 15, 16 and elsewhere: Here and elsewhere the author makes use of lists to analyse the figures. While this can be a useful scientific writing tool, overuse becomes a bit repetitive after a while and I found myself skipping to the end of the list without reading each point.

*We remove the listing (i.e. numbers) in the text when we refer to the regions when analysing Figures 5, 6 and 12 (Fig 13 of the revised manuscript), in sections 4.2 and 7 (including those on pages 14, 15, 16, 22 and 23 of the original manuscript). This comment was also made by reviewer #2. With these changes we hope to have improved the readability of the manuscript.*

15. Line 498: Why are many regions of combined RWB and IVT positioned at the equatorward exit regions and poleward entrance regions of the extratropical storm tracks?

*Most likely because here Rossby wave breaking occurs relatively frequent in these regions and because other weather systems, such as extratropical cyclones, warm conveyor belts and recurving tropical cyclones are relatively less relevant to the extreme precipitation events in*

*these regions, in contrast to the extratropical storm tracks. We rephrase the sentence for more clarity and add a citation to a study that point to the relevance of cyclonic wave breaking for precipitation extremes over high latitude regions and anticyclonic wave breaking for precipitation extremes in lower latitudes;*

*"Most of these subtropical (extratropical) regions in the Northern Hemisphere are positioned near the equatorward end (poleward start) of the Pacific and Atlantic storm tracks where anticyclonic (cyclonic) wave breaking has been suggested to favour precipitation extremes (Röthlisberger et al., 2016)."*

16. Line 532. I'm not sure figure 7 is necessary. What does it add to the arguments already stated in the paper?

*Figure 7 (Figure 8 of the revised manuscript) shows that, taking the extreme precipitation volumes as starting point, the depth of the Rossby wave breaking and the intensity of the moisture transport increases with larger precipitation volumes (see lines 524-544 of the original manuscript). These findings are highly relevant as this link has not been shown in any previous study, to the best of the author's knowledge. In short, we show here for the first time that the deeper (i.e., stronger) the wave breaking and the more intense the moisture transport, the more severe the extreme precipitation. Figure 8 and lines 546-559 of the original manuscript make this point from a reversed perspective, taking the depth of the wave breaking and intensity of the moisture transport as a starting point, and linking these characteristics to the average precipitation volumes, thus being a complementary analysis.*

17. Line 634 and elsewhere: Here the author refers to 'category 1' and later to categories 2-5. Perhaps I missed this, but where are these different categories defined? Are these the categories in the synthesis section? If so, they shouldn't be referred to in section 6.

*These five categories are actually explicitly described in the first paragraph of section 6.2, lines 595-601, of the original manuscript. This 'exclusive' matching approach, as opposed to the 'inclusive' matching approach used in section 4.2, are used in both sections 6.2 and 7. In order to make the reader in an early stage aware of the two different matching techniques, the manuscript contains a description of these two matching approaches in the first paragraph of section 2.5, supported by Table 1, stating the differences and in which sections which method is used. Also, prior to the description of the relevant results we explicitly describe these category definitions, see lines 436-443 (of the original manuscript) for the inclusive matching, and as mentioned, lines 595-601 (of the original manuscript) for the exclusive matching.*

18. Line 740: What do the numbers in brackets mean? >90% of what?

*These numbers refer to the percentages of EPEs that are linked to Rossby wave breaking and/or intense moisture transport. For more clarity, we replace these fractions from the end of the sentences to directly after the "EPE" phrase to which the fraction values relate.*

**Authors response to the comments of Reviewer 2**

The analysis builds on detailed and careful analysis of the data, on a large number of variables. I find figures excellent and well-illustrating results. However, the text sometimes is a bit too dense and difficult to follow due to overuse of bullet (numbered) lists, which breaks the flow of the text.
My suggestion, for example, is to reduce the abstract which I found far too long and detailed in the description of the results. See the attached pdf for detailed comments also on the abstract.

*We thank the reviewer for the positive feedback on the study.*

*We removed the detailed description on the five exclusive categories in the abstract and added instead a short sentence reflecting the essence (lines 21-22 of the revised manuscript);*

*"A detailed analysis shows that five categories with different combinations of wave breaking and intense moisture transport can reflect a large range of EPE-related weather systems across various climate zones".*

*In addition, also consistent with the comments of reviewer #1, we removed the use of numbered lists to improve the readability of the text in sections 4.1, 4.2 and 7 where results are described for different regions (pages 14-16, 22-23 of the original manuscript). At other locations, for example the description of the categories for matching EPEs with PV and/or IVT structures, we keep the numbered lists as they refer directly to the category numbers mentioned in Table 1 and at other locations in the text.*

Major comments:

Seasonality is not explicitely considered in this analysis. I wonder how the repartition of these five categories in the space could change in the seasons.

*This is indeed a good and very interesting question. We also thought of addressing this point during preparation of the manuscript. However, since the large amount of material and the currently already very long manuscript, we decided not to address this aspect in the manuscript. Central focus of the manuscript is to show the importance of wave breaking and moisture transport for extreme precipitation events using two very powerful meteorological variables; PV and IVT. However, we plan to address the seasonality in a follow-up study with a specific focus on the partition in five categories (Figure 13 of the revised manuscript). It would be very interesting to see how the EPE fractions attributed to the different categories will change with the seasons. One matter that may complicate the analysis here is the definition of the extreme precipitation events that is currently based on annual percentile-based thresholds. A seasonal analysis may also require percentile-based thresholds with reference to the specific seasons. In short, a seasonal analysis may become quite complex, and as the current analysis is already very complex and comprehensive, we prefer to address the seasonal aspect in a separate study. We include a sentence in the conclusions that acknowledges that the seasonality of this analysis is not included and states the intention for this in a follow-up study (lines 865-866 of the revised manuscript);*

*"Seasonal aspects of this new classification are not addressed in this work and are planned as a part of a follow-up study."*

Lines 323-325: The definition of IVTpct is not totally clear: Do you subtract to the full IVT the 95_ percentile of the annual distribution at each grid point ? If yes, state this.

*Not precisely. We define the IVT structures for the $IVT_{pct}$ form as follows; (1) we calculate at each grid point the $95^{th}$ percentile value of the IVT magnitude from 6-hourly fields throughout the period under consideration (1979-2018). Then, for detection of $IVT_{pct}$ structures, we only consider the grid points where IVT values exceed this $95^{th}$ percentile, and (2), we use a threshold of 200 kg m$^{-1}$ s$^{-1}$ as a lower limit on these IVT fields (note that these are not subtracted IVT fields, but full IVT fields). This means that $IVT_{pct}$ structures include grid points where IVT magnitude values exceed both the annual $95^{th}$ percentile and a lower limit of 200 kg m$^{-1}$ s$^{-1}$. Technically, we first placed IVT values in 6-hourly fields to zero values when they didn't exceed the annual $95^{th}$ percentiles, and then defined the IVT structures by IVT values that exceed 200 kg m$^{-1}$ s$^{-1}$. We rephrase the text for more clarity, and also correct a minor mistake in the text; "of daily mean IVT" in line 323 of the original manuscript is incorrect; the*

*annual 95th percentile values are computed based on 6-hourly fields. We thank the reviewer for commenting on the unclarity of the $IVT_{pct}$ definition in the text.*

*Text revisions (first paragraph of section 2.4 of the revised manuscript, lines 305-309); "……we use the following two IVT forms and criteria: (1) a static threshold of 150 kg m$^{-1}$ s$^{-1}$ on filtered IVT whereby the 91-day running mean is subtracted from the full IVT ($IVT_{filt}$), and (2) exceedances of the annual 95$^{th}$ percentile of 6-hourly full IVT over the period of 1979-2018 and a minimum threshold of 200 kg m$^{-1}$ s$^{-1}$ ($IVT_{pct}$). Regarding the latter form, $IVT_{pct}$ structures thus consists of grid points where full IVT values both exceed the annual 95$^{th}$ percentile and a lower limit of 200 kg m$^{-1}$ s$^{-1}$.".*

Lines 585-594: The difference between subtropical and extratropical circulation around EPE it is not very clear. Perhaps commenting more on the anomaly panels, one could see better the difference in baroclinicity with a more pronounced back-tilted cyclones in the extratropics.

*We rephrase a part of the text and refer here explicitly to stronger back-tilted cyclones in the extratropics based on the anomaly panels (line 646 of the revised manuscript);*

*"…a more pronounced backward tilting cyclonic structure with height (Fig. 10h,i,k,l) and multiple…".*

Lines 710-713: I found weird the explanation of white areas in Fig.12 . It is stated that these areas coincides with those where cyclones and warm-conveyor belts have the highest relevance. Correct. But the same areas should also display substantial IVT values, like over the western Atlantic. If you have cyclones and WCB, usually there are also high IVT values (at least), in addition to upper level waves. I wonder if this lack of classification in this region it is not due to the definition of IVTpct objects which set to a too high treshold for this region in which very frequently IVT is very high. This is not clear to me and it deserves a clarification.

*It is indeed correct that $IVT_{pct}$ structures are not very dominant in relation to EPEs in these regions due to their definition. For comparison, the $IVT_{filt}$ structures are stronger associated with EPEs than $IVT_{pct}$ structures in these 'white' regions from Figure 12 (Figure 13), see also Figures 6b and S1b (S2b) from the original (revised) manuscript. The point we want to make here is that different combinations of intense moisture transport (>95$^{th}$ percentile) and/or wave breaking are of reduced importance in these regions as EPE formation is primarily controlled here by cyclones and/or warm conveyor belts that, admittedly, may have high IVT values, but not the intense moisture transport in which we are interested. Instead, different combinations of wave breaking and intense moisture transport show up as very important for EPE formation in many other regions such as the (semi)arid subtropics. In the revised manuscript we will include a statement that makes the reader aware that warm conveyor belts and extratropical cyclones usually go along with IVT values but not necessarily the unusually high IVT values (>95$^{th}$ percentile) as used for our definition of intense moisture transport (lines 769-701 of the revised manuscript);*

*"It should be noted here that cyclones and warm conveyor belts typically go along with high IVT values, but not necessarily the unusually high IVT values (>95$^{th}$ percentile) used for our definition of intense moisture transport."*

In addition to these few major points I have noted few technical corrections on the annoted pdf attached below.

***Author's response to the comments of Reviewer 2 as provided in the pdf***

*1. We removed the long and detailed sentence in the abstract and replaced it by a short sentence, see also the response above.*

*2. the typo "discussing" on line 445 of the original manuscript is corrected.*

*3. we revise the sentence on lines 462-464 of the original manuscript and explicitly say "… at some locations equatorward …" (lines 482-483 of the revised manuscript).*

*4. Figure 3 and caption. As far as I understand the comment, "daily" in the figure is spelled correctly which is visible when zooming in. In the caption, we replaced "IVT structures" by "IVT$_{pct}$ structures" for more clarity.*

***References***

[revised manuscript text omitted]

Not Highlight

| Page 17: [4] Formatted | Microsoft Office User | 08/12/2020 18:07:00 |
|---|---|---|

Not Highlight

| Page 17: [5] Deleted | Microsoft Office User | 23/12/2020 14:16:00 |
|---|---|---|

| Page 17: [5] Deleted | Microsoft Office User | 23/12/2020 14:16:00 |
|---|---|---|

| Page 17: [6] Deleted | Microsoft Office User | 30/11/2020 14:01:00 |
|---|---|---|

| Page 17: [6] Deleted | Microsoft Office User | 30/11/2020 14:01:00 |
|---|---|---|

| Page 17: [6] Deleted | Microsoft Office User | 30/11/2020 14:01:00 |
|---|---|---|

| Page 17: [6] Deleted | Microsoft Office User | 30/11/2020 14:01:00 |
|---|---|---|

| Page 17: [6] Deleted | Microsoft Office User | 30/11/2020 14:01:00 |
|---|---|---|